# Near Optimal Policy Optimization via REPS

**Aldo Pacchiano**
Microsoft Research
apacchiano@microsoft.com

**Jonathan Lee**
Stanford University
jnl@stanford.edu

**Peter L. Bartlett**
UC Berkeley
peter@berkeley.edu

**Ofir Nachum**
Google
ofirnachum@google.com

## Abstract

Since its introduction a decade ago, *relative entropy policy search* (REPS) has demonstrated successful policy learning on a number of simulated and real-world robotic domains, not to mention providing algorithmic components used by many recently proposed reinforcement learning (RL) algorithms. While REPS is well-known in the community, there exist no guarantees on its performance when using stochastic, gradient-based solvers. In this paper we aim to fill this gap by providing guarantees and convergence rates for the sub-optimality of a policy learned using first-order optimization methods applied to the REPS objective. We first consider the setting in which we are given access to exact gradients and demonstrate how near-optimality of the objective translates to near-optimality of the policy. We then consider the setting of stochastic gradients and introduce a technique that uses *generative* access to the underlying Markov decision process to compute parameter updates that maintain favorable convergence to the optimal regularized policy.

## 1 Introduction

Introduced by Peters et al. [23], *relative entropy policy search* (REPS) is an algorithm for learning agent policies in a reinforcement learning (RL) context. REPS has demonstrated successful policy learning in a variety of challenging simulated and real-world robotic tasks, encompassing table tennis [23], tether ball [12], beer pong [1], and ball-in-a-cup [8], among others. Beyond these direct applications of REPS, the mathematical tools and algorithmic components underlying REPS have inspired and been utilized as a foundation for a number of later algorithms, with their own collection of practical successes [13, 25, 20, 22, 15, 2, 18, 21].

At its core, the REPS algorithm is derived via an application of convex duality [22, 19], in which a Kullback Leibler (KL)-regularized version of the max-return objective in terms of state-action distributions is transformed into an $\mathrm{logsumexp}$ objective in terms of state-action *advantages* (*i.e.*, the difference of the value of the state-action pair compared to the value of the state alone, with respect to some learned state value function). If this dual objective is optimized, then the optimal policy of the original primal problem may be derived as a $\mathrm{softmax}$ of the state-action advantages. This basic derivation may be generalized, using any number of entropic regularizers on the original primal to yield a dual problem in the form of a convex function of advantages, whose optimal values may be transformed back to optimal regularized policies [7].

While the motivation for the REPS objective through the lens of convex duality is attractive, it leaves two main questions unanswered regarding the theoretical soundness of using such an approach. First, in practice, the dual objective in terms of advantages is likely not optimized fully. Rather, standard gradient-based solvers only provide guarantees on the *near-optimality* of a returned candidate solution. While convex duality asserts a relationship between primal and dual variables at the *exact* optimum,

it is far from clear whether a near-optimal dual solution will be guaranteed to yield a near-optimal primal solution, and this is further complicated by the fact that the primal candidate solution must be transformed to yield an agent policy.

The second of the two main practical difficulties is due to the form of the dual objective. Specifically, the form of the dual objective as a convex function of advantages frustrates the use of gradient-based solvers in stochastic settings. That is, the advantage of a state-action pair consists of an expectation over next states – an expectation over the transition function associated with the underlying Markov decision process (MDP). In practical settings, one does not have explicit knowledge of this transition function. Rather, one only has access to stochastic samples from this transition function, and so calculation of unbiased gradients of the REPS objective is not directly feasible.

In this paper, we provide solutions to these two main difficulties. To the first issue, we present guarantees on the near-optimality of a derived policy from dual variables optimized via a first-order gradient method, relying on a key property of the REPS objective that ensures near-optimality in terms of gradient norms. To the second issue, we propose and analyze a stochastic gradient descent procedure that makes use of a plug-in estimator of the REPS gradients. Under some mild assumptions on the MDP, our estimators need only sample transitions from a behavior policy rather than full access to a generative model (where one can uniformly sample transitions). We combine these results to yield high-probability convergence rates of REPS to a near-optimal policy. In this way, we show that REPS enjoys not only favorable practical performance but also strong theoretical guarantees.

## 2    Related Work

As REPS is a popular and influential work, there exist a number of previous papers that have studied its performance guarantees. These previous works predominantly study REPS as an iterative algorithm, where each step comprises of an exact optimization of the REPS objective and then the derived policy is used as the reference distribution for the KL regularization of the next step. This iterative scheme may be interpreted as a form of mirror descent or similar proximal algorithms [6], and this interpretation can provide guarantees on convergence to a near-optimal policy [29, 22]. However, because this approach assumes the ability to optimize the REPS objective exactly, it still suffers from the practical limitations discussed above; specifically (1) translation of near-optimality of advantages to near-optimality of the policy and (2) ability to compute unbiased gradients when one does not have explicit knowledge of the MDP dynamics. Our analysis attacks these issues head-on, providing guarantees on first-order optimization methods applied to the REPS objective. To maintain focus we do not consider iterative application of REPS, although extending our guarantees to the iterative setting is a promising direction for future research.

In a somewhat related vein, a number of works use REPS-inspired derivations to yield *dynamic programming* algorithms [13, 14, 26] and subsequently provide guarantees on the convergence of approximate dynamic programming in these settings. Our results focus on the use of REPS in a *convex programming* context, and optimizing these programs via standard gradient-based solvers.

The use of convex programming for RL in this way has recently received considerable interest. Works in this area typically propose to learn near-optimal policies through *saddle-point* optimization [9, 28, 10, 4, 11, 17]. Instead of solving the primal or dual max-return problem directly, these works optimize the Lagrangian in the form of a min-max bilinear problem. The Lagrangian form helps to mitigate the two main issues we identify with advantage learning, since (1) the candidate primal solution can be used to derive a policy in a significantly more direct fashion than using the candidate dual solution, and (2) the bilinear form of the Lagrangian is immediately amenable to stochastic gradient computation. In contrast to these works, our analysis focuses on learning exclusively in the dual (advantage) space. The first part of our results is most comparable to the work of [4], which proposes a saddle-point optimization with runtime $O(1/\epsilon)$, assuming access to known dynamics. While our results yield a $O(1/\epsilon^2)$ rate, we show that it can be achieved via optimizing the dual objective alone.

More similar to our work is the analysis of Bas-Serrano et al. [5], which considers an objective similar to REPS, but which is in terms of $Q$-values as opposed to state ($V$) values. Beyond these structural differences, our proof techniques also differ. For example, our result on the suboptimality of the policy derived from dual variables (Lemma 4), is arguably simpler from the analogous result in Bas-Serrano et al. [5], which uses a two-step process to first connect suboptimality of the dual variables to constraint violation in the primal, and then connects this to suboptimality of the policy.

# 3 Contributions

The main contributions of this paper are the following:

1. We prove several structural results regarding entropy regularized objectives for reinforcement learning and leverage them to prove convergence guarantees for Accelerated Gradient Descent on the dual (REPS) objective under mild assumptions on the MDP (see Theorem 2). For discounted MDPs we show that an $\epsilon$-optimal policy can be found after $\mathcal{O}(1/(1-\gamma)^2\epsilon^2)$ steps and an $\epsilon-$optimal regularized policy can be found in $\mathcal{O}(1/(1-\gamma)^2\epsilon)$ steps.

2. Similarly we show that a simple version of stochastic gradient descent using biased plug-in gradient estimators can be used to find an $\epsilon-$optimal policy after $\mathcal{O}(1/(1-\gamma)^8\epsilon^8)$ iterations (see Theorem 3) and an $\epsilon$-optimal regularized policy in $\mathcal{O}(1/(1-\gamma)^8\epsilon^4)$ steps. Although our rates are short of the ones achievable by alternating optimization methods, we are the first to show meaningful convergence guarantees for a purely dual approach based on on-policy access to samples from the underlying MDP.

3. In Appendix I, we extend our results beyond the REPS objective and consider the use of Tsallis Entropy regularizers. Similar to our results for the REPS objective we show that for discounted MDPs an $\epsilon-$optimal policy can be found after $\mathcal{O}(1/(1-\gamma)^2\epsilon^2)$ steps and an $\epsilon-$optimal regularized policy can be found in $\mathcal{O}(1/(1-\gamma)^2\epsilon)$ steps.

# 4 Background

In this section we review the basics of Markov decision processes and their Linear Programming primal and dual formulations (see section 4.1) and some facts about the geometry of convex functions.

## 4.1 RL as an LP

We consider a discounted Markov decision process (MDP) described by a tuple $\mathcal{M} = (\mathcal{S}, \mathcal{A}, P, \mathbf{r}, \boldsymbol{\mu}, \gamma)$, where $\mathcal{S}$ is a finite state space, $\mathcal{A}$ is a finite action space, $P$ is a transition probability matrix, $\mathbf{r}$ is a reward vector, $\boldsymbol{\mu}$ is an initial state distribution, and $\gamma \in (0,1)$ is a discount factor. We make the following assumption regarding the reward values $\{\mathbf{r}_{s,a}\}$.

**Assumption 1** (Unit rewards). *For all $s, a \in \mathcal{S} \times \mathcal{A}$, the rewards satisfy, $\mathbf{r}_{s,a} \in [0,1]$.*

The agent interacts with $\mathcal{M}$ via a policy $\pi : \mathcal{S} \to \Delta_{\mathcal{A}}$. The agent is initialized at a state $s_0$ sampled from an initial state distribution $\boldsymbol{\mu}$ and at time $k = 0, 1, \ldots$ it uses its policy to sample an action $a_k \sim \pi(s_k)$. The MDP provides an immediate reward $\mathbf{r}_{s_k, a_k}$ and transitions randomly to a next state $s_{k+1}$ according to probabilities $\mathbf{P}_a(s_{k+1}|s_k)$. Given a policy $\pi$ we define its infinite-horizon discounted reward as $V_\pi := \mathbb{E}^\pi\left[\sum_{k=0}^\infty \gamma^k \mathbf{r}_{s_k, a_k}\right]$, where we use $\mathbb{E}^\pi$ to denote the expectation over trajectories induced by the MDP $\mathcal{M}$ and policy $\pi$. In RL, the agent's objective is to find an optimal policy $\pi_\star$; that is, find a policy maximizing $V_\pi$ over all policy mappings $\pi : \mathcal{S} \to \Delta_{\mathcal{A}}$. We denote the optimal policy as $\pi_\star := \arg\max_\pi V_\pi$.

We now review the definitions of state value functions and visitation distributions:

**Definition 1.** *We define the value vector $\mathbf{v}^\pi \in \mathbb{R}^{|\mathcal{S}|}$ of a policy $\pi$ as $\mathbf{v}_s^\pi := \mathbb{E}^\pi\left[\sum_{k=0}^\infty \gamma^k r_{s_k, a_k}|s_0 = s\right]$.*

**Definition 2.** *Given a policy $\pi$ we define its state-action visitation distribution $\boldsymbol{\lambda}^\pi \in \mathbb{R}^{|\mathcal{S}| \times |\mathcal{A}|}$ as, $\boldsymbol{\lambda}_{s,a}^\pi := (1-\gamma)\mathbb{E}^\pi\left[\sum_{k=0}^\infty \gamma^k \mathbf{1}(s_k = s, a_k = a)\right]$. Notice that by definition $\sum_{s,a} \boldsymbol{\lambda}_{s,a} = 1$.*

We note that any vector of nonnegative entries $\boldsymbol{\lambda}$ may be used to define a policy $\pi_{\boldsymbol{\lambda}}$ as:

$$\pi_{\boldsymbol{\lambda}}(a|s) := \frac{\boldsymbol{\lambda}_{s,a}}{\sum_{a' \in \mathcal{A}} \boldsymbol{\lambda}_{s,a'}}. \tag{1}$$

Note that $\pi_{\boldsymbol{\lambda}^\pi} = \pi$, while the visitation distribution $\boldsymbol{\lambda}^{\pi_{\boldsymbol{\lambda}}}$ of $\pi_{\boldsymbol{\lambda}}$ is not necessarily $\boldsymbol{\lambda}$.

**Definition 3.** *Given a policy $\pi$ we define its state visitation distribution as, $\boldsymbol{\lambda}_s^\pi := (1-\gamma)\mathbb{E}^\pi\left[\sum_{k=0}^\infty \gamma^k \mathbf{1}(s_k = s)\right]$. Notice that $\boldsymbol{\lambda}_s^\pi = \sum_a \boldsymbol{\lambda}_{s,a}^\pi$ and $\boldsymbol{\lambda}_{s,a}^\pi = \boldsymbol{\lambda}_s^\pi \cdot \pi(a|s)$.*

The optimal visitation distribution $\boldsymbol{\lambda}^*$ is defined as $\boldsymbol{\lambda}^* := \arg\max_{\boldsymbol{\lambda}^\pi} \sum_{s,a} \boldsymbol{\lambda}^\pi_{s,a} \mathbf{r}_{s,a}$. It can be shown [24, 9] that solving for the optimal visitation distribution is equivalent to the following linear program:

$$\max_{\boldsymbol{\lambda}_{s,a} \in \Delta_{\mathcal{S} \times \mathcal{A}}} \sum_{s,a} \boldsymbol{\lambda}_{s,a} \mathbf{r}_{s,a}, \quad \text{s.t.} \sum_a \boldsymbol{\lambda}_{s,a} = (1-\gamma)\boldsymbol{\mu}_s + \gamma \sum_{s',a} \mathbf{P}_a(s|s') \boldsymbol{\lambda}_{s',a} \quad \forall s \in \mathcal{S}. \quad \text{(Primal-}\boldsymbol{\lambda}\text{)}$$

Where we write $\mathbf{P} \in \mathbb{R}^{|S||A| \times |S|}$ to denote the transition operator. Specifically, the $|\mathcal{S}|$ constraints of Primal-$\boldsymbol{\lambda}$ restrict any feasible $\boldsymbol{\lambda}$ to be the state-action visitations for some policy $\pi$ (given by $\pi_{\boldsymbol{\lambda}}$). The dual of this LP is given by,

$$\min_{\mathbf{v}} \ (1-\gamma) \sum_{s \in \mathcal{S}} \boldsymbol{\mu}_s \mathbf{v}_s, \quad \text{s.t.} \ 0 \geq \mathbf{A}^{\mathbf{v}}_{s,a} \quad \forall s \in \mathcal{S}, a \in \mathcal{A}, \quad \text{(Dual-}\mathbf{v}\text{)}$$

where $\mathbf{A}^{\mathbf{v}}_{s,a} = \mathbf{r}_{s,a} - \mathbf{v}_s + \gamma \sum_{s'} \mathbf{P}_a(s'|s)\mathbf{v}_{s'}$ is the advantage evaluated at $s, a \in \mathcal{S} \times \mathcal{A}$. It can be shown [24, 9] that the unique primal solution $\boldsymbol{\lambda}^*$ is exactly $\boldsymbol{\lambda}^{\pi*}$ and the unique dual solution $\mathbf{v}^*$ is $\mathbf{v}^{\pi*}$.

We finalize this section by defining the notion of suboptimality satisfied by the final policy produced by the algorithms that we propose.

**Definition 4.** *Let $\epsilon > 0$. We say that policy $\pi$ is $\epsilon$-optimal if $\max_{s \in \mathcal{S}} |\mathbf{v}^\pi_s - \mathbf{v}^{\pi*}_s| \leq \epsilon$.*

Our objective is to design algorithms such that for any $\epsilon > 0$, can return an $\epsilon$-optimal policy.

## 4.2 Regularized Policy Search

Following Belousov & Peters [7], we consider regularizing Primal-$\boldsymbol{\lambda}$ with a convex function $F : \Delta_{|\mathcal{S}| \times |\mathcal{A}|} \to \mathbb{R} \cup \{\infty\}$. The resulting regularized LP is given by,

$$\max_{\boldsymbol{\lambda}_{s,a} \in \Delta_{\mathcal{S} \times \mathcal{A}}} \sum_{s,a} \boldsymbol{\lambda}_{s,a} \mathbf{r}_{s,a} - F(\boldsymbol{\lambda}) := J_P(\boldsymbol{\lambda}), \quad \text{s.t.} \sum_a \boldsymbol{\lambda}_{s,a} = (1-\gamma)\boldsymbol{\mu}_s + \gamma \sum_{s',a} \mathbf{P}_a(s|s')\boldsymbol{\lambda}_{s',a}.$$

$$\text{(PrimalReg-}\boldsymbol{\lambda}\text{)}$$

Henceforth we denote the primal objective function as $J_P(\boldsymbol{\lambda}) = \sum_{s,a} \boldsymbol{\lambda}_{s,a} \mathbf{r}_{s,a} - F(\boldsymbol{\lambda})$. Any feasible $\boldsymbol{\lambda}$ that satisfies the $|\mathcal{S}|$ constraints in this regularized LP is the (true) state-action visitation distribution for some policy $\pi$; therefore, the optimal $\boldsymbol{\lambda}^*$ of this problem can be used to derive an optimal $F$-regularized max-return policy $\pi_{F,*} := \pi_{\boldsymbol{\lambda}^*}$. To simplify our derivations, we introduce the definition of the convex conjugate of a convex function, oftentimes referred to as the Fenchel conjugate:

**Definition 5** (Fenchel Conjugate). *Let $F : \mathcal{D} \to \mathbb{R}$ be a convex function over a convex domain $\mathcal{D} \subseteq \mathbb{R}^d$. We denote its $\mathcal{D}$-constrained Fenchel conjugate as $F^* : \mathbb{R}^n \to \mathbb{R}$ defined as $F^*(\mathbf{u}) = \max_{\mathbf{x} \in \mathcal{D}} \langle \mathbf{x}, \mathbf{u} \rangle - F(\mathbf{x})$.*

The dual $J_D$ of the regularized problem is given by the following optimization problem [7, 19]:

$$\min_{\mathbf{v}} J_D(\mathbf{v}) := (1-\gamma) \sum_s \mathbf{v}_s \boldsymbol{\mu}_s + F^*(\mathbf{A}^{\mathbf{v}}), \quad (2)$$

where $F^*$ is the $\Delta_{\mathcal{S} \times \mathcal{A}}$-constrained Fenchel conjugate of $F$. The vector quantity inside $F^*$ is known as the *advantage*. That is, it quantifies the advantage (the difference in estimated value) of taking an action $a$ at $s$, with respect to some state value function $\mathbf{v}$. Using Fenchel-Rockafellar duality, the optimal solution $\mathbf{v}^*$ of the dual function $J_D$ may be used to derive an optimal primal solution $\boldsymbol{\lambda}^*$ as:

$$\boldsymbol{\lambda}^* \in \nabla F^*(\mathbf{A}^{\mathbf{v}^*}). \quad (3)$$

---

**Algorithm 1** Relative Entropy Policy Search [Sketch].

**Input:** Initial iterate $\mathbf{v}_0$, accuracy level $\epsilon > 0$, gradient optimization algorithm $\mathcal{O}$.

1. Optimize the objective in 2 using $\mathcal{O}$ to yield a candidate dual solution $\hat{\mathbf{v}}^*$ where $F$ satisfies Equation 4.

2. Use the candidate dual solution to derive a candidate primal solution $\hat{\boldsymbol{\lambda}}^*$ using 3.

3. Extract a candidate policy $\pi_{\hat{\boldsymbol{\lambda}}^*}$ from $\hat{\boldsymbol{\lambda}}^*$ via Equation 1.

**Return:** $\pi_{\hat{\boldsymbol{\lambda}}^*}$.

---

Relative Entropy Policy Search (REPS) is derived by setting $F(\boldsymbol{\lambda}) := D_{\mathrm{KL}}(\boldsymbol{\lambda}\|\mathbf{q})$, the KL-divergence of $\boldsymbol{\lambda}$ from some reference distribution $\mathbf{q} \in \Delta_{|\mathcal{S}||\mathcal{A}|}$. The reader should think of $\mathbf{q}$ as the visitation distribution of a behavior policy. As we can see, the derivation we provide here further generalizes to arbitrary regularizers $F$. We focus on a specific $F$ given by

$$F(\boldsymbol{\lambda}) := \frac{1}{\eta} \sum_{s,a} \boldsymbol{\lambda}_{s,a} \left( \log\left( \frac{\boldsymbol{\lambda}_{s,a}}{\mathbf{q}_{s,a}} \right) - 1 \right), \tag{4}$$

for some scalar $\eta > 0$. In this case $F^* : \mathbb{R}^{|S|\times|A|} \to \mathbb{R}$ equals $F^*(\mathbf{u}) = \frac{1}{\eta} \log\left( \sum_{s,a} \exp\left(\eta\mathbf{u}_{s,a}\right) \mathbf{q}_{s,a} \right) + \frac{1}{\eta}$, its gradient satisfies $[\nabla F^*(\mathbf{u})]_{s,a} = \frac{\exp(\eta\mathbf{u}_{s,a})\mathbf{q}_{s,a}}{\sum_{s',a'} \exp(\eta\mathbf{u}_{s',a'})\mathbf{q}_{s',a'}}$ and therefore the dual function equals:

$$J_D(\mathbf{v}) := (1-\gamma)\sum_s \mathbf{v}_s\boldsymbol{\mu}_s + \frac{1}{\eta}\log\left( \sum_{s,a}\exp\left(\eta\mathbf{A}^{\mathbf{v}}_{s,a}\right)\mathbf{q}_{s,a} \right) + \frac{1}{\eta}, \qquad \text{(DualReg-v)}$$

And the dual problem equals the unconstrained minimization problem:

$$\min_{\mathbf{v}} J_D(\mathbf{v}) \tag{5}$$

The objective of REPS is to find the minimizer $\mathbf{v}^\star$ of DualReg-v (with regularization level $\eta$).

Algorithm 1 raises two practical issues discussed in Section 1. Specifically, optimization algorithms applied to REPS will typically only give guarantees on the near-optimality of $\hat{\mathbf{v}}^*$. We will need to translate near-optimality of $\hat{\mathbf{v}}^*$ to near-primal-optimality (w.r.t. $J_P(\boldsymbol{\lambda}^\star)$) of $\hat{\boldsymbol{\lambda}}^*$, and then translate that to near-optimality of the final returned policy $\pi_{\hat{\boldsymbol{\lambda}}^*}$. Secondly, first-order optimization of the REPS objective requires access to a gradient $\nabla_{\mathbf{v}} J_D(\mathbf{v})$, which involved computing $\nabla F^*(\mathbf{A}^{\mathbf{v}})$. Exact computation of this quantity is often infeasible in practical scenarios where one does not have access to $\mathbf{P}$, but rather only stochastic *generative* access to samples from $\mathbf{P}$. We show how to compute approximate (biased) gradients of $J_D(\mathbf{v})$ using samples from a distribution $\mathbf{q}_{s,a}$ (here thought of as a behavior policy) and how to use them to derive convergence rates for Relative Entropy Policy Search.

## 5 Relative Entropy Policy Search

We start by deriving some general results regarding the geometry of regularized linear programs. Our first result (Lemma 2) characterizes the smoothness properties of a regularized LP. This will prove crucial in later sections where we make use of this result to derive convergence rates for the REPS objective. We start by recalling the definitions of both strong convexity and smoothness of a function.

**Definition 6.** *A function* $f : \mathbb{R}^n \to \mathbb{R}$ *is* $\beta-$*strongly convex w.r.t norm* $\|\cdot\|$ *if* $f(\mathbf{x}) \geq f(\mathbf{y}) + \langle \nabla f(\mathbf{y}), \mathbf{x} - \mathbf{y} \rangle + \frac{\beta}{2}\|\mathbf{x} - \mathbf{y}\|^2$.

Let's also define smoothness:

**Definition 7.** *A function* $h$ *is* $\alpha-$*smooth*[1] *w.r.t. norm* $\|\cdot\|_\star$ *if:*

$$h(\mathbf{u}) \leq h(\mathbf{w}) + \langle \nabla h(\mathbf{w}), \mathbf{u} - \mathbf{w} \rangle + \frac{\alpha}{2}\|\mathbf{u} - \mathbf{w}\|_\star^2. \tag{6}$$

We will now characterize the smoothness properties of the dual of a regularized linear program. Let's start by considering the generic linear program:

$$\max_{\boldsymbol{\lambda}\in\mathcal{D}} \langle \mathbf{r}, \boldsymbol{\lambda} \rangle, \quad \text{s.t. } \mathbf{E}\boldsymbol{\lambda} = \mathbf{b},$$

where $\mathbf{r} \in \mathbb{R}^n$, $\mathbf{E} \in \mathbb{R}^{m\times n}$, and $\mathbf{b} \in \mathbb{R}^m$ and $\mathcal{D}$ is a convex domain. Let's regularize this objective using a function $F$ that is $\beta$-strongly convex with respect to norm $\|\cdot\|$:

$$\max_{\boldsymbol{\lambda}\in\mathcal{D}} \langle \mathbf{r}, \boldsymbol{\lambda} \rangle - F(\boldsymbol{\lambda}), \quad \text{s.t. } \mathbf{E}\boldsymbol{\lambda} = \mathbf{b}. \qquad \text{(RegLP)}$$

---

[1]Smoothness is independent of the convexity properties of $h$.

The Lagrangian of problem RegLP is given by $g_L(\boldsymbol{\lambda}, \mathbf{v}) = \langle \mathbf{r}, \boldsymbol{\lambda} \rangle - F(\boldsymbol{\lambda}) + \sum_{i=1}^{m} \mathbf{v}_i \left( \mathbf{b}_i - (\mathbf{E}\boldsymbol{\lambda})_i \right)$.
Therefore, the dual function $g_D : \mathbb{R}^m \to \mathbb{R}$ with respect to the original primal regularized LP is,

$$g_D(\mathbf{v}) = \langle \mathbf{v}, \mathbf{b} \rangle + \max_{\boldsymbol{\lambda} \in \mathcal{D}} \langle \boldsymbol{\lambda}, \mathbf{r} - \mathbf{v}^\top \mathbf{E} \rangle - F(\boldsymbol{\lambda}) = \langle \mathbf{v}, \mathbf{b} \rangle + F^*(\mathbf{r} - \mathbf{v}^\top \mathbf{E}),$$

where the last equality follows from the definition of the Fenchel conjugate of $F$. It is possible to relate the smoothness properties of $F^*$ with the strong convexity of $F$. A crucial result that we will use in our results is the following:

**Lemma 1.** *If $F$ is $\beta$-strongly convex w.r.t. $\| \cdot \|$ over $\mathcal{D}$ then $F^*$ is $\frac{1}{\beta}$-smooth w.r.t the dual $\| \cdot \|_\star$.*

Definitions 6 and 7 are stated in terms of a generic norm $\| \cdot \|$ and its dual $\| \cdot \|_\star$. When applied to the REPS objective in Equation 2, using these general norm definitions of smoothness and strong convexity allow us to obtain guarantees with a milder dependence on $\mathcal{S}$ and $\mathcal{A}$ than would be possible if we were to use their $\ell_2$ norm characterization instead. We can use the result of Lemma 1 to characterize the smoothness properties of the dual function $J_D$ of a generic regularized LP.

**Lemma 2.** *Consider the regularized LP RegLP with $\mathbf{r} \in \mathbb{R}^n$, $\mathbf{E} \in \mathbb{R}^{m \times n}$, $\mathbf{b} \in \mathbb{R}^m$, and where $F$ is $\beta$-strongly convex w.r.t. norm $\| \cdot \|$. The dual function $g_D : \mathbb{R}^m \to \mathbb{R}$ of this regularized LP is $\frac{\|\mathbf{E}\|_{\cdot,\star}^2}{\beta}$-smooth w.r.t. to the dual norm $\| \cdot \|_\star$, where we use $\|\mathbf{E}\|_{\cdot,\star}$ to denote the $\| \cdot \|$ norm over the $\| \cdot \|_\star$ norm of $\mathbf{E}'s$ rows.*

As a consequence of Lemma 2 we can bound the smoothness parameter of $J_D$ in the REPS objective:

**Lemma 3.** *The dual function $J_D(\mathbf{v})$ is $(|\mathcal{S}| + 1)\eta$-smooth in the $\| \cdot \|_\infty$ norm.*

## 5.1 Structural results for the REPS objective

Armed with Lemma 2 we are ready to derive some useful structural properties of the REPS objective. In this section we present two main results. First we show that under some mild assumptions it is possible to relate the gradient magnitude of any candidate solution to $J_D$ with its suboptimality gap and second, we show an $l_\infty$ bound for the norm of the optimal dual solution $\mathbf{v}^\star$. For most of the analysis we make the following assumptions:

**Assumption 2.** *There is $\beta > 0$ such that $\mathbf{q}_{s,a} \geq \beta \quad \forall s, a \in \mathcal{S} \times \mathcal{A}$.*

We introduce the following assumption on the discounted state visitation distribution of arbitrary policies $\pi$ in the MDP, paraphrased from Wang [27]:

**Assumption 3.** *There exists $\rho > 0$ such that for any policy $\pi$, the discounted state visitation distribution $\boldsymbol{\lambda}^\pi$ defined as $\boldsymbol{\lambda}_s^\pi = \sum_a \boldsymbol{\lambda}_{s,a}^\pi$ satisfies $\boldsymbol{\lambda}_s^\pi \geq \rho$ for all states $s \in \mathcal{S}$.*

Suppose we have a candidate dual solution $\widetilde{\mathbf{v}}$ for $J_D(\mathbf{v})$ in DualReg-v with its corresponding candidate primal solution $\widetilde{\boldsymbol{\lambda}} = \frac{\exp(\eta \mathbf{A}^{\check{\mathbf{v}}}) \cdot \mathbf{q}}{\widetilde{Z}}$ where the operators $\exp$ and $\cdot$ act pointwise and $\widetilde{Z} = \sum_{a,s} \exp(\eta \mathbf{A}^{\check{\mathbf{v}}}) \mathbf{q}_{s,a}$. We denote the corresponding candidate policy (computed using Equation 1) associated with $\widetilde{\mathbf{v}}$ as $\widetilde{\pi}(a|s)$. This candidate policy induces a discounted visitation distribution $\boldsymbol{\lambda}^{\widetilde{\pi}}$ that may be substantially different from $\widetilde{\boldsymbol{\lambda}}$. We now show that it is possible to control the deviation of primal objective value of $\boldsymbol{\lambda}^{\widetilde{\pi}}$ from $J_P(\boldsymbol{\lambda})$ in terms of $\|\nabla J_D(\widetilde{\mathbf{v}})\|_1$:

**Lemma 4.** *Let $\widetilde{\mathbf{v}} \in \mathbb{R}^{|\mathcal{S}|}$ be arbitrary and let $\widetilde{\boldsymbol{\lambda}}$ be its corresponding candidate primal variable. If $\|\nabla_{\mathbf{v}} J_D(\widetilde{\mathbf{v}})\|_1 \leq \epsilon$ and Assumptions 2 and 3 hold then whenever $|\mathcal{S}| \geq 2$:*

$$J_P(\boldsymbol{\lambda}^{\widetilde{\pi}}) \geq J_P(\boldsymbol{\lambda}_\eta^\star) - \epsilon \left( \frac{1+c}{1-\gamma} + \|\widetilde{\mathbf{v}}\|_\infty \right),$$

*where $c = \frac{1+\log(\frac{1}{\rho^3 \beta})}{\eta}$ and $\boldsymbol{\lambda}_\eta^\star$ is the $J_P$ optimum.*

We finish this section by proving a bound on the norm of the dual variables. This bound will inform our optimization algorithms as it will allow us to set up the right constraints.

**Lemma 5.** *Under Assumptions 1, 2 and 3, the optimal dual variables are bounded as*

$$\|\mathbf{v}^*\|_\infty \leq \frac{1}{1-\gamma} \left( 1 + \frac{\log \frac{|\mathcal{S}||\mathcal{A}|}{\beta \rho}}{\eta} \right) =: D. \tag{7}$$

From now on we use the notation $D$ to refer to the quantity on the RHS of Equation 7.

## 5.2 Convergence rates

As a warm up we derive convergence rates for the case when we have access to exact knowledge of the transition dynamics $\mathbf{P}$ and therefore exact gradients. We analyze the effects of using Accelerated Gradient Descent (see Section F for a full description of the algorithm) on the REPS objective $J_D(\mathbf{v})$.

**Theorem 1** (Accelerated Gradient Descent for general norms. Theorem 4.1 in Allen-Zhu & Orecchia [3]). *Let $D_\star$ be an upper bound to $\|\mathbf{x}_0, \mathbf{x}_\star\|_2^2$. Given an $\alpha-$smooth function $h$ w.r.t. the $\|\cdot\|_\star$ norm over domain $\mathcal{D}$, then $T$ iterations of Algorithm 4 ensure $h(\mathbf{y}_t) - h(\mathbf{x}^\star) \leq \frac{4\alpha D_*}{T^2}$.*

We want to find almost optimal solutions (in function value). Let's define an $\epsilon-$optimal solution:

**Definition 8.** *Let $\epsilon > 0$. We say that $\mathbf{x}$ is an $\epsilon-$optimal solution of an $\alpha-$smooth function $h : \mathbb{R}^d \to \mathbb{R}$ if $h(\mathbf{x}) - h(\mathbf{x}^\star) \leq \epsilon$, where $h(\mathbf{x}^\star) = \min_{\mathbf{x} \in \mathbb{R}^d} h(\mathbf{x})$.*

We can also show the following bound on the gradient norm for any $\epsilon-$optimal solutions of $h$.

**Lemma 6.** *If $\mathbf{x}$ is an $\epsilon-$optimal solution for the $\alpha-$smooth function $h : \mathbb{R}^d \to \mathbb{R}$ w.r.t. norm $\|\cdot\|_\star$ then the gradient of $h$ at $\mathbf{x}$ satisfies $\|\nabla h(\mathbf{x})\| \leq \sqrt{2\alpha\epsilon}$.*

When $h = J_D$ the DualReg-v function in the reinforcement learning setting, we set $\|\cdot\|_\star = \|\cdot\|_\infty$ and $\|\cdot\| = \|\cdot\|_1$. We are ready to prove convergence guarantees for Algorithm 4 when applied to the objective $J_D$.

**Lemma 7.** *Let Assumptions 1, 2 and 3 hold. Let $\mathcal{D} = \{\mathbf{v} \text{ s.t. } \|\mathbf{v}\|_\infty \leq D\}$ and $c' = \frac{\log \frac{|\mathcal{S}||A|}{\beta\rho}}{\eta}$. After $T$ steps of Algorithm 4, the objective function $J_D$ evaluated at the iterate $\mathbf{v}_T = y_T$ satisfies:*

$$J_D(\mathbf{v}_T) - J_D(\mathbf{v}^*) \leq 4\eta(|\mathcal{S}| + 1)^2 \frac{(1 + c')^2}{(1 - \gamma)^2 T^2},$$

*Proof.* This result follows simply by invoking the guarantees of Theorem 1, making use of the fact that $J_D$ is $(|\mathcal{S}| + 1)\eta-$smooth as proven by Lemma 3, observing that as a consequence of Lemma 5, $\mathbf{v}^\star \in \mathcal{D}$ and using the inequality $\|\mathbf{x}\|_2^2 \leq |\mathcal{S}|\|\mathbf{x}\|_\infty^2$ for $\mathbf{x} \in \mathbb{R}^{|\mathcal{S}|}$. $\square$

Lemma 7 can be easily turned into the following guarantee on the final dual function value:

**Corollary 1.** *Let $\epsilon > 0$. If Algorithm 4 is ran for at least $T$ rounds where $T \geq 2\eta^{1/2}(|\mathcal{S}|+1)\frac{(1+c')}{(1-\gamma)\sqrt{\epsilon}}$ then $\mathbf{v}_T$ is an $\epsilon-$optimal solution for the dual objective $J_D$.*

If $T$ satisfies the conditions of Corollary 1 a simple use of Lemma 6 allows us to bound the $\|\cdot\|_1$ norm of the dual function's gradient at $\mathbf{v}_T$:

$$\|\nabla J_D(\mathbf{v}_T)\|_1 \leq \sqrt{2(|\mathcal{S}| + 1)\eta\epsilon}$$

If we denote as $\pi_T$ to be the policy induced by $\boldsymbol{\lambda}^{\mathbf{v}_T}$, and $\boldsymbol{\lambda}_\eta^\star$ is the candidate dual solution corresponding to $\mathbf{v}^\star$. A simple application of Lemma 4 yields:

$$J_P(\boldsymbol{\lambda}^{\pi_T}) \geq J_P(\boldsymbol{\lambda}_\eta^\star) - \frac{\sqrt{2(|\mathcal{S}| + 1)\eta\epsilon}}{1 - \gamma}\left(2 + \frac{1 + \log\frac{|\mathcal{S}||\mathcal{A}|}{\beta^2\rho^4}}{\eta}\right)$$

The following is the equivalent version of optimality for regularized objectives:

**Definition 9.** *Let $\epsilon > 0$. We say $\tilde{\pi}$ is an $\epsilon-$optimal regularized policy if $J_P(\boldsymbol{\lambda}^{\tilde{\pi}}) \geq J_P(\boldsymbol{\lambda}_\eta^\star) - \epsilon$.*

This leads us to the main result of this section:

**Corollary 2.** *For any $\xi > 0$, and let $c'' = \frac{1 + \log\frac{|\mathcal{S}||\mathcal{A}|}{\beta^2\rho^4}}{\eta}$. If $T \geq 4\eta(|\mathcal{S}| + 1)^{3/2}\frac{(2 + c'')^2}{(1-\gamma)^2\xi}$ then $J_P(\boldsymbol{\lambda}^{\pi_T}) \geq J_P(\boldsymbol{\lambda}_\eta^\star) - \xi$.*

Thus Algorithm 4 achieves an $\mathcal{O}(1/(1 - \gamma)^2\epsilon)$ rate to an $\epsilon-$optimal regularized policy. We proceed to show that an appropriate choice for $\eta$ can be leveraged to obtain an $\epsilon-$optimal policy.

**Theorem 2.** *For any $\epsilon > 0$, let $\eta = \frac{1}{2\epsilon\log(\frac{|\mathcal{S}||\mathcal{A}|}{\beta})}$. If $T \geq (|\mathcal{S}| + 1)^{3/2}\frac{(2+c'')^2}{(1-\gamma)^2\epsilon^2}$, then $\pi_T$ is an $\epsilon-$optimal policy.*

The main difficulty in deriving the guarantees of Theorem 2 lies in the need to translate the function value optimality guarantees of Accelerated Gradient Descent into $\epsilon$-optimality guarantees for the candidate policy $\pi_T$. This is where our results from Lemma 4 have proven fundamental. It remains to show that it is possible to obtain an $\epsilon-$optimal policy when access to the true model is only via samples.

## 6 Stochastic Gradients

In this section we show how to obtain stochastic (albeit biased) gradient estimators $\widehat{\nabla}_{\mathbf{v}} J_D(\mathbf{v})$ for $\nabla_{\mathbf{v}} J_D(\mathbf{v})$ (see Algorithm 2). We use $\widehat{\nabla}_{\mathbf{v}} J_D(\mathbf{v})$ to perform biased stochastic gradient descent steps on $J_D(\mathbf{v})$ (see Algorithm 3). In Lemma 8 we prove guarantees for the bias and variance of this estimator and show rates for convergence in function value to the optimum of $J_D(\mathbf{v})$ in Lemma 10. We turn these results into guarantees for $\epsilon-$optimality of the final candidate policy in Theorem 3. Let's start by noting that:

$$(\nabla_{\mathbf{v}} J_D(\mathbf{v}))_s = (1 - \gamma)\boldsymbol{\mu}_s + \mathbb{E}_{(s',a,s'') \sim \mathbf{q} \times \mathbf{P}_a(\cdot|s')}\Big[\mathbf{B}_{s',a}^{\mathbf{v}}\left(\gamma\mathbf{1}(s'' = s) - \mathbf{1}(s' = s)\right)\Big],$$

Where $\mathbf{B}_{s,a}^{\mathbf{v}} = \frac{\exp(\eta \mathbf{A}_{s,a}^{\mathbf{v}})}{\mathbf{Z}}$ and $\mathbf{Z} = \sum_{s,a} \exp\left(\eta \mathbf{A}_{s,a}^{\mathbf{v}}\right) \mathbf{q}_{s,a}$. We will make use of this characterization to devise a plug-in estimator for this quantity:

---

**Algorithm 2** Biased Gradient Estimator

---

**Input** Number of samples $t$.
Collect samples $\{(s_\ell, a_\ell, s_\ell')\}_{\ell=1}^t$ such that $(s_\ell, a_\ell) \sim \mathbf{q}$ while $s_\ell' \sim \mathbf{P}_{a_\ell}(\cdot|s_\ell)$
**for** $(s,a) \in \mathcal{S} \times \mathcal{A}$ **do**
  Build empirical estimators $\widehat{\mathbf{A}}^{\mathbf{v}}(t) \in \mathbb{R}^{|\mathcal{S}| \times |\mathcal{A}|}$ and $\widehat{\mathbf{q}}(t) \in \mathbb{R}^{|\mathcal{S}| \times |\mathcal{A}|}$.
  Compute estimators $\widehat{\mathbf{B}}_{s,a}^{\mathbf{v}}(t) = \frac{\exp(\eta \widehat{\mathbf{A}}_{s,a}^{\mathbf{v}}(t))}{\widehat{\mathbf{Z}}(t)}$.
  Where $\widehat{\mathbf{Z}}(t) = \sum_{s,a} \exp(\eta \widehat{\mathbf{A}}_{s,a}^{\mathbf{v}}(t))\widehat{\mathbf{q}}_{s,a}(t)$.
**end**
Produce a final sample $(s_{t+1}, a_{t+1}) \sim \mathbf{q}$ and $s_{t+1}' \sim \mathbf{P}_{a_{t+1}}(\cdot|s_{t+1})$.
Compute $\widehat{\nabla}_{\mathbf{v}} J_D(\mathbf{v})$ such that: $\left(\hat{\triangledown}_{\mathbf{v}} J_D(\mathbf{v})\right)_s = (1-\gamma)\boldsymbol{\mu}_s + \widehat{\mathbf{B}}_{s_{t+1},a_{t+1}}(t)\left(\gamma\mathbf{1}(s_{t+1}' = s) - \mathbf{1}(s_{t+1} = s)\right)$.
**Output:** $\widehat{\nabla}_{\mathbf{v}} J_D(\mathbf{v})$.

---

We now proceed to bound the bias of this estimator:

**Lemma 8.** *Let* $\delta, \xi \in (0,1)$ *with* $\xi \leq \min(\beta, \frac{1}{4})$. *With probability at least* $1 - \delta$ *for all* $t \in \mathbb{N}$ *such that* $\frac{t}{\ln\ln(2t)} \geq \frac{120(\ln\frac{41.6|\mathcal{S}||\mathcal{A}|}{\delta} + 1)}{\beta \xi^2} \max\left(480\eta^2\gamma^2\|\mathbf{v}\|_\infty^2, 1\right)$, *the plugin estimator* $\widehat{\nabla}_{\mathbf{v}} J_D(\mathbf{v})$ *satisfies:*

$$\max_{u \in \{1,2,\infty\}} \|\hat{\mathbf{g}} - \mathbb{E}_{t+1}[\hat{\mathbf{g}}]\|_u \leq \frac{8}{\beta} \; ; \quad \max_{u \in \{1,2,\infty\}} \|\mathbb{E}_{t+1}[\hat{\mathbf{g}}] - \mathbf{g}\|_u \leq 8\xi \; ;$$

$$\mathbb{E}\left[\|\hat{\mathbf{g}} - \mathbb{E}_{t+1}[\hat{\mathbf{g}}]\|_2^2 \Big| \widehat{\mathbf{B}}^{\mathbf{v}}(t)\right] \leq \frac{8}{\beta},$$

*where* $\hat{\mathbf{g}} = \widehat{\nabla}_{\mathbf{v}} J_D(\mathbf{v})$, $\mathbf{g} = \nabla_{\mathbf{v}} J_D(\mathbf{v})$, *and* $\mathbb{E}_{t+1}[\cdot] = \mathbb{E}_{s_{t+1},a_{t+1},s_{t+1}'}[\cdot|\widehat{\mathbf{B}}^{\mathbf{v}}(t)]$.

We will now make use of Lemma 8 along with the following guarantee for projected Stochastic Gradient Descent to prove convergence guarantees for Algorithm 3.

---

**Algorithm 3** Biased Stochastic Gradient Descent

---

**Input** Desired accuracy $\epsilon$, learning rates $\{\tau_t\}_{t=1}^\infty$, and number-of-samples function $n : \mathbb{N} \to \mathbb{N}$.
Initialize $\mathbf{v}_0 = \mathbf{0}$ for $t = 1, \cdots, T$ **do**
  Get $\widehat{\nabla}_{\mathbf{v}} J_D(\mathbf{v})$ with $n(t)$ samples via Algorithm 2.
  Perform update: $\mathbf{v}_t' \leftarrow \mathbf{v}_t - \tau_t \widehat{\nabla}_{\mathbf{v}} J_D(\mathbf{v})$; $\mathbf{v}_t \leftarrow \Pi_{\mathcal{D}}(\mathbf{v}_t')$, where $\Pi_{\mathcal{D}}$ denotes the projection to $\mathcal{D} = \{\mathbf{v} \text{ s.t. } \|\mathbf{v}\|_\infty \leq D\}$.
**end**
**Output:** $\mathbf{v}_T$.

---

The following holds:

**Lemma 9.** *Let $f : \mathbb{R}^d \to \mathbb{R}$ be an $L-$smooth function. We consider the following update:*

$$\mathbf{x}'_{t+1} = \mathbf{x}_t - \tau\left(\nabla f(\mathbf{x}_t) + \boldsymbol{\epsilon}_t + \mathbf{b}_t\right) \;;\quad \mathbf{x}_{t+1} = \Pi_{\mathcal{D}}(\mathbf{x}'_{t+1}).$$

*If $\tau \leq \frac{2}{L}$ then:*

$$f(\mathbf{x}_{t+1}) - f(\mathbf{x}_\star) \leq \frac{\|\mathbf{x}_t - \mathbf{x}_\star\|^2 - \|\mathbf{x}_{t+1} - \mathbf{x}_\star\|^2}{2\tau} + 2\tau\|\nabla f(\mathbf{x}_t)\|^2 + 5\tau\|\mathbf{b}_t\|^2 + 5\tau\|\boldsymbol{\epsilon}_t\|^2$$
$$+ \|\mathbf{b}_t\|_1\|\mathbf{x}_t - \mathbf{x}_\star\|_\infty - \langle\boldsymbol{\epsilon}_t, \mathbf{x}_t - \mathbf{x}_\star\rangle.$$

Lemma 8 implies the following guarantee for the following projected stochastic gradient algorithm with biased gradients $\widehat{\nabla}_{\mathbf{v}} J_D(\mathbf{v})R$:

**Lemma 10.** *We assume $\eta \geq \frac{4}{\beta}$. Set $\xi_t = \frac{8|\mathcal{S}|\eta D}{\sqrt{t}}$ and $\tau_t = \frac{1}{16|\mathcal{S}|\eta\sqrt{t}}$. If we take $t$ gradient steps using $n(t)$ samples from $\mathbf{q} \times \mathbf{P}$ (possibly reusing the samples for multiple gradient computations) with $n(t)$ satisfying $n(t) \geq \frac{525t\left(\ln\frac{100|\mathcal{S}||\mathcal{A}|t^2}{\delta}+1\right)^3}{\beta|\mathcal{S}|^2}$. Then for all $t \geq 1$ we have that with probability at least $1 - 3\delta$ and simultaneously for all $t \in \mathbb{N}$ such that $t \geq \frac{64|\mathcal{S}|^2\eta^2 D^2}{\beta}$:*

$$J_D\left(\frac{1}{t}\sum_{\ell=1}^{t}\mathbf{v}_\ell\right) \leq J_D(\mathbf{v}_\star) + \widetilde{\mathcal{O}}\left(\frac{D^2|\mathcal{S}|\eta}{\sqrt{t}}\right).$$

Lemma 10 implies that making use of $N$ samples it is possible to find a candidate $\bar{\mathbf{v}}_N$ such that $J_D(\bar{\mathbf{v}}_N) \leq J_D(\mathbf{v}_\star) + \widetilde{\mathcal{O}}\left(\frac{D^2\eta}{\beta\sqrt{N}}\right)$. This in turn implies by a simple use of Lemma 6 that $\|\nabla J_D(\bar{\mathbf{v}}_N)\|_1 \leq \widetilde{\mathcal{O}}\left(\frac{|\mathcal{S}|^{1/2}D\eta}{\sqrt{\beta}N^{1/4}}\right)$. If we denote as $\bar{\pi}_N$ to the policy induced by $\boldsymbol{\lambda}^{\bar{\mathbf{v}}_N}$, a simple application of Lemma 4 yields:

$$J_P(\boldsymbol{\lambda}^{\bar{\pi}_N}) \geq J_P(\boldsymbol{\lambda}^\star_\eta) - \widetilde{\mathcal{O}}\left(\frac{|\mathcal{S}|^{1/2}D\eta}{(1-\gamma)\sqrt{\beta}N^{1/4}}\right)$$

Thus Algorithm 3 achieves an $\mathcal{O}(1/(1-\gamma)^8\epsilon^4)$ rate of convergence to an $\epsilon-$optimal regularized policy. We proceed to we can set $\eta$ to obtain an $\epsilon-$optimal policy:

**Theorem 3** (Informal)**.** *For any $\epsilon > 0$ let $\eta = \frac{1}{2\epsilon\log(\frac{|\mathcal{S}||\mathcal{A}|}{\beta})}$. If $N \geq \widetilde{\mathcal{O}}\left(\frac{1}{\epsilon^8(1-\gamma)^8\beta^2}\right)$, then with probability at least $1 - \delta$ it is possible to find a candidate $\bar{\mathbf{v}}_N$ such that $\bar{\pi}_N$ is an $\epsilon-$optimal policy.*

## 7 Conclusion

This work presents an analysis of first-order optimization methods for the REPS objective in reinforcement learning. We prove convergence rates of $O(1/\epsilon^2)$ for accelerated gradient descent on the dual of the KL-regularized max-return LP in the case of a known transition function with convergence rate. For the unknown case, we propose a biased stochastic gradient descent method relying on samples from behavior policy and show that it converges to an optimal policy with rate $O(1/\epsilon^8)$. There are several interesting questions that remain open. First, while directly optimizing the dual via gradient methods is convenient from an algorithmic perspective, prior unregularized saddle-point methods have been shown to achieve a faster $O(1/\epsilon)$ convergence [4]. An important open direction is thus to understand if faster rates are possible in order to bridge this gap, or if optimizing the regularized dual directly is fundamentally limited. Second, we only considered MDPs with finite state and action spaces. It is therefore of interest to see if these ideas readily extend to infinite or very large spaces through function approximation.

## Acknowledgments and Disclosure of Funding

This research was supported by the Google-BAIR Commons program. JL is supported by NSF GRFP.

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
