# A  Appendix

## Contents of main article and appendix

# B  Geometry of regularized Linear Programs

We start by fleshing out the connection between strong convexity and smoothness charted in Lemma 1:

**Lemma 1.** *If $F$ is $\beta$-strongly convex w.r.t. $\|\cdot\|$ over $\mathcal{D}$ then $F^*$ is $\frac{1}{\beta}$-smooth w.r.t the dual $\|\cdot\|_\star$.*

*Proof.* Let $\mathbf{u}, \mathbf{w} \in \mathbb{R}^n$ and $\mathbf{x}, \mathbf{y} \in \mathcal{D}$ be such that $\nabla F^*(\mathbf{u}) = \mathbf{x}$ and $\nabla F^*(\mathbf{w}) = \mathbf{y}$. By definition this also implies that:

$$\langle \nabla F(\mathbf{x}) - \mathbf{u}, \mathbf{z}_1 - \mathbf{x} \rangle \geq 0, \quad \forall \mathbf{z} \in \mathcal{D}$$
$$\langle \nabla F(\mathbf{y}) - \mathbf{w}, \mathbf{z}_2 - \mathbf{y} \rangle \geq 0, \quad \forall \mathbf{z} \in \mathcal{D}$$

Setting $\mathbf{z}_1 = \mathbf{y}$ and $\mathbf{z}_2 = \mathbf{x}$ along with the definition of $\mathbf{x}, \mathbf{y}$ and summing the two inequalities:

$$\langle \nabla F(\mathbf{x}) - \nabla F(\mathbf{y}), \mathbf{y} - \mathbf{x} \rangle \geq \langle \nabla F^*(\mathbf{w}) - \nabla F^*(\mathbf{u}), \mathbf{u} - \mathbf{w} \rangle. \tag{8}$$

By strong convexity of $F$ over domain $\mathcal{D}$ we see that:

$$F(\mathbf{x}) \geq F(\mathbf{y}) + \langle \nabla F(\mathbf{y}), \mathbf{x} - \mathbf{y} \rangle + \frac{\beta}{2} \|\mathbf{x} - \mathbf{y}\|^2$$

$$F(\mathbf{y}) \geq F(\mathbf{x}) + \langle \nabla F(\mathbf{x}), \mathbf{y} - \mathbf{x} \rangle + \frac{\beta}{2} \|\mathbf{x} - \mathbf{y}\|^2$$

Summing both inequalities yields:

$$\beta \|\mathbf{x} - \mathbf{y}\|^2 \leq \langle \nabla F(\mathbf{x}) - \nabla F(\mathbf{y}), \mathbf{x} - \mathbf{y} \rangle$$

Plugging in the definition of $\mathbf{u}$ and $\mathbf{w}$ along with inequality 8:

$$\beta \|\nabla F^*(\mathbf{u}) - \nabla F^*(\mathbf{w})\|^2 \leq \langle \mathbf{u} - \mathbf{w}, \nabla F^*(\mathbf{u}) - \nabla F^*(\mathbf{w}) \rangle \overset{(i)}{\leq} \|\mathbf{u} - \mathbf{w}\|_* \|\nabla F^*(\mathbf{u}) - \nabla F^*(\mathbf{w})\|.$$

Where inequality $(i)$ holds by Cauchy-Schwartz and consequently:

$$\|\nabla F^*(\mathbf{u}) - \nabla F^*(\mathbf{w})\| \leq \frac{1}{\beta} \|\mathbf{u} - \mathbf{w}\|_*$$

By the mean value theorem there exists $\mathbf{z} \in [\mathbf{u}, \mathbf{w}]$:

$$
\begin{aligned}
F^*(\mathbf{u}) &= F^*(\mathbf{w}) + \langle \nabla F^*(\mathbf{z}), \mathbf{w} - \mathbf{u} \rangle \\
&= F^*(\mathbf{w}) + \langle \nabla F^*(\mathbf{w}), \mathbf{w} - \mathbf{u} \rangle + \langle \nabla F^*(\mathbf{z}) - \nabla F^*(\mathbf{w}), \mathbf{w} - \mathbf{u} \rangle \\
&\leq F^*(\mathbf{w}) + \langle \nabla F^*(\mathbf{w}), \mathbf{w} - \mathbf{u} \rangle + \|\nabla F^*(\mathbf{z}) - \nabla F^*(\mathbf{w})\| \|\mathbf{w} - \mathbf{u}\|_* \\
&\leq F^*(\mathbf{w}) + \langle \nabla F^*(\mathbf{w}), \mathbf{w} - \mathbf{u} \rangle + \frac{1}{\beta} \|\mathbf{z} - \mathbf{w}\|_* \|\mathbf{w} - \mathbf{u}\|_* \\
&\leq F^*(\mathbf{w}) + \langle \nabla F^*(\mathbf{w}), \mathbf{w} - \mathbf{u} \rangle + \frac{1}{\beta} \|\mathbf{w} - \mathbf{u}\|_*^2
\end{aligned}
$$

Which concludes the proof.

$\square$

The proof of Lemma 1 yields the following useful result that characterizes the smoothness properties of the dual function in a regularized LP:

## B.1  Proof of Lemma 2

**Lemma 2.** *Consider the regularized LP RegLP with $\mathbf{r} \in \mathbb{R}^n$, $\mathbf{E} \in \mathbb{R}^{m \times n}$, $\mathbf{b} \in \mathbb{R}^m$, and where $F$ is $\beta$-strongly convex w.r.t. norm $\|\cdot\|$. The dual function $g_D : \mathbb{R}^m \to \mathbb{R}$ of this regularized LP is $\frac{\|\mathbf{E}\|_{\cdot,\star}^2}{\beta}$-smooth w.r.t. to the dual norm $\|\cdot\|_\star$, where we use $\|\mathbf{E}\|_{\cdot,\star}$ to denote the $\|\cdot\|$ norm over the $\|\cdot\|_\star$ norm of $\mathbf{E}'s$ rows.*

*Proof.* Recall that:
$$g_D(v) = \langle v, b \rangle + F^*(r - v^\top E).$$

Notice that:

$$\nabla_v g_D(v) = b + E \nabla F^*(r - v^\top E).$$

And therefore for any two $v_1, v_2$:

$$
\begin{aligned}
\|\nabla g_D(v_1) - \nabla g_D(v_2)\| &= \|E \left( \nabla F^*(r - v_1^\top E) - \nabla F^*(r - v_2^\top E) \right)\| \\
&\overset{(i)}{\leq} \|E\|_{\cdot,*} \|\nabla F^*(r - v_1^\top E) - \nabla F^*(r - v_2^\top E)\| \\
&\overset{(ii)}{\leq} \|E\|_{\cdot,*} \frac{1}{\beta} \|v_1^\top E - v_2^\top E\|_* \\
&\overset{(ii)}{\leq} \frac{\|E\|_{\cdot,*}^2}{\beta} \|v_1 - v_2\|_*
\end{aligned}
$$

The result follows.

$\square$

We can apply Lemma 2 to problem PrimalReg-$\boldsymbol{\lambda}$ and thus characterize the smoothness properties of the dual function $J_D$.

## B.2 Proof of Lemma 3

**Lemma 3.** *The dual function $J_D(\mathbf{v})$ is $(|\mathcal{S}| + 1)\eta$-smooth in the $\| \cdot \|_\infty$ norm.*

*Proof.* Recall that PrimalReg-$\boldsymbol{\lambda}$ can be written as RegLP:
$$
\max_{\boldsymbol{\lambda} \in \mathcal{D}} \langle \mathbf{r}, \boldsymbol{\lambda} \rangle - F(\boldsymbol{\lambda})
$$
$$
\text{s.t. } \mathbf{E}\boldsymbol{\lambda} = b.
$$

Where the regularizer $\left( F(\boldsymbol{\lambda}) := \frac{1}{\eta} \sum_{s,a} \boldsymbol{\lambda}_{s,a} \left( \log \left( \frac{\boldsymbol{\lambda}_{s,a}}{\mathbf{q}_{s,a}} \right) - 1 \right) \right)$ is $\frac{1}{\eta} - \| \cdot \|_1$ strongly convex. In this problem $\mathbf{r}$ corresponds to the reward vector, the vector $\mathbf{b} = (1 - \gamma)\boldsymbol{\mu} \in \mathbb{R}^{|\mathcal{S}|}$ and matrix $\mathbf{E} \in \mathbb{R}^{|\mathcal{S}| \times |\mathcal{S}| \times |\mathcal{A}|}$ takes the form:

$$
\mathbf{E}[s, s', a] = \begin{cases} \gamma \mathbf{P}_a(s|s') & \text{if } s \neq s' \\ 1 - \gamma \mathbf{P}_a(s|s) & \text{o.w.} \end{cases}
$$

Therefore

$$\|\mathbf{E}\|_{1,\infty} \leq S + 1$$

The result follows as a corollary of Lemma 1.

$\square$

# C  Proof of Lemma 4

The objective of this section is to show that a candidate dual variable $\widetilde{\mathbf{v}}$ having small gradient gives rise to a policy whose true visitation distribution has large primal value $J_P$.

**Lemma 4.** *Let $\widetilde{\mathbf{v}} \in \mathbb{R}^{|\mathcal{S}|}$ be arbitrary and let $\widetilde{\boldsymbol{\lambda}}$ be its corresponding candidate primal variable. If $\|\nabla_{\mathbf{v}} J_D(\widetilde{\mathbf{v}})\|_1 \leq \epsilon$ and Assumptions 2 and 3 hold then whenever $|\mathcal{S}| \geq 2$:*

$$
J_P(\boldsymbol{\lambda}^{\widetilde{\pi}}) \geq J_P(\boldsymbol{\lambda}_\eta^\star) - \epsilon \left( \frac{1+c}{1-\gamma} + \|\widetilde{\mathbf{v}}\|_\infty \right),
$$

*where $c = \frac{1 + \log(\frac{1}{\rho^3 \beta})}{\eta}$ and $\boldsymbol{\lambda}_\eta^\star$ is the $J_P$ optimum.*

*Proof.* For any $\boldsymbol{\lambda}$ and $\mathbf{v}$ let the lagrangian $J_L(\boldsymbol{\lambda}, \mathbf{v})$ be defined as,

$$J_L(\boldsymbol{\lambda}, \mathbf{v}) = (1-\gamma)\langle \boldsymbol{\mu}, \mathbf{v} \rangle + \left\langle \boldsymbol{\lambda}, \mathbf{A}^{\mathbf{v}} - \frac{1}{\eta}\left( \log\left(\frac{\boldsymbol{\lambda}}{\mathbf{q}}\right) - 1 \right) \right\rangle$$

Note that $J_D(\widetilde{\mathbf{v}}) = J_L(\widetilde{\boldsymbol{\lambda}}, \widetilde{\mathbf{v}})$ and that in fact $J_L$ is linear in $\bar{\mathbf{v}}$; *i.e.*,

$$J_L(\widetilde{\boldsymbol{\lambda}}, \bar{\mathbf{v}}) = J_L(\widetilde{\boldsymbol{\lambda}}, \widetilde{\mathbf{v}}) + \langle \nabla_{\mathbf{v}} J_L(\widetilde{\boldsymbol{\lambda}}, \widetilde{\mathbf{v}}), \bar{\mathbf{v}} - \widetilde{\mathbf{v}} \rangle.$$

Using Holder's inequality we have:

$$J_L(\widetilde{\boldsymbol{\lambda}}, \bar{\mathbf{v}}) \geq J_L(\widetilde{\boldsymbol{\lambda}}, \widetilde{\mathbf{v}}) - \|\nabla_{\mathbf{v}} J_L(\widetilde{\boldsymbol{\lambda}}, \widetilde{\mathbf{v}})\|_1 \cdot \|\bar{\mathbf{v}} - \widetilde{\mathbf{v}}\|_\infty = J_D(\widetilde{\mathbf{v}}) - \|\nabla_{\mathbf{v}} J_L(\widetilde{\boldsymbol{\lambda}}, \widetilde{\mathbf{v}})\|_1 \cdot \|\bar{\mathbf{v}} - \widetilde{\mathbf{v}}\|_\infty.$$

Let $\boldsymbol{\lambda}_\star$ be the candidate primal solution to the optimal dual solution $\mathbf{v}_\star = \arg\min_{\mathbf{v}} J_D(\mathbf{v})$. By weak duality we have that $J_D(\widetilde{\mathbf{v}}) \geq J_P(\boldsymbol{\lambda}^\star) = J_D(\mathbf{v}_\star)$, and since by assumption $\|\nabla_{\mathbf{v}} J_L(\widetilde{\boldsymbol{\lambda}}, \widetilde{\mathbf{v}})\|_1 \leq \epsilon$:

$$J_L(\widetilde{\boldsymbol{\lambda}}, \bar{\mathbf{v}}) \geq J_P(\boldsymbol{\lambda}^\star) - \epsilon\|\bar{\mathbf{v}} - \widetilde{\mathbf{v}}\|_\infty. \tag{9}$$

In order to use this inequality to lower bound the value of $J_P(\boldsymbol{\lambda}^{\widetilde{\pi}})$, we will need to choose an appropriate $\bar{\mathbf{v}}$ such that the LHS reduces to $J_P(\boldsymbol{\lambda}^{\widetilde{\pi}})$ while keeping the $\ell_\infty$ norm on the RHS small. Thus we consider setting $\bar{\mathbf{v}}$ as:

$$\bar{\mathbf{v}}_s = \mathbb{E}_{a,s' \sim \widetilde{\pi} \times \mathcal{T}}\left[ \mathbf{z}_s + \mathbf{r}_{s,a} - \frac{1}{\eta}\left( \log\left(\frac{\boldsymbol{\lambda}^{\widetilde{\pi}}_{s,a}}{\mathbf{q}_{s,a}}\right) - 1 \right) + \gamma\bar{\mathbf{v}}_{s'} \right]$$

Where $\mathbf{z} \in \mathbb{R}^{|S|}$ is some function to be determined later. It is clear that an appropriate $\mathbf{z}$ exists as long as $\mathbf{z}, \mathbf{r}, \frac{1}{\eta}\left( \log\left(\frac{\boldsymbol{\lambda}^{\widetilde{\pi}}_{s,a}}{\mathbf{q}_{s,a}}\right) - 1 \right)$ are uniformly bounded. Furthermore:

$$\|\bar{\mathbf{v}}\|_\infty \leq \frac{\max_{s,a}\left| \mathbf{z}_s + \mathbf{r}_{s,a} - \frac{1}{\eta}\left( \log\left(\frac{\boldsymbol{\lambda}^{\widetilde{\pi}}_{s,a}}{\mathbf{q}_{s,a}}\right) - 1 \right) \right|}{1-\gamma} \leq \frac{\|\mathbf{z}\|_\infty + \|\mathbf{r}\|_\infty + \frac{1}{\eta}\left\| \log\left(\frac{\boldsymbol{\lambda}^{\widetilde{\pi}}_{s,a}}{\mathbf{q}_{s,a}}\right) - 1 \right\|_\infty}{1-\gamma} \tag{10}$$

Notice that by Assumptions 2 and 3, we have that $\rho, \beta \leq \frac{1}{2}$. This is because for all $\pi$, Assumption 3 implies that:

$$0 \leq 2\rho \leq |\mathcal{S}|\rho \leq \sum_s \boldsymbol{\lambda}^\pi_s = 1$$

The proof for $\beta \leq \frac{1}{2}$ is symmetric. Due to Assumption 2 the $\|\cdot\|_\infty$ norm of $\log(\frac{\boldsymbol{\lambda}^{\widetilde{\pi}}}{\mathbf{q}}) - \mathbf{1}_{|\mathcal{S}||\mathcal{A}|}$ satisfies:

$$\left\| \log\left(\frac{\boldsymbol{\lambda}^{\widetilde{\pi}}}{\mathbf{q}}\right) - \mathbf{1}_{|\mathcal{S}||\mathcal{A}|} \right\|_\infty \leq 1 + \left\| \log\left(\frac{\boldsymbol{\lambda}^{\widetilde{\pi}}}{\mathbf{q}}\right) \right\|_\infty \leq 1 + \max(|\log(\rho/\beta)|, \log(1/\beta)) \leq 1 + \log(1/\rho) + \log(1/\beta).$$

Notice the following relationships hold:

$$\left\langle \widetilde{\boldsymbol{\lambda}}, \mathbf{A}^{\bar{\mathbf{v}}} - \frac{1}{\eta}\left( \log\left(\frac{\widetilde{\boldsymbol{\lambda}}}{\mathbf{q}}\right) - 1 \right) \right\rangle = \sum_s \widetilde{\boldsymbol{\lambda}}_s \left( \mathbb{E}_{a,s' \sim \widetilde{\pi} \times \mathbf{P}}\left[ \mathbf{r}_{s,a} + \gamma\bar{\mathbf{v}}_{s'} - \bar{\mathbf{v}}_s - \frac{1}{\eta}\left( \log\left(\frac{\widetilde{\boldsymbol{\lambda}}_{s,a}}{\mathbf{q}_{s,a}}\right) - 1 \right) \right] \right)$$

$$= \sum_s \widetilde{\boldsymbol{\lambda}}_s \left( \mathbb{E}_{a,s' \sim \widetilde{\pi} \times \mathbf{P}}\left[ \frac{1}{\eta}\left( \log\left(\frac{\boldsymbol{\lambda}^{\widetilde{\pi}}_{s,a}}{\mathbf{q}_{s,a}}\right) - 1 \right) - \frac{1}{\eta}\left( \log\left(\frac{\widetilde{\boldsymbol{\lambda}}_{s,a}}{\mathbf{q}_{s,a}}\right) - 1 \right) - \mathbf{z}_s \right] \right)$$

$$= \sum_s \widetilde{\boldsymbol{\lambda}}_s \left( \mathbb{E}_{a,s' \sim \widetilde{\pi} \times \mathbf{P}}\left[ \frac{1}{\eta}\log\left(\boldsymbol{\lambda}^{\widetilde{\pi}}_{s,a}\right) - \frac{1}{\eta}\log\left(\widetilde{\boldsymbol{\lambda}}_{s,a}\right) - \mathbf{z}_s \right] \right)$$

$$= \sum_s \widetilde{\boldsymbol{\lambda}}_s \left( \frac{1}{\eta}\log\left(\boldsymbol{\lambda}^{\widetilde{\pi}}_s\right) - \frac{1}{\eta}\log\left(\widetilde{\boldsymbol{\lambda}}_s\right) - \mathbf{z}_s \right) \tag{11}$$

Where $\widetilde{\boldsymbol{\lambda}}_s = \sum_a \widetilde{\boldsymbol{\lambda}}_{s,a}$ and $\boldsymbol{\lambda}_s^{\widetilde{\pi}} = \sum_a \boldsymbol{\lambda}_{s,a}^{\widetilde{\pi}}$. Note that by definition:

$$(1-\gamma)\langle \boldsymbol{\mu}, \bar{\mathbf{v}} \rangle = \left\langle \boldsymbol{\lambda}^{\widetilde{\pi}}, \mathbf{z} + \mathbf{r} - \frac{1}{\eta} \left( \log \left( \frac{\boldsymbol{\lambda}^{\widetilde{\pi}}}{\mathbf{q}} \right) - 1 \right) \right\rangle = J_P(\boldsymbol{\lambda}^{\widetilde{\pi}}) + \langle \boldsymbol{\lambda}^{\widetilde{\pi}}, \mathbf{z} \rangle. \qquad (12)$$

Let's expand the definition of $J_L(\widetilde{\boldsymbol{\lambda}}, \bar{\mathbf{v}})$ using Equations 11 and 12:

$$J_L(\widetilde{\boldsymbol{\lambda}}, \bar{\mathbf{v}}) = (1-\gamma)\langle \boldsymbol{\mu}, \bar{\mathbf{v}} \rangle + \left\langle \widetilde{\boldsymbol{\lambda}}, \mathbf{A}^{\bar{\mathbf{v}}} - \frac{1}{\eta} \left( \log \left( \frac{\widetilde{\boldsymbol{\lambda}}}{\mathbf{q}} \right) - 1 \right) \right\rangle$$

$$= J_P(\boldsymbol{\lambda}^{\widetilde{\pi}}) + \langle \boldsymbol{\lambda}^{\widetilde{\pi}}, \mathbf{z} \rangle + \sum_s \widetilde{\boldsymbol{\lambda}}_s \left( \frac{1}{\eta} \log \left( \boldsymbol{\lambda}_s^{\widetilde{\pi}} \right) - \frac{1}{\eta} \log \left( \widetilde{\boldsymbol{\lambda}}_s \right) - \mathbf{z}_s \right)$$

$$= J_P(\boldsymbol{\lambda}^{\widetilde{\pi}}) + \sum_s \left( \mathbf{z}_s (\boldsymbol{\lambda}_s^{\widetilde{\pi}} - \widetilde{\boldsymbol{\lambda}}_s) + \frac{1}{\eta} \widetilde{\boldsymbol{\lambda}}_s \log \left( \frac{\boldsymbol{\lambda}_s^{\widetilde{\pi}}}{\widetilde{\boldsymbol{\lambda}}_s} \right) \right)$$

Since we want this expression to equal $J_P(\boldsymbol{\lambda}^{\widetilde{\pi}})$, we need to choose $\mathbf{z}$ such that:

$$\mathbf{z}_s = \frac{\frac{1}{\eta} \log \left( \frac{\boldsymbol{\lambda}_s^{\widetilde{\pi}}}{\widetilde{\boldsymbol{\lambda}}_s} \right)}{1 - \frac{\boldsymbol{\lambda}_s^{\widetilde{\pi}}}{\widetilde{\boldsymbol{\lambda}}_s}}.$$

By Assumption 3 we have that for all $s$:

$$\frac{\boldsymbol{\lambda}_s^{\widetilde{\pi}}}{\widetilde{\boldsymbol{\lambda}}_s} \geq \rho$$

Now we bound $\|\mathbf{z}_s\|_\infty$. Note that the function $h(\phi) = \frac{\log \phi}{1-\phi}$ is non decreasing and negative, and therefore the maximum of its absolute value is achieved at the lower end of its domain. This implies:

$$|\mathbf{z}_s| \leq \frac{|h(\rho)|}{\eta} = \frac{|\log(\rho)|}{\eta(1-\rho)} \leq \frac{2\log(1/\rho)}{\eta}, \quad \forall s \in \mathcal{S}.$$

And therefore Equation 10 implies:

$$\|\bar{\mathbf{v}}\|_\infty \leq \frac{\frac{2\log(1/\rho)}{\eta} + 1 + \frac{1+\log(1/\rho)+\log(1/\beta)}{\eta}}{1-\gamma} = \frac{1 + \frac{1+\log(\frac{1}{\rho^3 \beta})}{\eta}}{1-\gamma}$$

Putting these together we obtain the following version of equation 9:

$$J_L(\widetilde{\boldsymbol{\lambda}}, \bar{\mathbf{v}}) \geq J_P(\boldsymbol{\lambda}^\star) - \epsilon \left( \frac{1 + \frac{1+\log(\frac{1}{\rho^3 \beta})}{\eta}}{1-\gamma} + \|\widetilde{\mathbf{v}}\|_\infty \right)$$

As desired. □

## D   Proof of Lemma 5

In this section we derive an upper bound for the $l_\infty$ norm of the optimal solution $\mathbf{v}^\star$.

**Lemma 5.** *Under Assumptions 1, 2 and 3, the optimal dual variables are bounded as*

$$\|\mathbf{v}^*\|_\infty \leq \frac{1}{1-\gamma} \left( 1 + \frac{\log \frac{|S||A|}{\beta \rho}}{\eta} \right) =: D. \qquad (7)$$

*Proof.* Recall the Lagrangian form,

$$\min_{\mathbf{v}} \max_{\boldsymbol{\lambda}_{s,a} \in \Delta_{S \times A}} J_L(\boldsymbol{\lambda}, \mathbf{v}) := (1-\gamma)\langle \mathbf{v}, \boldsymbol{\mu} \rangle + \left\langle \boldsymbol{\lambda}, \mathbf{A}^{\mathbf{v}} - \frac{1}{\eta} \left( \log \left( \frac{\boldsymbol{\lambda}_{s,a}}{\mathbf{q}_{s,a}} \right) - 1 \right) \right\rangle.$$

The KKT conditions of $\boldsymbol{\lambda}^*, \mathbf{v}^*$ imply that for any $s, a$, either (1) $\boldsymbol{\lambda}_{s,a}^* = 0$ and $\frac{\partial}{\partial \boldsymbol{\lambda}_{s,a}} J_L(\boldsymbol{\lambda}^*, v^*) \leq 0$ or (2) $\frac{\partial}{\partial \boldsymbol{\lambda}_{s,a}} J_L(\boldsymbol{\lambda}^*, \mathbf{v}^*) = 0$. The partial derivative of $J_L$ is given by,

$$\frac{\partial}{\partial \boldsymbol{\lambda}_{s,a}} J_L(\boldsymbol{\lambda}^*, \mathbf{v}^*) = \mathbf{r}_{s,a} - \frac{1}{\eta} \log \left( \frac{\boldsymbol{\lambda}_{s,a}^*}{\mathbf{q}_{s,a}} \right) + \gamma \sum_{s'} P_a(s'|s) \mathbf{v}_{s'}^* - \mathbf{v}_s^*. \tag{13}$$

Thus, for any $s, a$, either

$$\boldsymbol{\lambda}_{s,a}^* = 0 \text{ and } \mathbf{v}_s^* \geq \mathbf{r}_{s,a} - \frac{1}{\eta} \log \left( \frac{\boldsymbol{\lambda}_{s,a}^*}{\mathbf{q}_{s,a}} \right) + \gamma \sum_{s'} P_a(s'|s) \mathbf{v}_{s'}^*, \tag{14}$$

or,

$$\boldsymbol{\lambda}_{s,a}^* > 0 \text{ and } \mathbf{v}_s^* = \mathbf{r}_{s,a} - \frac{1}{\eta} \log \left( \frac{\boldsymbol{\lambda}_{s,a}^*}{\mathbf{q}_{s,a}} \right) + \gamma \sum_{s'} P_a(s'|s) \mathbf{v}_{s'}^*. \tag{15}$$

Recall that $\boldsymbol{\lambda}^*$ is the discounted state-action visitations of some policy $\pi_\star$; *i.e.*, $\boldsymbol{\lambda}_{s,a}^* = \boldsymbol{\lambda}_s^{\pi_\star} \cdot \pi_\star(a|s)$ for some $\pi_\star$. Note that by Assumption 3, any policy $\pi$ has $\boldsymbol{\lambda}_s^{\pi_\star} > 0$ for all $s$. Accordingly, the KKT conditions imply,

$$\pi_\star(a|s) = 0 \text{ and } \mathbf{v}_s^* \geq \mathbf{r}_{s,a} - \frac{1}{\eta} \log \left( \frac{\boldsymbol{\lambda}_{s,a}^*}{\mathbf{q}_{s,a}} \right) + \gamma \sum_{s'} P_a(s'|s) \mathbf{v}_{s'}^*, \tag{16}$$

or,

$$\pi_\star(a|s) > 0 \text{ and } \mathbf{v}_s^* = \mathbf{r}_{s,a} - \frac{1}{\eta} \log \left( \frac{\boldsymbol{\lambda}_{s,a}^*}{\mathbf{q}_{s,a}} \right) + \gamma \sum_{s'} P_a(s'|s) \mathbf{v}_{s'}^*. \tag{17}$$

Equivalently,

$$\mathbf{v}_s^* = \mathbb{E}_{a \sim \pi_\star(s)} \left[ \mathbf{r}_{s,a} - \frac{1}{\eta} \log \left( \frac{\boldsymbol{\lambda}_{s,a}^*}{\mathbf{q}_{s,a}} \right) + \gamma \sum_{s'} P_a(s'|s) \mathbf{v}_{s'}^* \right] \tag{18}$$

$$= \frac{1}{\eta} \mathbb{E}_{a \sim \pi_\star(s)} \left[ -\log \left( \frac{\pi(a|s)}{\mathbf{q}_{a|s}} \right) \right] + \mathbb{E}_{a \sim \pi(s)} \left[ r_{s,a} - \frac{1}{\eta} \log \left( \frac{\boldsymbol{\lambda}_s^{\pi_\star}}{\mathbf{q}_s} \right) + \gamma \sum_{s'} P_a(s'|s) \mathbf{v}_{s'}^* \right]. \tag{19}$$

We may express these conditions as a Bellman recurrence for $\mathbf{v}_s^*$:

$$\mathbf{v}_s^* = \frac{1}{\eta} \mathbb{E}_{a \sim \pi_\star(s)} \left[ -\log \left( \frac{\pi(a|s)}{\mathbf{q}_{a|s}} \right) \right] + \mathbb{E}_{a \sim \pi_\star(s)} \left[ \mathbf{r}_{s,a} - \frac{1}{\eta} \log \left( \frac{\boldsymbol{\lambda}_s^{\pi_\star}}{\mathbf{q}_s} \right) + \gamma \sum_{s'} P_a(s'|s) \mathbf{v}_{s'}^* \right]. \tag{20}$$

The solution to these Bellman equations is bounded when $\mathbb{E}_{a \sim \pi_\star(s)} \left[ -\log \left( \frac{\pi_\star(a|s)}{\mathbf{q}_{a|s}} \right) \right]$, $\mathbf{r}_{s,a}$, and $\log \left( \frac{\boldsymbol{\lambda}_s^{\pi}}{\mathbf{q}_s} \right)$ are bounded [24]. And indeed, by Assumptions 3 and 1, each of these is bounded by within $[\log \beta, \log |A|]$, $[0, 1]$, and $[\log \rho, -\log \beta]$, respectively. We may thus bound the solution as,

$$\|\mathbf{v}^*\|_\infty \leq \frac{1}{1 - \gamma} \left( 1 + \frac{\log \frac{|S||A|}{\beta \rho}}{\eta} \right). \tag{21}$$

$\square$

# E  Convergence rates for REPS

We start with the proof of Lemma 6 which we restate for convenience:

**Lemma 11.** *If* $\mathbf{x}$ *is an* $\epsilon-$*optimal solution for the* $\alpha-$*smooth function* $h : \mathbb{R}^d \to \mathbb{R}$ *w.r.t. norm* $\|\cdot\|_\star$ *then the gradient of* $h$ *at* $\mathbf{x}$ *satisfies:*

$$\|\nabla h(\mathbf{x})\| \leq \sqrt{2\alpha\epsilon}.$$

*Proof.* Let $\mathbf{x} \in \mathbb{R}^d$ be an arbitrary point and let $\mathbf{x}'$ equal the point resulting of the update

$$\mathbf{x}' = \arg\min_{\mathbf{y} \in \mathcal{D}} \frac{1}{\alpha} \langle \nabla h(\mathbf{x}), \mathbf{y} - \mathbf{x} \rangle + \frac{\|\mathbf{y} - \mathbf{x}\|_\star^2}{2} \tag{22}$$

Notice that by smoothness of $h$:

$$h(\mathbf{x}') \leq h(\mathbf{x}) + \langle \nabla h(\mathbf{x}), \mathbf{x}' - \mathbf{x} \rangle + \frac{\alpha}{2} \|\mathbf{x}' - \mathbf{x}\|_\star^2 = h(\mathbf{x}) - \frac{1}{2\alpha} \|\nabla h(\mathbf{x})\|^2 \tag{23}$$

Since $h(\mathbf{x}^\star) \leq h(\mathbf{x}')$ and $\mathbf{x}$ is $\epsilon-$optimal:

$$\frac{1}{2\alpha} \|\nabla h(\mathbf{x})\|^2 + h(\mathbf{x}^\star) \overset{(i)}{\leq} \frac{1}{2\alpha} \|\nabla h(\mathbf{x})\|^2 + h(\mathbf{x}') \overset{(ii)}{\leq} h(\mathbf{x}) \overset{(iii)}{\leq} h(\mathbf{x}^\star) + \epsilon$$

Inequality $(i)$ holds because $h(\mathbf{x}^\star) \leq h(\mathbf{x}')$, inequality $(ii)$ by Equation 23 and $(iii)$ by $\epsilon-$optimality of $\mathbf{x}$. Therefore:

$$\frac{1}{2\alpha} \|\nabla h(\mathbf{x})\|^2 \leq \epsilon.$$

The result follows. $\qquad\square$

We also show that the gradient norm of a smooth function over a bounded domain containing the optimum can be bounded:

**Lemma 12.** *If $h$ is an $\alpha-$smooth function w.r.t. norm $\|\cdot\|_\star$, and $\mathbf{x}^\star$ is such that $\nabla h(\mathbf{x}^\star) = \mathbf{0}$ then:*

$$\|\nabla h(\mathbf{x})\| \leq \alpha \|\mathbf{x} - \mathbf{x}^\star\|_\star.$$

*And therefore whenever $\|\mathbf{x} - \mathbf{x}^\star\|_\star \leq D$ we have that:*

$$\|\nabla h(\mathbf{x})\| \leq \alpha D.$$

*Proof.* Since $h$ is $\alpha-$smooth:

$$h(\mathbf{x}) \leq h(\mathbf{x}^\star) + \langle \nabla h(\mathbf{x}^\star), \mathbf{x} - \mathbf{x}^\star \rangle + \frac{\alpha}{2} \|\mathbf{x} - \mathbf{x}^\star\|_\star^2 = h(\mathbf{x}^\star) + \frac{\alpha}{2} \|\mathbf{x} - \mathbf{x}^\star\|_\star^2$$

Therefore:

$$h(\mathbf{x}) - h(\mathbf{x}^\star) \leq \frac{\alpha}{2} \|\mathbf{x} - \mathbf{x}^\star\|_\star^2.$$

Therefore, as a consequence of Lemma 6:

$$\|\nabla h(\mathbf{x})\| \leq \alpha D.$$

The result follows.

$\qquad\square$

### E.1 Proof of Theorem 2

We can now prove the estimation guarantees whenever exact gradients are available.

**Theorem 4.** *For any $\epsilon > 0$, let $\eta = \frac{1}{2\epsilon \log(\frac{|\mathcal{S}||\mathcal{A}|}{\beta})}$. If $T \geq (|\mathcal{S}| + 1)^{3/2} \frac{(2+c'')^2}{(1-\gamma)^2 \epsilon^2}$, then $\pi_T$ is an $\epsilon-$optimal policy.*

*Proof.* As a consequence of Corollary 2, we can conclude that:

$$J_P(\boldsymbol{\lambda}^{\pi_T}) \geq J_P(\boldsymbol{\lambda}^{\star, \eta}) - \frac{\epsilon}{2}.$$

Where $\boldsymbol{\lambda}_\eta^\star$ is the regularized optimum. Recall that:

$$J_P(\boldsymbol{\lambda}) = \sum_{s,a} \boldsymbol{\lambda}_{s,a} \mathbf{r}_{s,a} - \frac{1}{\eta} \sum_{s,a} \boldsymbol{\lambda}_{s,a} \left( \log\left( \frac{\boldsymbol{\lambda}_{s,a}}{\mathbf{q}_{s,a}} \right) - 1 \right).$$

Since $\boldsymbol{\lambda}^{\star,\eta}$ is the maximizer of the regularized objective, it satisfies $J_P(\boldsymbol{\lambda}^{\star,\eta}) \geq J_P(\boldsymbol{\lambda}^*)$ where $\boldsymbol{\lambda}^\star$ is the visitation frequency of the optimal policy corresponding to the unregularized objective. We can conclude that:

$$\sum_{s,a} \boldsymbol{\lambda}_{s,a}^{\pi_T} \mathbf{r}_{s,a} \geq \sum_{s,a} \boldsymbol{\lambda}_{s,a}^{\star} \mathbf{r}_{s,a} + \frac{1}{\eta}\left(\sum_{s,a} \boldsymbol{\lambda}_{s,a}^{\pi_T}\left(\log\left(\frac{\boldsymbol{\lambda}_{s,a}^{\pi_T}}{\mathbf{q}_{s,a}}\right) - 1\right) - \sum_{s,a} \boldsymbol{\lambda}_{s,a}^{\star}\left(\log\left(\frac{\boldsymbol{\lambda}_{s,a}^{\star}}{\mathbf{q}_{s,a}}\right) - 1\right)\right) - \frac{\epsilon}{2}$$

$$= \sum_{s,a} \boldsymbol{\lambda}_{s,a}^{\star} \mathbf{r}_{s,a} + \frac{1}{\eta}\left(\sum_{s,a} \boldsymbol{\lambda}_{s,a}^{\pi_T}\left(\log\left(\frac{\boldsymbol{\lambda}_{s,a}^{\pi_T}}{\mathbf{q}_{s,a}}\right)\right) - \sum_{s,a} \boldsymbol{\lambda}_{s,a}^{\star}\left(\log\left(\frac{\boldsymbol{\lambda}_{s,a}^{\star}}{\mathbf{q}_{s,a}}\right)\right)\right) - \frac{\epsilon}{2}$$

$$\geq \sum_{s,a} \boldsymbol{\lambda}_{s,a}^{\star} \mathbf{r}_{s,a} - \frac{2}{\eta}\log(\frac{|\mathcal{S}||\mathcal{A}|}{\beta}) - \frac{\epsilon}{2}$$

And therefore if $\eta = \frac{1}{4\epsilon \log(\frac{|\mathcal{S}||\mathcal{A}|}{\beta})}$, we can conclude that:

$$\sum_{s,a} \boldsymbol{\lambda}_{s,a}^{\pi_T} \mathbf{r}_{s,a} \geq \sum_{s,a} \boldsymbol{\lambda}_{s,a}^{\star} \mathbf{r}_{s,a} - \epsilon.$$

$\square$

# F   Accelerated Gradient Descent

---
**Algorithm 4** Accelerated Gradient Descent

---
**Input** Initial point $\mathbf{x}_0$, domain $\mathcal{D}$, distance generating function $w$.
$\mathbf{y}_0 \leftarrow \mathbf{x}_0, \quad \mathbf{z}_0 \leftarrow \mathbf{x}_0.$
**for** $t = 0, \cdots, T$ **do**
$\quad \eta_{t+1} = \frac{t+2}{2\alpha}$ and $\tau_t = \frac{2}{t+2}.$

$$\mathbf{x}_{t+1} \leftarrow (1 - \tau_t)\mathbf{y}_t + \tau_t \mathbf{z}_t$$

$$\mathbf{y}_{t+1} \leftarrow \arg\min_{\mathbf{y} \in \mathcal{D}} \frac{1}{\alpha}\langle \nabla h(\mathbf{x}_t), \mathbf{y} - \mathbf{x}_t\rangle + \frac{\|\mathbf{y} - \mathbf{x}_t\|_\star^2}{2}.$$

$$z_{t+1} \leftarrow \arg\min_{\mathbf{z} \in \mathcal{D}} \eta_t\langle \nabla h(\mathbf{x}_t), \mathbf{z} - \mathbf{z}_t\rangle + D_w(\mathbf{z}_t, \mathbf{z}).$$

**end**
For some stepsize parameter sequence $\eta_t$.

---

Algorithm 4 satisfies the following convergence guarantee:

# G   Stochastic Gradient Descent

In this section we will have all the proofs and results corresponding to Section 6 in the main. We start by showing the proof of Lemma 9.

**Lemma 9.** *Let $f : \mathbb{R}^d \to \mathbb{R}$ be an $L-$smooth function. We consider the following update:*

$$\mathbf{x}_{t+1}' = \mathbf{x}_t - \tau\left(\nabla f(\mathbf{x}_t) + \boldsymbol{\epsilon}_t + \mathbf{b}_t\right); \quad \mathbf{x}_{t+1} = \Pi_{\mathcal{D}}(\mathbf{x}_{t+1}').$$

*If $\tau \leq \frac{2}{L}$ then:*

$$f(\mathbf{x}_{t+1}) - f(\mathbf{x}_\star) \leq \frac{\|\mathbf{x}_t - \mathbf{x}_\star\|^2 - \|\mathbf{x}_{t+1} - \mathbf{x}_\star\|^2}{2\tau} + 2\tau\|\nabla f(\mathbf{x}_t)\|^2 + 5\tau\|\mathbf{b}_t\|^2 + 5\tau\|\boldsymbol{\epsilon}_t\|^2$$
$$+ \|\mathbf{b}_t\|_1\|\mathbf{x}_t - \mathbf{x}_\star\|_\infty - \langle\boldsymbol{\epsilon}_t, \mathbf{x}_t - \mathbf{x}_\star\rangle.$$

*Proof.* Through the proof we use the notation $\|\cdot\|$ to denote the $L_2$ norm. By smoothness the following holds:

$$f(\mathbf{x}_{t+1}) \leq f(\mathbf{x}_t) + \langle \nabla f(\mathbf{x}_t), \mathbf{x}_{t+1} - \mathbf{x}_t \rangle + \frac{L}{2}\|\mathbf{x}_{t+1} - \mathbf{x}_t\|_\infty^2 \leq f(\mathbf{x}_t) + \langle \nabla f(\mathbf{x}_t), \mathbf{x}_{t+1} - \mathbf{x}_t \rangle + \frac{L}{2}\|\mathbf{x}_{t+1} - \mathbf{x}_t\|^2$$

Since $\mathbf{x}_{t+1} = \Pi_{\mathcal{D}}(\mathbf{x}'_{t+1})$ and by properties of a convex projection:

$$\langle \mathbf{x}'_{t+1} - \mathbf{x}_{t+1}, \mathbf{x}_t - \mathbf{x}_{t+1} \rangle \leq 0.$$

And therefore:

$$\langle \mathbf{x}_t - \tau\left(\nabla f(\mathbf{x}_t) + \mathbf{b}_t + \boldsymbol{\epsilon}_t\right) - \mathbf{x}_{t+1}, \mathbf{x}_t - \mathbf{x}_{t+1} \rangle \leq 0.$$

Which in turn implies that :

$$\|\mathbf{x}_t - \mathbf{x}_{t+1}\|^2 \leq \tau \langle \nabla f(\mathbf{x}_t) + \mathbf{b}_t + \boldsymbol{\epsilon}_t, \mathbf{x}_t - \mathbf{x}_{t+1} \rangle.$$

We can conclude that:

$$f(\mathbf{x}_{t+1}) \leq f(\mathbf{x}_t) - \frac{\|\mathbf{x}_t - \mathbf{x}_{t+1}\|^2}{\tau} + \langle \mathbf{b}_t + \boldsymbol{\epsilon}_t, \mathbf{x}_t - \mathbf{x}_{t+1} \rangle + \frac{L}{2}\|\mathbf{x}_{t+1} - \mathbf{x}_t\|^2. \qquad (24)$$

By convexity:

$$f(\mathbf{x}_\star) \geq f(\mathbf{x}_t) + \langle \nabla f(\mathbf{x}_t), \mathbf{x}_\star - \mathbf{x}_t \rangle.$$

And therefore $f(\mathbf{x}_t) \leq f(\mathbf{x}_\star) + \langle \nabla f(\mathbf{x}_t), \mathbf{x}_t - \mathbf{x}_\star \rangle$.

Combining this last result with Equation 24:

$$f(\mathbf{x}_{t+1}) \leq f(\mathbf{x}_\star) + \langle \nabla f(\mathbf{x}_t), \mathbf{x}_t - \mathbf{x}_\star \rangle + \left(\frac{L}{2} - \frac{1}{\tau}\right)\|\mathbf{x}_{t+1} - \mathbf{x}_t\|^2 + \langle \mathbf{b}_t + \boldsymbol{\epsilon}_t, \mathbf{x}_t - \mathbf{x}_{t+1} \rangle. \quad (25)$$

Now observe that as a consequence of the contraction property of projections

$$\begin{aligned}\|\mathbf{x}_{t+1} - \mathbf{x}_\star\|^2 &\leq \|\mathbf{x}_t - \tau\left(\nabla f(\mathbf{x}_t) + \mathbf{b}_t + \boldsymbol{\epsilon}_t\right) - \mathbf{x}_\star\|^2 \\ &= \|\mathbf{x}_t - \mathbf{x}_\star\|^2 + \tau^2\|\nabla f(\mathbf{x}_t) + \mathbf{b}_t + \boldsymbol{\epsilon}_t\|^2 - 2\tau\langle \nabla f(\mathbf{x}_t) + \mathbf{b}_t + \boldsymbol{\epsilon}_t, \mathbf{x}_t - \mathbf{x}_\star \rangle.\end{aligned}$$

And therefore:

$$\langle \nabla f(\mathbf{x}_t), \mathbf{x}_t - \mathbf{x}_\star \rangle \leq \frac{\|\mathbf{x}_t - \mathbf{x}_\star\|^2 - \|\mathbf{x}_{t+1} - \mathbf{x}_\star\|^2}{2\tau} + \frac{\tau}{2}\|\nabla f(\mathbf{x}_t) + \mathbf{b}_t + \boldsymbol{\epsilon}_t\|^2 - \langle \mathbf{b}_t + \boldsymbol{\epsilon}_t, \mathbf{x}_t - \mathbf{x}_\star \rangle.$$

Substituting this last inequality into Equation 25:

$$f(\mathbf{x}_{t+1}) - f(\mathbf{x}_\star) \leq \frac{\|\mathbf{x}_t - \mathbf{x}_\star\|^2 - \|\mathbf{x}_{t+1} - \mathbf{x}_\star\|^2}{2\tau} + \frac{\tau}{2}\|\nabla f(\mathbf{x}_t) + \mathbf{b}_t + \boldsymbol{\epsilon}_t\|^2 - \langle \mathbf{b}_t + \boldsymbol{\epsilon}_t, \mathbf{x}_t - \mathbf{x}_\star \rangle +$$

$$\qquad (26)$$

$$\left(\frac{L}{2} - \frac{1}{\tau}\right)\|\mathbf{x}_{t+1} - \mathbf{x}_t\|^2 + \langle \mathbf{b}_t + \boldsymbol{\epsilon}_t, \mathbf{x}_t - \mathbf{x}_{t+1} \rangle \qquad (27)$$

Notice that as a consequence of the contraction property of projections:

$$\|\mathbf{x}_{t+1} - \mathbf{x}_t\|^2 \le \|\mathbf{x}_t - \tau\left(\nabla f(\mathbf{x}_t) + \mathbf{b}_t + \boldsymbol{\epsilon}_t\right) - \mathbf{x}_t\|$$
$$= \tau\|\nabla f(\mathbf{x}_t) + \mathbf{b}_t + \boldsymbol{\epsilon}_t\|$$

And therefore

$$\langle \mathbf{b}_t + \boldsymbol{\epsilon}_t, \mathbf{x}_t - \mathbf{x}_{t+1}\rangle \le \|\mathbf{b}_t + \boldsymbol{\epsilon}_t\|\|\mathbf{x}_t - \mathbf{x}_{t+1}\| \le \tau\|\mathbf{b}_t + \boldsymbol{\epsilon}_t\|\|\nabla f(\mathbf{x}_t) + \mathbf{b}_t + \boldsymbol{\epsilon}_t\|$$

:

Substituting this back into 27 and assuming $\frac{L}{2} \le \frac{1}{\tau}$:

$$
\begin{aligned}
f(\mathbf{x}_{t+1}) - f(\mathbf{x}_\star) &\le \frac{\|\mathbf{x}_t - \mathbf{x}_\star\|^2 - \|\mathbf{x}_{t+1} - \mathbf{x}_\star\|^2}{2\tau} + \frac{\tau}{2}\|\nabla f(\mathbf{x}_t) + \mathbf{b}_t + \boldsymbol{\epsilon}_t\|^2 - \langle \mathbf{b}_t + \boldsymbol{\epsilon}_t, \mathbf{x}_t - \mathbf{x}_\star\rangle + \\
&\quad \tau\|\mathbf{b}_t + \boldsymbol{\epsilon}_t\|\|\nabla f(\mathbf{x}_t) + \mathbf{b}_t + \boldsymbol{\epsilon}_t\| \\
&\le \frac{\|\mathbf{x}_t - \mathbf{x}_\star\|^2 - \|\mathbf{x}_{t+1} - \mathbf{x}_\star\|^2}{2\tau} + \tau\|\nabla f(\mathbf{x}_t) + \mathbf{b}_t + \boldsymbol{\epsilon}_t\|^2 + \frac{\tau}{2}\|\mathbf{b}_t + \boldsymbol{\epsilon}_t\|^2 - \langle \mathbf{b}_t + \boldsymbol{\epsilon}_t, \mathbf{x}_t - \mathbf{x}_\star\rangle \\
&\overset{(i)}{\le} \frac{\|\mathbf{x}_t - \mathbf{x}_\star\|^2 - \|\mathbf{x}_{t+1} - \mathbf{x}_\star\|^2}{2\tau} + 2\tau\|\nabla f(\mathbf{x}_t)\|^2 + 5\tau\|\mathbf{b}_t\|^2 + 5\tau\|\boldsymbol{\epsilon}_t\|^2 - \langle \mathbf{b}_t + \boldsymbol{\epsilon}_t, \mathbf{x}_t - \mathbf{x}_\star\rangle \\
&\le \frac{\|\mathbf{x}_t - \mathbf{x}_\star\|^2 - \|\mathbf{x}_{t+1} - \mathbf{x}_\star\|^2}{2\tau} + 2\tau\|\nabla f(\mathbf{x}_t)\|^2 + 5\tau\|\mathbf{b}_t\|^2 + 5\tau\|\boldsymbol{\epsilon}_t\|^2 + \|\mathbf{b}_t\|_1\|\mathbf{x}_t - \mathbf{x}_\star\|_\infty - \langle \boldsymbol{\epsilon}_t, \mathbf{x}_t - \mathbf{x}
\end{aligned}
$$

Inequality $(i)$ is a result of a repeated use of Young's inequality. The last inequality is a result of Cauchy-Schwartz.

$\square$

## H  Stochastic Gradients Analysis

We will make use of the following concentration inequality:

**Lemma 13** (Uniform empirical Bernstein bound). *In the terminology of Howard et al. [16], let $S_t = \sum_{i=1}^t Y_i$ be a sub-$\psi_P$ process with parameter $c > 0$ and variance process $W_t$. Then with probability at least $1 - \delta$ for all $t \in \mathbb{N}$*

$$
\begin{aligned}
S_t &\le 1.44\sqrt{(W_t \vee m)\left(1.4\ln\ln\left(2\left(\frac{W_t}{m} \vee 1\right)\right) + \ln\frac{5.2}{\delta}\right)} \\
&\quad + 0.41c\left(1.4\ln\ln\left(2\left(\frac{W_t}{m} \vee 1\right)\right) + \ln\frac{5.2}{\delta}\right)
\end{aligned}
$$

*where $m > 0$ is arbitrary but fixed.*

*Proof.* Setting $s = 1.4$ and $\eta = 2$ in the polynomial stitched boundary in Equation (10) of Howard et al. [16] shows that $u_{c,\delta}(v)$ is a sub-$\psi_G$ boundary for constant $c$ and level $\delta$ where

$$
\begin{aligned}
u_{c,\delta}(v) &= 1.44\sqrt{(v \vee 1)\left(1.4\ln\ln\left(2(v \vee 1)\right) + \ln\frac{5.2}{\delta}\right)} \\
&\quad + 1.21c\left(1.4\ln\ln\left(2(v \vee 1)\right) + \ln\frac{5.2}{\delta}\right).
\end{aligned}
$$

By the boundary conversions in Table 1 in Howard et al. [16] $u_{c/3,\delta}$ is also a sub-$\psi_P$ boundary for constant $c$ and level $\delta$. The desired bound then follows from Theorem 1 by Howard et al. [16]. $\square$

The following estimation bound holds:

**Lemma 14.** *Let $\{(s_\ell, a_\ell, s'_\ell)\}_{\ell=1}^\infty$ be samples generated as above. Let $N_t(s,a) = \sum_{\ell=1}^t \mathbf{1}(s_\ell, a_\ell = s, a)$. Let $\delta \in (0,1)$. With probability at least $1 - (2|\mathcal{S}||\mathcal{A}|\delta)$ for all $t$ such that $\ln(2t) + \ln\frac{5.2}{\delta} \le \frac{t\beta}{6}$ and for all $s, a \in \mathcal{S} \times \mathcal{A}$ simultaneously:*

$$N_t(s,a) \in \left[ \frac{t\mathbf{q}_{s,a}}{4}, \frac{7t\mathbf{q}_{s,a}}{4} \right]$$

*Additionally define $\widehat{\mathbf{q}}_{s,a} = \frac{N_t(s,a)}{t}$. For any $\epsilon \in (0,1)$ with probability at least $1 - (2|\mathcal{S}||\mathcal{A}|\delta)$ and for all $t$ such that $\frac{t}{\ln\ln(2t)} \ge \frac{1+\ln\frac{5.2}{\delta}}{\beta\epsilon^2}$:*

$$|\widehat{\mathbf{q}}_{s,a} - \mathbf{q}_{s,a}| \le 3.69\epsilon\mathbf{q}_{s,a}.$$

*Proof.* We start by producing a lower bound for $N_t(s,a)$. Consider the martingale sequence $Z_{s,a}(\ell) = \mathbf{1}(s_\ell = s, a_\ell = a) - \mathbf{q}_{s,a}$ with the variance process $V_t = \sum_{\ell=1}^t \mathbb{E}\left[Z_{s,a}^2(\ell)|\mathcal{F}_{\ell-1}\right]$ satisfying $\mathbb{E}[Z_{s,a}^2(\ell)|\mathcal{F}_{\ell-1}] \le \mathbf{q}_{s,a}$. The martingale process $Z_{s,a}(\ell)$ satisfies the sub-$\psi_P$ condition of [16] with constant $c = 1$ (see Bennet case in Table 3 of [16]). By Lemma 13, and setting $m = \mathbf{q}_{s,a}$ we conclude that with probability at least $1 - \delta$ for all $t \in \mathbb{N}$ :

$$N_t(s,a) \ge t\mathbf{q}_{s,a} - 1.44\sqrt{\mathbf{q}_{s,a}t\left(\ln\ln(2t) + \ln\frac{5.2}{\delta}\right)} - 0.41\left(1.4\ln\ln(2t) + \ln\frac{5.2}{\delta}\right) \quad (28)$$

$$\overset{(i)}{\ge} t\mathbf{q}_{s,a} - \frac{t\mathbf{q}_{s,a}}{2} - \frac{3}{2}\left(\ln\ln(2t) + \ln\frac{5.2}{\delta}\right)$$

$$= \frac{t\mathbf{q}_{s,a}}{2} - \frac{3}{2}\left(\ln\ln(2t) + \ln\frac{5.2}{\delta}\right)$$

Inequality $(i)$ holds because $\sqrt{\mathbf{q}_{s,a}t\left(\ln\ln(2t) + \ln\frac{5.2}{\delta}\right)} \le \frac{\mathbf{q}_{s,a}t}{2} + \frac{\ln\ln(2t)+\ln\frac{5.2}{\delta}}{2}$. As a consequence of Assumption 2 we can infer that with probability at least $1 - \delta$ for all $t$ such that $\ln\ln(2t) + \ln\frac{5.2}{\delta} \le \frac{t\beta}{6} \le \frac{t\mathbf{q}_{s,a}}{6}$ :

$$N_t(s,a) \ge \frac{t\mathbf{q}_{s,a}}{4}$$

The same sequence of inequalities but inverted implies the upper bound result. The last result is a simple consequence of the union bound. To obtain the stronger bound we start by noting that since $\frac{t}{\ln\ln(2t)} \ge \frac{1+\ln\frac{5.2}{\delta}}{\beta\epsilon^2} \ge \frac{1+\ln\frac{5.2}{\delta}}{\mathbf{q}_{s,a}\epsilon^2}$ for all $(s,a)$ we can transform Equation 28 as:

$$N_t(s,a) \ge t\mathbf{q}_{s,a} - 1.44\sqrt{\mathbf{q}_{s,a}t\left(\ln\ln(2t) + \ln\frac{5.2}{\delta}\right)} - 0.41\left(1.4\ln\ln(2t) + \ln\frac{5.2}{\delta}\right)$$

$$\ge t\mathbf{q}_{s,a} - 2.88\sqrt{\mathbf{q}_{s,a}t\ln\ln(2t)(1 + \ln\frac{5.2}{\delta})} - 0.81\ln\ln(2t)(1 + \ln\frac{5.2}{\delta})$$

$$\ge t\mathbf{q}_{s,a} - 3.69\sqrt{\mathbf{q}_{s,a}t\ln\ln(2t)(1 + \ln\frac{5.2}{\delta})}$$

$$\ge t\mathbf{q}_{s,a} - 3.69\mathbf{q}_{s,a}\epsilon$$

The same sequence of inequalities but inverted implie the upper bound. This finishes the proof.

$\square$

The gradients of $J_D(\mathbf{v})$ can be written as:

$$(\nabla_\mathbf{v} J_D(\mathbf{v}))_s = (1-\gamma)\boldsymbol{\mu}_s + \gamma \sum_{s',a} \frac{\exp\left(\eta\mathbf{A}_{s',a}^\mathbf{v}\right)\mathbf{q}_{s',a}}{\mathbf{Z}} P_a(s|s') -$$

$$\sum_a \frac{\exp\left(\eta\mathbf{A}_{s,a}^\mathbf{v}\right)\mathbf{q}_{s,a}}{\mathbf{Z}},$$

Where $Z = \sum_{s,a} \exp\left(\eta \mathbf{A}_{s,a}^{\mathbf{v}}\right) \mathbf{q}_{s,a}$. We will work under the assumption that $\mathbf{q}_{s,a} \propto \exp(\eta \mathbf{A}_{s,a}^{\mathbf{v}'})$ for some value vector $\mathbf{v}'$. Given a value vector $\mathbf{v}$ we denote its induced policy $\pi^{\mathbf{v}}$ as:

$$\pi^{\mathbf{v}}(a|s) = \frac{\exp\left(\eta \mathbf{A}_{s,a}^{\mathbf{v}}\right) \mathbf{q}_{s,a}}{\mathbf{Z}_s}$$

Where $\mathbf{Z}_s = \sum_a \exp\left(\eta \mathbf{A}_{s,a}^{\mathbf{v}}\right) \mathbf{q}_{s,a}$. If we define $\mathbf{q}_s = \sum_a \mathbf{q}_{s,a}$, and we define $\mathbf{q}_{a|s} = \frac{\mathbf{q}_{s,a}}{\mathbf{q}_s}$ then we can write:

$$\pi^{\mathbf{v}}(a|s) = \frac{\exp\left(\eta \mathbf{A}_{s,a}^{\mathbf{v}}\right) \mathbf{q}_{a|s}}{\mathbf{Z}_{a|s}}$$

Where $\mathbf{Z}_s = \sum_a \exp\left(\eta \mathbf{A}_{s,a}^{\mathbf{v}}\right) \mathbf{q}_{a|s}$. We work under the assumption that $\mathbf{q}_{a|s}$ is a policy, and therefore known to the learner. We start by showing how to maintain a good estimator $\widehat{\mathbf{A}}_{s,a}^{\mathbf{v}}$ using stochastic gradient descent over a quadratic objective. Let $\mathbf{W}_{s,a}^{\mathbf{v}} = \sum_{s'} P_a(s'|s)\mathbf{v}_{s'}$ so that $\mathbf{A}_{s,a}^{\mathbf{v}} = \mathbf{r}_{s,a} - \mathbf{v}_s + \gamma \mathbf{W}_{s,a}^{\mathbf{v}}$ where both $\mathbf{W}^{\mathbf{v}}$ and $\widehat{\mathbf{W}}^{\mathbf{v}}$ are seen as vectors in $\mathbb{R}^{|S| \times |A|}$.

If we had access to an estimator $\widehat{\mathbf{W}}^{\mathbf{v}}$ of $\mathbf{W}^{\mathbf{v}}$ such that for some $\epsilon \in (0,1)$:

$$\|\mathbf{W}^{\mathbf{v}} - \widehat{\mathbf{W}}^{\mathbf{v}}\|_{\infty} \leq \epsilon. \tag{29}$$

We can use $\widehat{\mathbf{W}}^{\mathbf{v}}$ to produce an estimator of $\mathbf{A}_{s,a}^{\mathbf{v}}$ via $\widehat{\mathbf{A}}_{s,a}^{\mathbf{v}} = \mathbf{r}_{s,a} - \mathbf{v}_s + \gamma \widehat{\mathbf{W}}_{s,a}^{\mathbf{v}}$ such that:

$$\|\widehat{\mathbf{A}}^{\mathbf{v}} - \mathbf{A}^{\mathbf{v}}\|_{\infty} \leq \gamma \epsilon.$$

We now consider the problem of estimating $\mathbf{W}^{\mathbf{v}}$ from samples. We assume the following stochastic setting:

1. The learner receives samples $\{(s_\ell, a_\ell, s'_\ell)\}_{\ell=1}^{\infty}$ such that $(s_\ell, a_\ell) \sim \mathbf{q}$ while $s'_\ell \sim P_{a_\ell}(\cdot|s_\ell)$. Let $N_t(s,a) = \sum_{\ell=1}^{t} \mathbf{1}(s_\ell, a_\ell = s, a)$.

2. Define $\widehat{\mathbf{W}}_{s,a}^{\mathbf{v}}(t) = \frac{1}{N_t(s,a)} \sum_{\ell=1}^{T} \mathbf{1}(s_\ell, a_\ell = s, a)\mathbf{v}_{s'_\ell}$. Notice that for all $s, a \in \mathcal{S} \times \mathcal{A}$, the estimator's noise $\xi_{s,a}(t) = \widehat{\mathbf{W}}_{s,a}^{\mathbf{v}}(t) - \mathbf{W}_{s,a}^{\mathbf{v}}$ satisfies $\mathbb{E}[\xi_{s,a}(t)|\mathcal{F}_{t-1}] = 0$ and $|\xi_{s,a}(t)| \leq 2\|\mathbf{v}'\|_{\infty}$. Where $\mathcal{F}_{t-1}$ is the sigma algebra corresponding to all the algorithmic choices up to round $t-1$.

**Lemma 15.** *Let $\{(s_\ell, a_\ell, s'_\ell)\}_{\ell=1}^{\infty}$ samples generated as above. Let $\widehat{\mathbf{W}}^{\mathbf{v}}(t)$ be the empirical estimator of $\mathbf{W}^{\mathbf{v}}$ defined as:*

$$\widehat{\mathbf{W}}_{s,a}^{\mathbf{v}}(t) = \frac{1}{N_t(s,a)} \sum_{\ell=1}^{t} \mathbf{1}(s_\ell, a_\ell = s, a)\mathbf{v}_{s'_\ell}.$$

*Where $N_t(s,a) = \sum_{\ell=1}^{t} \mathbf{1}(s_\ell, a_\ell = s, a)$. Let $\delta \in (0,1)$. With probability at least $1 - (2|S||A|)\delta$ for all $t \in \mathbb{N}$ such that $\ln\ln(2t) + \ln\frac{5.2}{\delta} \leq \frac{t\beta}{6}$ and for all $(s,a) \in \mathcal{S}$ simultaneously:*

$$|\mathbf{W}_{s,a}^{\mathbf{v}} - \widehat{\mathbf{W}}_{s,a}^{\mathbf{v}}(t)| \leq 8\|\mathbf{v}\|_{\infty} \left(\sqrt{\frac{\ln\ln(2t) + \ln\frac{10.4}{\delta}}{t\beta}} + \frac{\ln\ln(2t) + \ln\frac{10.4}{\delta}}{t\beta}\right).$$

*Proof.* Consider the martingale difference sequence $X_{s,a}(\ell) = \mathbf{1}(s_\ell, a_\ell = s, a)\left(\mathbf{W}_{s,a}^{\mathbf{v}} - \mathbf{v}_{s'_\ell}\right)$. Notice that for all $s, a \in \mathcal{S} \times \mathcal{A}$ $|X_{s,a}(t)| \leq 2\|\mathbf{v}'\|_{\infty}$ The process $S_t = \sum_{\ell=1}^{t} X_{s,a}(\ell)$ with variance process $W_t = \sum_{\ell=1}^{t} \mathbb{E}\left[X_{s,a}^2(\ell)|\mathcal{F}_{\ell-1}\right]$ satisfies the sub-$\psi_P$ condition of [16] with constant $c = 2\|\mathbf{v}'\|_{\infty}$ (see Bennet case in Table 3 of [16]). By Lemma 13 the bound:

$$S_t \leq 1.44\sqrt{(W_t \vee m)\left(1.4\ln\ln\left(2(W_t/m \vee 1)\right) + \ln\frac{5.2}{\delta}\right)} + 0.81\|\mathbf{v}\|_{\infty}\left(1.4\ln\ln\left(2\left(\frac{W_t}{m} \vee 1\right)\right) + \ln\frac{5.2}{\delta}\right)$$

holds for all $t \in \mathbb{N}$ with probability at least $1 - \delta$. Notice that $\mathbb{E}[X_{s,a}^2(\ell)|\mathcal{F}_{\ell-1}] \leq 4\|\mathbf{v}\|_\infty^2 \mathrm{Var}_\mathbf{q}(\mathbf{1}_{s,a}) = 4\|\mathbf{v}\|_\infty^2 \mathbf{q}_{s,a}(1 - \mathbf{q}_{s,a}) \leq \mathbf{q}_{s,a}\|\mathbf{v}\|_\infty^2$ and therefore $W_t \leq t\mathbf{q}_{s,a}\|\mathbf{v}\|_\infty^2$. We set $m = \mathbf{q}_{s,a}\|\mathbf{v}\|_\infty^2$. And obtain that with probability $1 - \delta$ and for all $t \in \mathbb{N}$:

$$\left| \underbrace{\frac{1}{N_t(s,a)} \sum_{\ell=1}^t \mathbf{1}(s_\ell = s, a_\ell = 1)\mathbf{v}_{s_\ell'} }_{\widehat{\mathbf{W}}_{s,a}^{\mathbf{v}}(t)} - \mathbf{W}_{s,a}^{\mathbf{v}} \right| \leq \frac{1}{N_t(s,a)}\left(1.44\|\mathbf{v}\|_\infty \sqrt{\mathbf{q}_{s,a}t\left(\ln\ln(2t) + \ln\frac{10.4}{\delta}\right)} + \right.$$

$$\left. 0.81\|\mathbf{v}\|_\infty\left(1.4\ln\ln(2t) + \ln\frac{10.2}{\delta}\right)\right) \qquad (30)$$

As a consequence of Lemma 14 we know that with probability at least $1 - \delta$ for all $t$ such that $\ln\ln(2t) + \ln\frac{5.2}{\delta} \leq \frac{t\beta}{6} \leq \frac{t\mathbf{q}_{s,a}}{6}$:

$$N_t(s,a) \geq \frac{t\mathbf{q}_{s,a}}{4}$$

Plugging this into Equation 30 and applying a union bound over all $s, a \in \mathcal{S} \times \mathcal{A}$ yields that for all $t$ such that $\ln\ln(2t) + \ln\frac{5.2}{\delta} \leq \frac{t\beta}{6} \leq \frac{t\mathbf{q}_{s,a}}{6}$ and with probability $1 - 2|S||A|\delta$ for all $s, a \in \mathcal{S}$ simultaneously:

$$|\mathbf{W}_{s,a}^{\mathbf{v}} - \widehat{\mathbf{W}}_{s,a}^{\mathbf{v}}(t)| \leq \frac{4}{t\mathbf{q}_{s,a}}\left(1.44\|\mathbf{v}\|_\infty\sqrt{t\mathbf{q}_{s,a}\ln\ln(2t) + t\ln\frac{10.4}{\delta}} + 0.81\|\mathbf{v}'\|_\infty\left(1.4\ln\ln(2t) + \ln\frac{10.4}{\delta}\right)\right)$$

$$\leq 8\|\mathbf{v}\|_\infty\left(\sqrt{\frac{\ln\ln(2t) + \ln\frac{10.4}{\delta}}{t\mathbf{q}_{s,a}}} + \frac{\ln\ln(2t) + \ln\frac{10.4}{\delta}}{t\mathbf{q}_{s,a}}\right)$$

$$\leq 8\|\mathbf{v}\|_\infty\left(\sqrt{\frac{\ln\ln(2t) + \ln\frac{10.4}{\delta}}{t\beta}} + \frac{\ln\ln(2t) + \ln\frac{10.4}{\delta}}{t\beta}\right).$$

The result follows. $\qquad \square$

We can now derive a concentration result for $\widehat{\mathbf{A}}_{s,a}^{\mathbf{v}}(t) = \mathbf{r}_{s,a} - \mathbf{v}_s + \gamma\widehat{\mathbf{W}}_{s,a}^{\mathbf{v}}(t)$, the advantage estimator resulting from $\widehat{\mathbf{W}}_{s,a}^{\mathbf{v}}(t)$:

**Corollary 3.** *Let $\delta \in (0,1)$. With probability at least $1 - (2|S||A|)\delta$ for all $t \in \mathbb{N}$ such that $\ln\ln(2t) + \ln\frac{5.2}{\delta} \leq \frac{t\beta}{6}$ and for all $(s,a) \in \mathcal{S}$ simultaneously:*

$$|\mathbf{A}_{s,a}^{\mathbf{v}} - \widehat{\mathbf{A}}_{s,a}^{\mathbf{v}}(t)| \leq 8\gamma\|\mathbf{v}\|_\infty\left(\sqrt{\frac{\ln\ln(2t) + \ln\frac{10.4}{\delta}}{t\beta}} + \frac{\ln\ln(2t) + \ln\frac{10.4}{\delta}}{t\beta}\right).$$

*And therefore:*

$$|\mathbf{A}_{s,a}^{\mathbf{v}} - \widehat{\mathbf{A}}_{s,a}^{\mathbf{v}}(t)| \leq 16\gamma\|\mathbf{v}\|_\infty\sqrt{\frac{\ln\ln(2t) + \ln\frac{10.4}{\delta}}{t\beta}}$$

### H.1 Estimating the Gradients

**Lemma 16.** *If $\xi \in \mathbb{R}$ such that $|\xi| \leq \epsilon < 1$, and $y \in \mathbb{R}$, then:*

$$\exp(y)(1 - \epsilon) \leq \exp(y + \xi) \leq \exp(y)(1 + 2\epsilon)$$

*Proof.* Notice that for $\epsilon \in (0,1)$:

$$\exp(\epsilon) \leq 1 + 2\epsilon, \quad \text{and } 1 - \epsilon \leq \exp(-\epsilon).$$

The result follows by noting that:

$$\exp(y)\exp(-|\xi|) \leq \exp(y + \xi) \leq \exp(y)\exp(|\xi|).$$

$\qquad \square$

A simple consequence of Lemma 16 is the following:

**Lemma 17.** *Let $\epsilon \in (0, 1/2)$. If $\mathbf{C}, \widehat{\mathbf{C}} \in \mathbb{R}^{|\mathcal{S}| \times |\mathcal{A}|}$ and $\widehat{\mathbf{b}}, \mathbf{b} \in \mathbb{R}_+^{|\mathcal{S}| \times |\mathcal{A}|}$ are two vectors satisfying:*

$$\|\widehat{\mathbf{C}} - \mathbf{C}\|_\infty \leq \epsilon, \qquad |\widehat{\mathbf{b}}_{s,a} - \mathbf{b}_{s,a}| \leq \epsilon \mathbf{b}_{s,a}.$$

*For all $s, a \in \mathcal{S} \times \mathcal{A}$ define $\mathbf{B}_{s,a} = \frac{\exp(\mathbf{C}_{s,a})}{\mathbf{Z}}$ and $\widehat{\mathbf{B}}_{s,a} = \frac{\exp(\widehat{\mathbf{C}}_{s,a})}{\widehat{\mathbf{Z}}}$ where $\mathbf{Z} = \sum_{s,a} \exp(\mathbf{C}_{s,a}) \mathbf{b}_{s,a}$ and $\widehat{\mathbf{Z}} = \sum_{s,a} \exp(\widehat{\mathbf{C}}_{s,a}) \widehat{\mathbf{b}}_{s,a}$:*

$$\left| \widehat{\mathbf{B}}_{s,a} - \mathbf{B}_{s,a} \right| \leq 38\epsilon \mathbf{B}_{s,a} \leq 38\epsilon.$$

*Proof.* Let's define an intermediate $\widetilde{\mathbf{B}}_{s,a} = \frac{\exp(\mathbf{C}_{s,a}) \widehat{\mathbf{b}}_{s,a}}{\widetilde{\mathbf{Z}}}$ where $\widetilde{\mathbf{Z}} = \sum_{s,a} \exp(\mathbf{C}_{s,a}) \widehat{\mathbf{b}}_{s,a}$. By Lemma 16 we can conclude that for any $s, a \in \mathcal{S} \times \mathcal{A}$:

$$\widetilde{\mathbf{B}}_{s,a} \frac{1 - \epsilon}{1 + 2\epsilon} \leq \widehat{\mathbf{B}}_{s,a} \leq \frac{1 + 2\epsilon}{1 - \epsilon} \widetilde{\mathbf{B}}_{s,a}$$

And therefore:

$$\widehat{\mathbf{B}}_{s,a}, \widetilde{\mathbf{B}}_{s,a} \in \left[ \widetilde{\mathbf{B}}_{s,a} \frac{1 - \epsilon}{1 + 2\epsilon}, \frac{1 + 2\epsilon}{1 - \epsilon} \widetilde{\mathbf{B}}_{s,a} \right]$$

Which in turn implies that:

$$\left| \widehat{\mathbf{B}}_{s,a} - \widetilde{\mathbf{B}}_{s,a} \right| \leq \left( \frac{1 + 2\epsilon}{1 - \epsilon} - \frac{1 - \epsilon}{1 + 2\epsilon} \right) \widetilde{\mathbf{B}}_{s,a} \leq 15\epsilon \widetilde{\mathbf{B}}_{s,a}.$$

We now bound $|\widetilde{\mathbf{B}}_{s,a} - \mathbf{B}_{s,a}|$. By assumption for all $s, a \in \mathcal{S} \times \mathcal{A}$, it follows that $\widehat{\mathbf{b}}_{s,a}(1 - \epsilon) \leq \mathbf{b}_{s,a} \leq \widehat{\mathbf{b}}_{s,a}(1 + \epsilon)$ and therefore:

$$\frac{\mathbf{B}_{s,a}}{1 + \epsilon} \leq \widetilde{\mathbf{B}}_{s,a} \leq \frac{\mathbf{B}_{s,a}}{1 - \epsilon}$$

And therefore:

$$\widetilde{\mathbf{B}}_{s,a}, \mathbf{B}_{s,a} \in \left[ \frac{\mathbf{B}_{s,a}}{1 + \epsilon}, \frac{\mathbf{B}_{s,a}}{1 - \epsilon} \right].$$

Hence:

$$\left| \widetilde{\mathbf{B}}_{s,a} - \mathbf{B}_{s,a} \right| \leq \left( \frac{1}{1 - \epsilon} - \frac{1}{1 + \epsilon} \right) \mathbf{B}_{s,a} \leq \frac{8}{3} \epsilon \mathbf{B}_{s,a}.$$

And therefore:

$$\left| \widehat{\mathbf{B}}_{s,a} - \mathbf{B}_{s,a} \right| \leq |\widehat{\mathbf{B}}_{s,a} - \widetilde{\mathbf{B}}_{s,a}| + |\widetilde{\mathbf{B}}_{s,a} - \mathbf{B}_{s,a}| \leq 15\epsilon \widetilde{\mathbf{B}}_{s,a} + \frac{8}{3} \epsilon \mathbf{B}_{s,a} \leq \left( 15\epsilon(1 + \frac{8}{3}\epsilon) + \frac{8}{3}\epsilon \right) \mathbf{B}_{s,a} \leq 38\epsilon \mathbf{B}_{s,a}.$$

The result follows.

$\square$

If we set $\mathbf{C} = \eta \mathbf{A}^\mathbf{v}, \widehat{\mathbf{C}} = \eta \widehat{\mathbf{A}}^\mathbf{v}$ we obtain the following corollary of Lemma 17:

**Corollary 4.** *Let $\epsilon \in (0, 1/2)$. If $\widehat{\mathbf{A}}^\mathbf{v}$ and $\widehat{\mathbf{q}}$ satisfies:*

$$\|\widehat{\mathbf{A}}^\mathbf{v} - \mathbf{A}^\mathbf{v}\|_\infty \leq \epsilon, \quad and \quad |\widehat{\mathbf{q}}_{s,a} - \mathbf{q}_{s,a}| \leq \epsilon \mathbf{q}_{s,a}$$

*Then:*

$$\left| \widehat{\mathbf{B}}_{s,a}^\mathbf{v} - \mathbf{B}_{s,a}^\mathbf{v} \right| \leq 111\eta\epsilon \mathbf{B}_{s,a}^\mathbf{v} \leq 111\eta\epsilon.$$

We can combine the sample complexity results of Corollary 3 and the approximation results of Corollary 4 and Lemma 14 to obtain:

**Corollary 5.** *If $\delta, \xi \in (0,1)$, with probability at least $1 - (4|\mathcal{S}||\mathcal{A}|\delta)$ for all $t$ such that:*

$$\frac{t}{\ln\ln(2t)} \geq \frac{120(\ln\frac{10.4}{\delta} + 1)}{\beta\xi^2} \max\left(480\eta^2\gamma^2\|\mathbf{v}\|_\infty^2, 1\right)$$

*then for all $(s,a) \in \mathcal{S} \times \mathcal{A}$ simultaneously:*

$$\left|\widehat{\mathbf{B}}_{s,a}^{\mathbf{v}}(t) - \mathbf{B}_{s,a}^{\mathbf{v}}\right| \leq \xi\mathbf{B}_{s,a}^{\mathbf{v}} \leq \frac{\xi}{\beta}, \quad \text{and} \quad \widehat{\mathbf{B}}_{s,a}^{\mathbf{v}} \leq \mathbf{B}_{s,a}^{\mathbf{v}}(1 + \frac{\xi}{\beta}) \leq \frac{1}{\beta}(1 + \frac{\xi}{\beta}).$$

## H.2 Biased Stochastic Gradients

Notice that:

$$(\nabla_{\mathbf{v}}J_D(\mathbf{v}))_s = (1-\gamma)\boldsymbol{\mu}_s + \gamma\sum_{s',a}\frac{\exp\left(\eta\mathbf{A}_{s',a}^{\mathbf{v}}\right)\mathbf{q}_{s',a}}{\mathbf{Z}}P_a(s|s') - \sum_a\frac{\exp\left(\eta\mathbf{A}_{s,a}^{\mathbf{v}}\right)\mathbf{q}_{s,a}}{\mathbf{Z}}$$

$$= (1-\gamma)\boldsymbol{\mu}_s + \gamma\mathbb{E}_{(s',a)\sim\mathbf{q},s''\sim P_a(\cdot|s')}\left[\mathbf{B}_{s',a}^{\mathbf{v}}\mathbf{1}(s''=s)\right] - \mathbb{E}_{(s',a)\sim\mathbf{q}}\left[\mathbf{B}_{s,a}^{\mathbf{v}}\mathbf{1}(s'=s)\right]$$

$$= (1-\gamma)\boldsymbol{\mu}_s + \mathbb{E}_{(s',a)\sim\mathbf{q},s''\sim P_a(\cdot|s')}\left[\mathbf{B}_{s',a}^{\mathbf{v}}\left(\gamma\mathbf{1}(s''=s) - \mathbf{1}(s'=s)\right)\right],$$

We now proceed to bound the bias of this estimator and prove a more fine grained version of Lemma 8.

**Lemma 18.** *Let $\delta, \xi \in (0,1)$. With probability at least $1-\delta$ for all $t \in \mathbb{N}$ such that*

$$\frac{t}{\ln\ln(2t)} \geq \frac{120(\ln\frac{41.6|\mathcal{S}||\mathcal{A}|}{\delta} + 1)}{\beta\xi^2} \max\left(480\eta^2\gamma^2\|\mathbf{v}\|_\infty^2, 1\right)$$

*the plugin estimator $\widehat{\nabla}_{\mathbf{v}}J_D(\mathbf{v})$ satisfies:*

$$\max_{u\in\{1,2,\infty\}}\left\|\widehat{\nabla}_{\mathbf{v}}J_D(\mathbf{v}) - \mathbb{E}_{s_{t+1},a_{t+1},s'_{t+1}}\left[\widehat{\nabla}_{\mathbf{v}}J_D(\mathbf{v})\Big|\widehat{\mathbf{B}}^{\mathbf{v}}(t)\right]\right\|_u \leq \frac{4}{\beta}(1 + \frac{\xi}{\beta}) \tag{31}$$

$$\max_{u\in\{1,2,\infty\}}\left\|\mathbb{E}\left[\widehat{\nabla}_{\mathbf{v}}J_D(\mathbf{v})\right] - \nabla_{\mathbf{v}}J_D(\mathbf{v})\right\|_u \leq 2(1+\gamma)\xi(1 + \frac{\xi}{\beta}), \tag{32}$$

$$\mathbb{E}\left[\left\|\widehat{\nabla}_{\mathbf{v}}J_D(\mathbf{v}) - \mathbb{E}_{s_{t+1},a_{t+1},s'_{t+1}}[\widehat{\nabla}_{\mathbf{v}}J_D(\mathbf{v})\Big|\widehat{\mathbf{B}}^{\mathbf{v}}(t)]\right\|_2^2\Big|\widehat{\mathbf{B}}^{\mathbf{v}}(t)\right] \leq (1+\gamma^2)(1+4\xi)\frac{1}{\beta}(1 + \frac{\xi}{\beta}) \tag{33}$$

*Proof.* As a consequence of Corollary 5, we can conclude that for all $t$ satisfying the assumptions of the Lemma and with probability at leat $1-\delta$ simultaneously for all $(s,a) \in \mathcal{S} \times \mathcal{A}$:

$$\left|\widehat{\mathbf{B}}_{s,a}^{\mathbf{v}}(t) - \mathbf{B}_{s,a}^{\mathbf{v}}\right| \leq \xi\mathbf{B}_{s,a}^{\mathbf{v}}(1 + \frac{\xi}{\beta}), \quad \text{and} \quad \widehat{\mathbf{B}}_{s,a}^{\mathbf{v}} \leq \mathbf{B}_{s,a}^{\mathbf{v}}(1 + \frac{\xi}{\beta}) \leq \frac{1}{\beta}(1 + \frac{\xi}{\beta}). \tag{34}$$

Let's start by bounding the first term. Notice that $\widehat{\nabla}_{\mathbf{v}}J_D(\mathbf{v}) - (1-\gamma)\boldsymbol{\mu}$ has at most 2 nonzero entries and therefore:

$$\max_{u\in\{1,2,\infty\}}\|\widehat{\nabla}_{\mathbf{v}}J_D(\mathbf{v}) - (1-\gamma)\boldsymbol{\mu}\|_u \leq \frac{2}{\beta}(1 + \frac{\xi}{\beta}).$$

Therefore for all $u \in \{1,2,\infty\}$:

$$\left\|\mathbb{E}_{s_{t+1},a_{t+1},s'_{t+1}}\left[\widehat{\nabla}_{\mathbf{v}}J_D(\mathbf{v}) - (1-\gamma)\boldsymbol{\mu}\Big|\widehat{\mathbf{B}}^{\mathbf{v}}(t)\right]\right\|_u \leq \mathbb{E}_{s_{t+1},a_{t+1},s'_{t+1}}\left[\|\widehat{\nabla}_{\mathbf{v}}J_D(\mathbf{v}) - (1-\gamma)\boldsymbol{\mu}\|_u\Big|\widehat{\mathbf{B}}^{\mathbf{v}}(t)\right] \leq \frac{2}{\beta}(1+\frac{\xi}{\beta}).$$

$$\left\| \widehat{\nabla}_{\mathbf{v}} J_D(\mathbf{v}) - \mathbb{E}\left[ \widehat{\nabla}_{\mathbf{v}} J_D(\mathbf{v}) \Big| \widehat{\mathbf{B}}^{\mathbf{v}}(t) \right] \right\|_u \le \left\| \widehat{\nabla}_{\mathbf{v}} J_D(\mathbf{v}) - (1-\gamma)\boldsymbol{\mu} \right\|_u + \left\| \mathbb{E}\left[ \widehat{\nabla}_{\mathbf{v}} J_D(\mathbf{v}) \Big| \widehat{\mathbf{B}}^{\mathbf{v}}(t) \right] - (1-\gamma)\boldsymbol{\mu} \right\|_u$$
$$\le \frac{4}{\beta}\left(1 + \frac{\xi}{\beta}\right)$$

Furthermore, notice that the following estimator of $\nabla_{\mathbf{v}} J_D(\mathbf{v})$ is unbiased:

$$\left( \widetilde{\nabla}_{\mathbf{v}} J_D(\mathbf{v}) \right)_s = (1-\gamma)\boldsymbol{\mu}_s + \mathbf{B}^{\mathbf{v}}_{s_{t+1},a_{t+1}}(t) \left( \gamma \mathbf{1}(s'_{t+1} = s) - \mathbf{1}(s_{t+1} = s) \right).$$

We conclude that for all $s \in \mathcal{S}$:

$$\left( \widehat{\nabla}_{\mathbf{v}} J_D(\mathbf{v}) \right)_s - \left( \widetilde{\nabla}_{\mathbf{v}} J_D(\mathbf{v}) \right)_s = \left( \gamma \mathbf{1}(s'_{t+1} = s) - \mathbf{1}(s_{t+1} = s) \right) \left( \widehat{\mathbf{B}}^{\mathbf{v}}_{s_{t+1},a_{t+1}}(t) - \mathbf{B}^{\mathbf{v}}_{s_{t+1},a_{t+1}}(t) \right)$$

Consequently $\widehat{\nabla}_{\mathbf{v}} J_D(\mathbf{v}) - \widetilde{\nabla}_{\mathbf{v}} J_D(\mathbf{v})$ has at most 2 nonzero entries. Now observe that any nonzero entry $s$ satisfies:

$$\left| \mathbb{E}\left[ \left( \widehat{\nabla}_{\mathbf{v}} J_D(\mathbf{v}) \right)_s \right] - (\nabla_{\mathbf{v}} J_D(\mathbf{v}))_s \right| = \left| \mathbb{E}_{s_{t+1},a_{t+1} \sim \mathbf{q}} \left[ \left( \widehat{\nabla}_{\mathbf{v}} J_D(\mathbf{v}) \right)_s - \left( \widetilde{\nabla}_{\mathbf{v}} J_D(\mathbf{v}) \right)_s \right] \right|$$
$$\le \mathbb{E}_{s_{t+1},a_{t+1} \sim \mathbf{q}} \left[ \left| \gamma \mathbf{1}(s'_{t+1} = s) - \mathbf{1}(s_{t+1} = s) \right| \left| \widehat{\mathbf{B}}^{\mathbf{v}}_{s_{t+1},a_{t+1}}(t) - \mathbf{B}^{\mathbf{v}}_{s_{t+1},a_{t+1}}(t) \right| \right]$$
$$\overset{(i)}{\le} \mathbb{E}_{s_{t+1},a_{t+1} \sim \mathbf{q}} \left[ \left( \gamma \mathbf{1}(s'_{t+1} = s) + \mathbf{1}(s_{t+1} = s) \right) \xi \mathbf{B}^{\mathbf{v}}_{s_{t+1},a_{t+1}} \left(1 + \frac{\xi}{\beta}\right) \right]$$
$$\le (1+\gamma)\xi\left(1 + \frac{\xi}{\beta}\right) \mathbb{E}_{s_{t+1},a_{t+1} \sim \mathbf{q}} \left[ \mathbf{B}_{s_{t+1},a_{t+1}} \right]$$
$$= (1+\gamma)\xi\left(1 + \frac{\xi}{\beta}\right)$$

Inequality $(i)$ holds by the triangle inequality and Equation 34 and because $\mathbf{B}^{\mathbf{v}}_{s,a} \ge 0$. This finishes the proof of the first result. Since $\widehat{\nabla}_{\mathbf{v}} J_D(\mathbf{v}) - \widetilde{\nabla}_{\mathbf{v}} J_D(\mathbf{v})$ has at most 2 nonzero entries for all $u \in \{1, 2, \infty\}$:

$$\left\| \mathbb{E}\left[ \left( \widehat{\nabla}_{\mathbf{v}} J_D(\mathbf{v}) \right)_s \right] - (\nabla_{\mathbf{v}} J_D(\mathbf{v}))_s \right\|_u \le 2(1+\gamma)\xi\left(1 + \frac{\xi}{\beta}\right)$$

The second inequality follows.

Recall that for any $s$:

$$\left( \widehat{\nabla}_{\mathbf{v}} J_D(\mathbf{v}) \right)_s = (1-\gamma)\boldsymbol{\mu}_s + \widehat{\mathbf{B}}^{\mathbf{v}}_{s_{t+1},a_{t+1}(t)} \left( \gamma \mathbf{1}(s'_{t+1} = s) - \mathbf{1}(s_{t+1} = s) \right).$$

Observe that:

$$\mathbb{E}\left[ \left\| \widehat{\nabla}_{\mathbf{v}} J_D(\mathbf{v}) - \mathbb{E}[\widehat{\nabla}_{\mathbf{v}} J_D(\mathbf{v}) \Big| \widehat{\mathbf{B}}^{\mathbf{v}}(t)] \right\|_2^2 \Big| \widehat{\mathbf{B}}^{\mathbf{v}}(t) \right] \le \mathbb{E}\left[ \left\| \widehat{\nabla}_{\mathbf{v}} J_D(\mathbf{v}) \right\|_2^2 \Big| \widehat{\mathbf{B}}^{\mathbf{v}}(t) \right]$$
$$= \sum_{s',a} \left( \widehat{\mathbf{B}}^{\mathbf{v}}_{s',a}(t) \right)^2 \gamma^2 \mathbf{q}_{s',a} P_a(s|s') +$$
$$\sum_a \left( \widehat{\mathbf{B}}^{\mathbf{v}}_{s,a}(t) \right)^2 \mathbf{q}_{s,a} (1 - 2\gamma) P_a(s|s)$$
$$\le (1+\gamma^2) \mathbb{E}_{(s',a) \sim \widehat{\mathbf{q}}(t)\widehat{\mathbf{B}}^{\mathbf{v}}(t)} \left[ \widehat{\mathbf{B}}^{\mathbf{v}}_{s',a}(t) \frac{\mathbf{q}_{s',a}}{\widehat{\mathbf{q}}_{s',a}} \right]$$
$$\overset{(i)}{\le} (1+\gamma^2)(1 + 4\xi) \frac{1}{\beta}\left(1 + \frac{\xi}{\beta}\right).$$

Inequality $(i)$ follows because $\widehat{\mathbf{B}}_{s,a} \mathbf{q}_{s,a} \le \frac{\mathbf{q}_{s,a}}{\widehat{\mathbf{q}}_{s,a}} \le (1+4\xi)$ and because by Corollary 5 we have that $\widehat{\mathbf{B}}^{\mathbf{v}}_{s,a} \le \frac{1}{\beta}\left(1 + \frac{\xi}{\beta}\right)$.

The result follows. $\qquad\qquad\square$

Combining the guarantees of Lemma 9 and 8 for Algorithm 3 applied to the objective function $J_D$:

**Lemma 19.** *Let* $\xi_t = \min(\sqrt{\frac{c'}{t}}, \beta)$ *for all* $t$ *where* $c' = 2(|\mathcal{S}|+1)^2\eta^2 D^2 + \frac{320}{\beta^2} + 240$ *and* $D = \frac{1}{1-\gamma}\left(1 + \frac{\log\frac{|\mathcal{S}||\mathcal{A}|}{\beta\rho}}{\eta}\right)$. *If* $n(t)$ *is such that:*

$$\frac{n(t)}{\ln\ln(2n(t))} \geq \frac{120\left(\ln\frac{83.2|\mathcal{S}||\mathcal{A}|t^2}{\delta} + 1\right)}{\beta\xi_t^2}\max\left(280\eta^2\gamma^2\|\mathbf{v}_t\|_\infty^2, 1\right) \tag{35}$$

*And* $\tau_t = \frac{c}{\sqrt{t}}$ *where* $c = \frac{D}{2\sqrt{c'}}$ *then for all* $t \geq 1$ *we have that with probability at least* $1 - 2\delta$ *and simulataneously for all* $T \in \mathbb{N}$ :

$$J_D\left(\frac{1}{T}\sum_{t=1}^T \mathbf{v}_t\right) \leq J_D(\mathbf{v}_\star) + \frac{36D}{\sqrt{T}}\max\left((|\mathcal{S}|+1)\eta D, \frac{18 + 16\sqrt{\ln\ln(2T) + \ln\frac{5.2}{\delta}}}{\beta}, 16\right)$$

*Proof.* We will make use of Lemmas 8 and 9. We identify $\boldsymbol{\epsilon}_t = \widehat{\nabla}_\mathbf{v} J_D(\mathbf{v}_t) - \mathbb{E}\left[\widehat{\nabla}_\mathbf{v} J_D(\mathbf{v}_t)\Big|\widehat{\mathbf{B}}^{\mathbf{v}_t}(n(t))\right]$ and $\mathbf{b}_t = \nabla_\mathbf{v} J_D(\mathbf{v}_t) - \mathbb{E}\left[\widehat{\nabla}_{\mathbf{v}_t} J_D(\mathbf{v}_t)\Big|\widehat{\mathbf{B}}^{\mathbf{v}_t}(n(t))\right]$. As a consequence of Cauchy-Schwartz and Lemma 8 we see that if $n(t)$ is such that:

$$\frac{n(t)}{\ln\ln(2n(t))} \geq \frac{120\left(\ln\frac{83.2|\mathcal{S}||\mathcal{A}|t^2}{\delta} + 1\right)}{\beta\xi_t^2}\max\left(280\eta^2\gamma^2\|\mathbf{v}_t\|_\infty^2, 1\right)$$

Then for all $t$ with probability at least $1 - \frac{\delta}{2t^2}$ the bounds in Equations 31, 32, and 33 in Lemma 8 hold and therefore:

$$|\langle\boldsymbol{\epsilon}_t, \mathbf{v}_t - \mathbf{v}_\star\rangle| \leq \|\mathbf{v}_t - \mathbf{v}_\star\|_\infty\|\boldsymbol{\epsilon}_t\|_1 \leq \frac{1}{1-\gamma}\left(1 + \frac{\log\frac{|\mathcal{S}||\mathcal{A}|}{\beta\rho}}{\eta}\right)\frac{4}{\beta}(1 + \frac{\xi_t}{\beta}) \stackrel{(i)}{\leq} \underbrace{\frac{1}{1-\gamma}\left(1 + \frac{\log\frac{|\mathcal{S}||\mathcal{A}|}{\beta\rho}}{\eta}\right)\frac{8}{\beta}}_{:=U_1}.$$

Where inequality $(i)$ holds by the assumption $\xi_t \leq \beta$. Notice that $X_t = \langle\boldsymbol{\epsilon}_t, \mathbf{v}_t - \mathbf{v}_\star\rangle$ is a martingale difference sequence. A simple application of Lemma 13 yields that with probability at least $1 - \delta$ for all $t \in \mathbb{N}$:

$$-\sum_{t=1}^T \langle\boldsymbol{\epsilon}_t, \mathbf{x}_t - \mathbf{x}_\star\rangle \leq 2U_1\sqrt{t\left(\ln\frac{2t^2}{\delta}\right)} \tag{36}$$

Similarly observe that for all $t$ with probability at least $1 - \frac{\delta}{2t^2}$, since the bounds in Equations 31, 32, and 33 in Lemma 8 hold,

$$\|\mathbf{b}_t\|_1 = \left\|\nabla_\mathbf{v} J_D(\mathbf{v}_t) - \mathbb{E}\left[\widehat{\nabla}_{\mathbf{v}_t} J_D(\mathbf{v}_t)\Big|\widehat{\mathbf{B}}^{\mathbf{v}_t}(n(t))\right]\right\|_1 \leq 2(1+\gamma)\xi_t(1 + \frac{\xi_t}{\beta}) \tag{37}$$

Notice that similarly and for all $t$ with probability at least $1 - \frac{\delta}{2t^2}$, since the bounds in Equations 31, 32, and 33 in Lemma 8 hold:

$$\|\boldsymbol{\epsilon}_t\|_2^2 \leq \frac{16}{\beta^2}\left(1 + \frac{\xi_t}{\beta}\right)^2, \quad \text{and} \quad \|\mathbf{b}_t\|_2^2 \leq 4(1+\gamma)^2\xi_t^2(1 + \frac{\xi_t}{\beta})^2$$

Finally we show a bound on the $l_2$ norm of the gradient of $J_D$. Since $\mathbf{v}_\star \in \mathcal{D} = \left\{\mathbf{v} \text{ s.t. } \|\mathbf{v}\|_\infty \leq \frac{1}{1-\gamma}\left(1 + \frac{\log\frac{|\mathcal{S}||\mathcal{A}|}{\beta\rho}}{\eta}\right)\right\}$. Recall that by Lemma 3, we have that $J_D$ is $(|\mathcal{S}|+1)\eta$-smooth in the $\|\cdot\|_\infty$ norm. Therefore by Lemma 12:

$$\|\nabla J_D(\mathbf{v}_t)\|_1 \le (|\mathcal{S}|+1)\frac{\eta}{1-\gamma}\left(1+\frac{\log\frac{|\mathcal{S}||\mathcal{A}|}{\beta\rho}}{\eta}\right)$$

Since $\|\nabla J_D(\mathbf{v}_t)\|_2 \le \|\nabla J_D(\mathbf{v}_t)\|_1$ this in turn implies that:

$$\|\nabla J_D(\mathbf{v}_t)\|_2^2 \le (|\mathcal{S}|+1)^2\frac{\eta^2}{(1-\gamma)^2}\left(1+\frac{\log\frac{|\mathcal{S}||\mathcal{A}|}{\beta\rho}}{\eta}\right)^2.$$

We now invoke the guarantees of Lemma 9 to show that with probability $1-2\delta$ and simultaneously for all $T \in \mathbb{N}$:

$$\sum_{t=1}^T J_D(\mathbf{v}_t) - J_D(\mathbf{v}_\star) \le \sum_{t=1}^T \frac{\|\mathbf{v}_t-\mathbf{v}_\star\|^2 - \|\mathbf{v}_{t+1}-\mathbf{v}_\star\|^2}{2\tau_t} +$$

$$\tau_t\left(2(|\mathcal{S}|+1)^2\frac{\eta^2}{(1-\gamma)^2}\left(1+\frac{\log\frac{|\mathcal{S}||\mathcal{A}|}{\beta\rho}}{\eta}\right)^2 + \frac{80}{\beta^2}\left(1+\frac{\xi_t}{\beta}\right)^2 + 20(1+\gamma)^2\xi_t^2(1+\frac{\xi_t}{\beta})^2\right) +$$

$$2(1+\gamma)\xi_t(1+\frac{\xi_t}{\beta})\times\frac{1}{1-\gamma}\left(1+\frac{\log\frac{|\mathcal{S}||\mathcal{A}|}{\beta\rho}}{\eta}\right) + 2U_1\sqrt{T\left(\ln\frac{2t^2}{\delta}\right)}$$

$$\overset{(i)}{\le} \sum_{t=1}^T \frac{\|\mathbf{v}_t-\mathbf{v}_\star\|^2 - \|\mathbf{v}_{t+1}-\mathbf{v}_\star\|^2}{2\tau_t} + \tau_t\left(2(|\mathcal{S}|+1)^2\eta^2 D^2 + \frac{320}{\beta^2} + 240\right) + 8D\xi_t +$$

$$2U_1\sqrt{T\left(\ln\frac{2t^2}{\delta}\right)}$$

Recall that $U_1 = \frac{1}{1-\gamma}\left(1+\frac{\log\frac{|\mathcal{S}||\mathcal{A}|}{\beta\rho}}{\eta}\right)\frac{8}{\beta} = \frac{8D}{\beta}$ and where $D = \frac{1}{1-\gamma}\left(1+\frac{\log\frac{|\mathcal{S}||\mathcal{A}|}{\beta\rho}}{\eta}\right)$. Inequality $(i)$ holds because $\xi_t \le \beta$ and because $\gamma \le 1$. Let $\tau_t = \frac{c}{\sqrt{t}}$ for some constant to be specified later and let's analyze the terms in the sum above that depend on these $\tau_t$ values:

$$\sum_{t=1}^T \frac{\|\mathbf{v}_t-\mathbf{v}_\star\|^2 - \|\mathbf{v}_{t+1}-\mathbf{v}_\star\|^2}{2\tau_t} = -\frac{\|\mathbf{v}_{T+1}-\mathbf{v}_\star\|^2}{2\tau_T} + \frac{1}{2c}\sum_{t=1}^T \|\mathbf{v}_t-\mathbf{v}_\star\|^2\left(\sqrt{t}-\sqrt{t-1}\right)$$

$$\le \frac{D^2}{2c}\sqrt{T}$$

The second term can be bounded as:

$$\sum_{t=1}^T \tau_t c' = cc'\sum_{t=1}^T \frac{1}{\sqrt{t}} \le cc'2\sqrt{T}$$

Where $c' = 2(|\mathcal{S}|+1)^2\eta^2 D^2 + \frac{320}{\beta^2} + 240$. Therefore under this assumption we obtain:

$$\sum_{t=1}^T J_D(\mathbf{v}_t) - J_D(\mathbf{v}_\star) \le \frac{D^2}{2c}\sqrt{T} + cc'2\sqrt{T} + 8D\left(\sum_{t=1}^T \xi_t\right) + 2U_1\sqrt{T\left(\ln\frac{2t^2}{\delta}\right)}.$$

The minimizing choice for $c$ equals $c = \frac{D}{2\sqrt{c'}}$. And in this case:

$$\sum_{t=1}^{T} J_D(\mathbf{v}_t) - J_D(\mathbf{v}_\star) \le 2D\sqrt{c'T} + 8D\left(\sum_{t=1}^{T} \xi_t\right) + 2U_1\sqrt{T\left(\ln\frac{2t^2}{\delta}\right)}$$

If we set $\xi_t = \min(\sqrt{\frac{c'}{t}}, \beta)$ we get:

$$\sum_{t=1}^{T} J_D(\mathbf{v}_t) - J_D(\mathbf{v}_\star) \le 18D\sqrt{c'T} + 2U_1\sqrt{T\left(\ln\frac{2t^2}{\delta}\right)}$$

$$\overset{(i)}{\le} 36D\max\left((|\mathcal{S}|+1)\eta D, \frac{18}{\beta}, 16\right)\sqrt{T} + 2U_1\sqrt{T\left(\ln\frac{2t^2}{\delta}\right)}$$

$$\le 36D\max\left((|\mathcal{S}|+1)\eta D, \frac{18 + 16\sqrt{\ln\ln(2T) + \ln\frac{5.2}{\delta}}}{\beta}, 16\right)\sqrt{T}$$

Inequality $(i)$ holds because $\sqrt{c'} \le 2\max\left((|\mathcal{S}|+1)\eta D, \frac{18}{\beta}, 16\right)$.

We conclude that:

$$J_D\left(\frac{1}{T}\sum_{t=1}^{T}\mathbf{v}_t\right) \overset{(i)}{\le} \frac{1}{T}\sum_{t=1}^{T} J_D(\mathbf{v}_t)$$

$$\le J_D(\mathbf{v}_\star) + \frac{36D}{\sqrt{T}}\max\left((|\mathcal{S}|+1)\eta D, \frac{18 + 16\sqrt{\ln\ln(2T) + \ln\frac{5.2}{\delta}}}{\beta}, 16\right)$$

Inequality $(i)$ holds by convexity of $J_D$. The result follows. $\qquad\square$

We are ready to present the proof of Lemma 10 which corresponds to a simplified version of Lemma 19.

### H.3   Proof of Lemma 10

**Lemma 10.** *We assume* $\eta \ge \frac{4}{\beta}$. *Set* $\xi_t = \frac{8|\mathcal{S}|\eta D}{\sqrt{t}}$ *and* $\tau_t = \frac{1}{16|\mathcal{S}|\eta\sqrt{t}}$. *If we take t gradient steps using* $n(t)$ *samples from* $\mathbf{q} \times \mathbf{P}$ *(possibly reusing the samples for multiple gradient computations) with* $n(t)$ *satisfying* $n(t) \ge \frac{525t\left(\ln\frac{100|\mathcal{S}||\mathcal{A}|t^2}{\delta}+1\right)^3}{\beta|\mathcal{S}|^2}$. *Then for all* $t \ge 1$ *we have that with probability at least* $1 - 3\delta$ *and simultaneously for all* $t \in \mathbb{N}$ *such that* $t \ge \frac{64|\mathcal{S}|^2\eta^2 D^2}{\beta}$:

$$J_D\left(\frac{1}{t}\sum_{\ell=1}^{t}\mathbf{v}_\ell\right) \le J_D(\mathbf{v}_\star) + \widetilde{\mathcal{O}}\left(\frac{D^2|\mathcal{S}|\eta}{\sqrt{t}}\right).$$

*Proof.* First note that the $c'$ of Lemma 19 satisfies $c' = \max\left(2(|\mathcal{S}|+1)^2\eta^2 D^2, \frac{320}{\beta^2}, 240\right)$ and therefore:

$$c' \le 8\max\left(8|\mathcal{S}|^2\eta^2 D^2, \frac{320}{\beta}\right)$$

Thus $\sqrt{c'} = \max(8|\mathcal{S}|\eta D, \frac{31}{\beta}) = 8|\mathcal{S}|\eta D$ (the last equality holds because $\eta \ge \frac{4}{\beta}$) and therefore:

$$\xi_t = \min(\frac{8|\mathcal{S}|\eta D}{\sqrt{t}}, \beta) = \frac{8|\mathcal{S}|\eta D}{\sqrt{t}}$$

The last equality holds because $t \geq \frac{64|\mathcal{S}|^2\eta^2 D^2}{\beta}$.

Then the condition in Equation 35 of Lemma 19 is satisfies whenever:

$$\frac{n(t)}{\ln\ln(2n(t))} \geq \frac{120t \times 280\eta^2 D^2 \left(\ln\frac{100|\mathcal{S}||\mathcal{A}|t^2}{\delta}+1\right)}{\beta 64|\mathcal{S}|^2\eta^2 D^2} = \frac{525t\left(\ln\frac{100|\mathcal{S}||\mathcal{A}|t^2}{\delta}+1\right)}{\beta|\mathcal{S}|^2} \tag{38}$$

And therefore if we set $n(t) = \frac{525t\left(\ln\frac{100|\mathcal{S}||\mathcal{A}|t^2}{\delta}+1\right)^3}{\beta|\mathcal{S}|^2} \geq \frac{525t\ln\ln(2t)\left(\ln\frac{100|\mathcal{S}||\mathcal{A}|t^2}{\delta}+1\right)}{\beta|\mathcal{S}|^2}\ln(\frac{2t^2}{\delta})$ we see that with probability at least $1-3\delta$ and simultaneously for all $t \in \mathbb{N}$:

$$J_D\left(\frac{1}{t}\sum_{\ell=1}^{t}\mathbf{v}_\ell\right) \leq J_D(\mathbf{v}_\star) + \frac{36D}{\sqrt{t}}\max\left((|\mathcal{S}|+1)\eta D, \frac{18+16\sqrt{\ln\ln(2t)+\ln\frac{5.2}{\delta}}}{\beta}, 16\right)$$

$$= J_D(\mathbf{v}_\star) + \frac{72D^2|\mathcal{S}|\eta}{\sqrt{t}}\left(5 + 4\sqrt{\ln\ln(2t)+\ln\frac{5.2}{\delta}}\right)$$

The last inequality holds since $\eta \geq \frac{4}{\beta}$. This implies that using a budget of $n(t)$ samples where $n(t)$ satisfies Inequality 38 we can take $t$ gradient steps.

$\square$

# I  Extended Results for Tsallis Entropy Regularizers

For $\alpha > 1$ recall the Tsallis entropy between distributions $\mathbf{q}, \boldsymbol{\lambda}$ equals:

$$D_\alpha^{\mathcal{T}}(\boldsymbol{\lambda} \| \mathbf{q}) = \frac{1}{\alpha-1}\left(\mathbb{E}_{(s,a)\sim\mathbf{q}}\left[\left(\frac{\boldsymbol{\lambda}_{s,a}}{\mathbf{q}_{s,a}}\right)^\alpha - 1\right]\right)$$

$$= \frac{1}{\alpha-1}\left(\mathbb{E}_{(s,a)\sim\boldsymbol{\lambda}}\left[\left(\frac{\boldsymbol{\lambda}_{s,a}}{\mathbf{q}_{s,a}}\right)^{\alpha-1} - 1\right]\right)$$

Let $F(\boldsymbol{\lambda}) = \frac{1}{\eta}D_\alpha^{\mathcal{T}}(\boldsymbol{\lambda} \| \mathbf{q})$. The Fenchel Dual of a Tsallis Entropy satisfies:

$$F^*(\mathbf{u}) = \left\langle \boldsymbol{\lambda}(\mathbf{u}), \mathbf{u} - \frac{(\mathbf{u}+x_*\mathbf{1})}{\alpha}\boldsymbol{\lambda}(\mathbf{u})^{\alpha-1} + \frac{1}{\eta(\alpha-1)}\mathbf{1}\right\rangle$$

Where $\boldsymbol{\lambda}(\mathbf{u}) = (\eta\mathbf{u}+\eta x_*\mathbf{1})^{1/(\alpha-1)}\left(\frac{\alpha-1}{\alpha}\right)^{1/(\alpha-1)}\mathbf{q}$ and where $x_* \in \mathbb{R}$ such that $\sum_{s,a}\boldsymbol{\lambda}_{s,a}(\mathbf{u}) = 1$ and $\boldsymbol{\lambda}_{s,a}(\mathbf{u}) \geq 0$ for all $s,a \in \mathcal{S} \times \mathcal{A}$. This implies that:

$$J_D^{\mathcal{T},\alpha}(\mathbf{v}) = (1-\gamma)\sum_s \mathbf{v}_s\boldsymbol{\mu}_s + \left\langle \boldsymbol{\lambda}(\mathbf{A}^{\mathbf{v}}), \mathbf{A}^{\mathbf{v}} - \frac{(\mathbf{A}^{\mathbf{v}}+x_*\mathbf{1})}{\alpha}\boldsymbol{\lambda}(\mathbf{A}^{\mathbf{v}})^{\alpha-1} + \frac{1}{\eta(\alpha-1)}\mathbf{1}\right\rangle$$

### I.0.1  Strong Convexity of Tsallis Entropy

In this section we show that whenever $\alpha \in (1,2]$, the Tsallis entropy is a strongly convex function of $\boldsymbol{\lambda}$ in the $\|\cdot\|_2$ norm,

**Lemma 20.** *If $\alpha \in (1,2]$, the function $F(\boldsymbol{\lambda}) = \frac{1}{\eta}D_\alpha^{\mathcal{T}}(\boldsymbol{\lambda} \| \mathbf{q})$ is $\frac{\alpha}{\eta}$-strongly convex in the $\|\cdot\|_2$ norm.*

*Proof.* It is easy to see that $\nabla_{\boldsymbol{\lambda}}^2 D_\alpha^{\mathcal{T}}(\boldsymbol{\lambda} \| \mathbf{q})$ is a diagonal matrix satisfying:

$$\left[\nabla_{\boldsymbol{\lambda}}^2 D_\alpha^{\mathcal{T}}(\boldsymbol{\lambda} \| \mathbf{q})\right]_{s,a} = \frac{\alpha\boldsymbol{\lambda}_{s,a}^{\alpha-2}}{\eta\mathbf{q}_{s,a}^{\alpha-1}}.$$

Whenever $\alpha \leq 2$, and noting that $\mathbf{q} \in [0,1]$ we conclude that any of these terms must be lower bounded by $\frac{\alpha}{\eta}$. The result follows.

$\square$

## I.1 Tsallis entropy version of Lemma 4

**Lemma 21.** *Let $\tilde{\mathbf{v}} \in \mathbb{R}^{|\mathcal{S}|}$ be arbitrary and let $\tilde{\boldsymbol{\lambda}}$ be its corresponding candidate primal variable (i.e. $\tilde{\boldsymbol{\lambda}} = \boldsymbol{\lambda}(\mathbf{A}^{\mathbf{v}})$). If $\|\nabla_{\mathbf{v}} J_D(\tilde{\mathbf{v}})\|_1 \leq \epsilon$ and Assumptions 3 and 2 hold then whenever $|\mathcal{S}| \geq 2$:*

$$J_P^{\mathcal{T},\alpha}(\boldsymbol{\lambda}^{\tilde{\pi}}) \geq J_P^{\mathcal{T},\alpha}(\boldsymbol{\lambda}_\eta^*) - \epsilon \left( \frac{1+c}{1-\gamma} + \|\tilde{\mathbf{v}}\|_\infty \right)$$

*Where $c = \frac{1}{\eta(\alpha-1)} \frac{1}{\beta^{\alpha-1}} \left( \max(\alpha-1, \frac{2}{\rho^{\alpha-1}}) + 2 \right)$ and $\boldsymbol{\lambda}_\eta^\star$ is the $J_P$ optimum.*

*Proof.* For any $\boldsymbol{\lambda}$ and $\mathbf{v}$ let the lagrangian $J_L(\boldsymbol{\lambda}, \mathbf{v})$ be defined as,

$$J_L(\boldsymbol{\lambda}, \mathbf{v}) = (1-\gamma)\langle \boldsymbol{\mu}, \mathbf{v} \rangle + \left\langle \boldsymbol{\lambda}, \mathbf{A}^{\mathbf{v}} - \frac{1}{\eta(\alpha-1)} \left( \left( \frac{\boldsymbol{\lambda}}{\mathbf{q}} \right)^{\alpha-1} - 1 \right) \right\rangle$$

Note that $J_D(\tilde{\mathbf{v}}) = J_L(\tilde{\boldsymbol{\lambda}}, \tilde{\mathbf{v}})$ and that in fact $J_L$ is linear in $\bar{\mathbf{v}}$; *i.e.*,

$$J_L(\tilde{\boldsymbol{\lambda}}, \bar{\mathbf{v}}) = J_L(\tilde{\boldsymbol{\lambda}}, \tilde{\mathbf{v}}) + \langle \nabla_{\mathbf{v}} J_L(\tilde{\boldsymbol{\lambda}}, \tilde{\mathbf{v}}), \bar{\mathbf{v}} - \tilde{\mathbf{v}} \rangle.$$

Using Holder's inequality we have:

$$J_L(\tilde{\boldsymbol{\lambda}}, \bar{\mathbf{v}}) \geq J_L(\tilde{\boldsymbol{\lambda}}, \tilde{\mathbf{v}}) - \|\nabla_{\mathbf{v}} J_L(\tilde{\boldsymbol{\lambda}}, \tilde{\mathbf{v}})\|_1 \cdot \|\bar{\mathbf{v}} - \tilde{\mathbf{v}}\|_\infty = J_D(\tilde{\mathbf{v}}) - \|\nabla_{\mathbf{v}} J_L(\tilde{\boldsymbol{\lambda}}, \tilde{\mathbf{v}})\|_1 \cdot \|\bar{\mathbf{v}} - \tilde{\mathbf{v}}\|_\infty.$$

Let $\boldsymbol{\lambda}_\star$ be the candidate primal solution to the optimal dual solution $\mathbf{v}_\star = \arg\min_{\mathbf{v}} J_D(\mathbf{v})$. By weak duality we have that $J_D(\tilde{\mathbf{v}}) \geq J_P(\boldsymbol{\lambda}^\star) = J_D(\mathbf{v}_\star)$, and since by assumption $\|\nabla_{\mathbf{v}} J_L(\tilde{\boldsymbol{\lambda}}, \tilde{\mathbf{v}})\|_1 \leq \epsilon$:

$$J_L(\tilde{\boldsymbol{\lambda}}, \bar{\mathbf{v}}) \geq J_P(\boldsymbol{\lambda}^\star) - \epsilon\|\bar{\mathbf{v}} - \tilde{\mathbf{v}}\|_\infty. \tag{39}$$

In order to use this inequality to lower bound the value of $J_P(\boldsymbol{\lambda}^{\tilde{\pi}})$, we will need to choose an appropriate $\bar{\mathbf{v}}$ such that the LHS reduces to $J_P(\boldsymbol{\lambda}^{\tilde{\pi}})$ while keeping the $\ell_\infty$ norm on the RHS small. Thus we consider setting $\bar{\mathbf{v}}$ as:

$$\bar{\mathbf{v}}_s = \mathbb{E}_{a,s'\sim\tilde{\pi}\times\mathcal{T}} \left[ \mathbf{z}_s + \mathbf{r}_{s,a} - \frac{1}{\eta(\alpha-1)} \left( \left( \frac{\boldsymbol{\lambda}_{s,a}^{\tilde{\pi}}}{\mathbf{q}_{s,a}} \right)^{\alpha-1} - 1 \right) + \gamma\bar{\mathbf{v}}_{s'} \right]$$

Where $\mathbf{z} \in \mathbb{R}^{|\mathcal{S}|}$ is some function to be determined later. It is clear that an appropriate $\mathbf{z}$ exists as long as $\mathbf{z}, \mathbf{r}, \frac{1}{\eta(\alpha-1)} \left( \left( \frac{\boldsymbol{\lambda}_{s,a}^{\tilde{\pi}}}{\mathbf{q}_{s,a}} \right) - 1 \right)^{\alpha-1}$ are uniformly bounded. Furthermore:

$$\|\bar{\mathbf{v}}\|_\infty \leq \frac{\max_{s,a}\left| \mathbf{z}_s + \mathbf{r}_{s,a} - \frac{1}{\eta(\alpha-1)}\left( \left( \frac{\boldsymbol{\lambda}_{s,a}^{\tilde{\pi}}}{\mathbf{q}_{s,a}} \right)^{\alpha-1} - 1 \right) \right|}{1-\gamma} \leq \frac{\|\mathbf{z}\|_\infty + \|\mathbf{r}\|_\infty + \frac{1}{\eta(\alpha-1)}\left\| \left( \frac{\boldsymbol{\lambda}_{s,a}^{\tilde{\pi}}}{\mathbf{q}_{s,a}} \right)^{\alpha-1} - 1 \right\|_\infty}{1-\gamma} \tag{40}$$

We proceed to bound the norm of $\left\| \left( \frac{\boldsymbol{\lambda}_{s,a}^{\tilde{\pi}}}{\mathbf{q}_{s,a}} \right)^{\alpha-1} - 1 \right\|_\infty$. Observe that by Assumptions 2 and 3, for all states $s, a \in \mathcal{S} \times \mathcal{A}$, the ratio $|\frac{\boldsymbol{\lambda}_{s,a}^{\tilde{\pi}}}{\mathbf{q}_{s,a}}| \leq \frac{1}{\beta}$ and therefore:

$$\left\| \left( \frac{\boldsymbol{\lambda}_{s,a}^{\tilde{\pi}}}{\mathbf{q}_{s,a}} \right)^{\alpha-1} - 1 \right\|_\infty \leq 1 + \frac{1}{\beta^{\alpha-1}}$$

Notice the following relationships hold:

$$\left\langle \widetilde{\boldsymbol{\lambda}}, \mathbf{A}^{\bar{\mathbf{v}}} - \frac{1}{\eta(\alpha-1)}\left(\left(\frac{\widetilde{\boldsymbol{\lambda}}}{\mathbf{q}}\right)^{\alpha-1} - 1\right)\right\rangle = \sum_s \widetilde{\boldsymbol{\lambda}}_s \left(\mathbb{E}_{a,s'\sim\widetilde{\pi}\times\mathbf{P}}\left[\mathbf{r}_{s,a} + \gamma\bar{\mathbf{v}}_{s'} - \bar{\mathbf{v}}_s - \frac{1}{\eta(\alpha-1)}\left(\left(\frac{\widetilde{\boldsymbol{\lambda}}_{s,a}}{\mathbf{q}_{s,a}}\right)^{\alpha-1} - 1\right)\right]\right)$$

$$= \sum_s \widetilde{\boldsymbol{\lambda}}_s \left(\mathbb{E}_{a,s'\sim\widetilde{\pi}\times\mathbf{P}}\left[\frac{1}{\eta(\alpha-1)}\left(\left(\frac{\boldsymbol{\lambda}^{\widetilde{\pi}}_{s,a}}{\mathbf{q}_{s,a}}\right)^{\alpha-1} - 1\right) - \frac{1}{\eta(\alpha-1)}\left(\left(\frac{\widetilde{\boldsymbol{\lambda}}_{s,a}}{\mathbf{q}_{s,a}}\right)^{\alpha-1} - 1\right) - \right.$$

$$= \sum_s \widetilde{\boldsymbol{\lambda}}_s \left(\mathbb{E}_{a,s'\sim\widetilde{\pi}\times\mathbf{P}}\left[\frac{1}{\eta(\alpha-1)}\left(\frac{\boldsymbol{\lambda}^{\widetilde{\pi}}_{s,a}}{\mathbf{q}_{s,a}}\right)^{\alpha-1} - \frac{1}{\eta(\alpha-1)}\left(\frac{\widetilde{\boldsymbol{\lambda}}_{s,a}}{\mathbf{q}_{s,a}}\right)^{\alpha-1} - \mathbf{z}_s\right]\right)$$

$$= \sum_s \widetilde{\boldsymbol{\lambda}}_s \left(\frac{1}{\eta(\alpha-1)}\left(\left(\frac{\boldsymbol{\lambda}^{\widetilde{\pi}}_s}{\mathbf{q}_s}\right)^{\alpha-1} - \left(\frac{\widetilde{\boldsymbol{\lambda}}_s}{\mathbf{q}_s}\right)^{\alpha-1}\right)\left[\sum_a \frac{\widetilde{\pi}^\alpha(a|s)}{\mathbf{q}_{a|s}^{\alpha-1}}\right] - \mathbf{z}_s\right)$$

$$\tag{41}$$

Where $\widetilde{\boldsymbol{\lambda}}_s = \sum_a \widetilde{\boldsymbol{\lambda}}_{s,a}$ and $\boldsymbol{\lambda}^{\widetilde{\pi}}_s = \sum_a \boldsymbol{\lambda}^{\widetilde{\pi}}_{s,a}$. Note that by definition:

$$(1-\gamma)\langle\boldsymbol{\mu},\bar{\mathbf{v}}\rangle = \left\langle\boldsymbol{\lambda}^{\widetilde{\pi}}, \mathbf{z} + \mathbf{r} - \frac{1}{\eta(\alpha-1)}\left(\left(\frac{\boldsymbol{\lambda}^{\widetilde{\pi}}}{\mathbf{q}}\right)^{\alpha-1} - 1\right)\right\rangle = J_P(\boldsymbol{\lambda}^{\widetilde{\pi}}) + \langle\boldsymbol{\lambda}^{\widetilde{\pi}}, \mathbf{z}\rangle. \tag{42}$$

Let's expand the definition of $J_L(\widetilde{\boldsymbol{\lambda}}, \bar{\mathbf{v}})$ using Equations 11 and 12:

$$J_L(\widetilde{\boldsymbol{\lambda}}, \bar{\mathbf{v}}) = (1-\gamma)\langle\boldsymbol{\mu}, \bar{\mathbf{v}}\rangle + \left\langle\widetilde{\boldsymbol{\lambda}}, \mathbf{A}^{\bar{\mathbf{v}}} - \frac{1}{\eta(\alpha-1)}\left(\left(\frac{\widetilde{\boldsymbol{\lambda}}}{\mathbf{q}}\right)^{\alpha-1} - 1\right)\right\rangle$$

$$= J_P(\boldsymbol{\lambda}^{\widetilde{\pi}}) + \langle\boldsymbol{\lambda}^{\widetilde{\pi}}, \mathbf{z}\rangle + \sum_s \widetilde{\boldsymbol{\lambda}}_s\left(\frac{1}{\eta(\alpha-1)}\left(\left(\frac{\boldsymbol{\lambda}^{\widetilde{\pi}}_s}{\mathbf{q}_s}\right)^{\alpha-1} - \left(\frac{\widetilde{\boldsymbol{\lambda}}_s}{\mathbf{q}_s}\right)^{\alpha-1}\right)\left[\sum_a \frac{\widetilde{\pi}^\alpha(a|s)}{\mathbf{q}_{a|s}^{\alpha-1}}\right] - \mathbf{z}_s\right)$$

$$= J_P(\boldsymbol{\lambda}^{\widetilde{\pi}}) + \sum_s\left(\mathbf{z}_s(\boldsymbol{\lambda}^{\widetilde{\pi}}_s - \widetilde{\boldsymbol{\lambda}}_s) + \frac{\widetilde{\boldsymbol{\lambda}}_s}{\eta(\alpha-1)}\left(\left(\frac{\boldsymbol{\lambda}^{\widetilde{\pi}}_s}{\mathbf{q}_s}\right)^{\alpha-1} - \left(\frac{\widetilde{\boldsymbol{\lambda}}_s}{\mathbf{q}_s}\right)^{\alpha-1}\right)\left[\sum_a \frac{\widetilde{\pi}^\alpha(a|s)}{\mathbf{q}_{a|s}^{\alpha-1}}\right]\right)$$

Since we want this expression to equal $J_P(\boldsymbol{\lambda}^{\widetilde{\pi}})$, we need to choose $\mathbf{z}$ such that:

$$\mathbf{z}_s = \frac{\frac{1}{\eta(\alpha-1)}\left(\left(\frac{\boldsymbol{\lambda}^{\widetilde{\pi}}_s}{\mathbf{q}_s}\right)^{\alpha-1} - \left(\frac{\widetilde{\boldsymbol{\lambda}}_s}{\mathbf{q}_s}\right)^{\alpha-1}\right)\left[\sum_a \frac{\widetilde{\pi}^\alpha(a|s)}{\mathbf{q}_{a|s}^{\alpha-1}}\right]}{1 - \frac{\boldsymbol{\lambda}^{\widetilde{\pi}}_s}{\widetilde{\boldsymbol{\lambda}}_s}}$$

Observe that $\mathbf{z}_s = \frac{\frac{1}{\eta(\alpha-1)}\left((\boldsymbol{\lambda}^{\widetilde{\pi}}_s)^{\alpha-1} - (\widetilde{\boldsymbol{\lambda}}_s)^{\alpha-1}\right)\left[\sum_a \frac{\widetilde{\pi}^\alpha(a|s)}{\mathbf{q}_{s,a}^{\alpha-1}}\right]}{1 - \frac{\boldsymbol{\lambda}^{\widetilde{\pi}}_s}{\widetilde{\boldsymbol{\lambda}}_s}}$ and therefore, since for all $s$ and when

$\alpha \geq 1$ by Assumption 2 we have that $\sum_a \frac{\widetilde{\pi}^\alpha(a|s)}{\mathbf{q}_{s,a}^{\alpha-1}} \leq \frac{1}{\beta^{\alpha-1}}$,

$$|\mathbf{z}_s| \leq \frac{1}{\eta(\alpha-1)}\frac{1}{\beta^{\alpha-1}}\frac{\left|(\boldsymbol{\lambda}^{\widetilde{\pi}}_s)^{\alpha-1} - \widetilde{\boldsymbol{\lambda}}_s^{\alpha-1}\right|}{\left|1 - \frac{\boldsymbol{\lambda}^{\widetilde{\pi}}_s}{\widetilde{\boldsymbol{\lambda}}_s}\right|}$$

Let $\frac{\boldsymbol{\lambda}^{\widetilde{\pi}}_s}{\widetilde{\boldsymbol{\lambda}}_s} = \frac{1}{\theta}$ where $\theta \in [0, \frac{1}{\rho}]$. Then,

$$|\mathbf{z}_s| \leq \frac{1}{\eta(\alpha-1)\beta^{\alpha-1}}\boldsymbol{\lambda}^{\widetilde{\pi}}_s\frac{|1 - \theta^{\alpha-1}|}{|1 - \frac{1}{\theta}|}$$

It is easy to see that when $\alpha \geq 0$ the function $f(\theta) = \frac{1-\theta^{\alpha-1}}{1-\frac{1}{\theta}} = \frac{\theta-\theta^\alpha}{\theta-1}$ is decreasing in the interval $(0,1]$ and increasing afterwards. Furthermore, by L'Hopital's rule, $f(1) = 1 - \alpha$ and $f(\frac{1}{\rho}) = \frac{\frac{1}{\rho^\alpha}-\frac{1}{\rho}}{\frac{1}{\rho}-1} \leq \frac{2}{\rho^{\alpha-1}}$ since $\rho \leq \frac{1}{2}$. This implies,

$$|\mathbf{z}_s| \leq \frac{1}{\eta(\alpha-1)} \frac{1}{\beta^{\alpha-1}} \max(\alpha-1, \frac{2}{\rho^{\alpha-1}}).$$

And therefore Equation 40 implies:

$$\|\bar{\mathbf{v}}\|_\infty \leq \frac{\frac{1}{\eta(\alpha-1)}\frac{1}{\beta^{\alpha-1}}\max(\alpha-1,\frac{2}{\rho^{\alpha-1}}) + 1 + \frac{1}{\eta(\alpha-1)}\left(\frac{1}{\beta^{\alpha-1}}+1\right)}{1-\gamma} = \frac{\frac{1}{\eta(\alpha-1)}\frac{1}{\beta^{\alpha-1}}\left(\max(\alpha-1,\frac{2}{\rho^{\alpha-1}})+2\right)+1}{1-\gamma}$$

Putting these together we obtain the following version of equation 39:

$$J_L(\widetilde{\boldsymbol{\lambda}}, \bar{\mathbf{v}}) \geq J_P(\boldsymbol{\lambda}^\star) - \epsilon \left( \frac{\frac{1}{\eta(\alpha-1)}\frac{1}{\beta^{\alpha-1}}\left(\max(\alpha-1,\frac{2}{\rho^{\alpha-1}})+2\right)+1}{1-\gamma} + \|\widetilde{\mathbf{v}}\|_\infty \right)$$

$\square$

## I.2 Extension of Lemma 5 to Tsallis Entropy

**Lemma 22.** *Under Assumptions 1, 2 and 3, the optimal dual variables are bounded as*

$$\|\mathbf{v}^*\|_\infty \leq \frac{1}{1-\gamma}\left(1 + \frac{2}{\eta(\alpha-1)\beta^{\alpha-1}}\right) = D_{\mathcal{D},\alpha}. \tag{43}$$

*Proof.* Recall the Lagrangian form,

$$\min_{\mathbf{v}}, \max_{\boldsymbol{\lambda}_{s,a} \in \Delta_{S \times A}} J_L(\boldsymbol{\lambda}, \mathbf{v}) := (1-\gamma)\langle \mathbf{v}, \boldsymbol{\mu} \rangle + \left\langle \boldsymbol{\lambda}, \mathbf{A}^{\mathbf{v}} - \frac{1}{\eta(\alpha-1)}\left( \left(\frac{\boldsymbol{\lambda}_{s,a}}{\mathbf{q}_{s,a}}\right)^{\alpha-1} - 1 \right) \right\rangle.$$

The KKT conditions of $\boldsymbol{\lambda}^*, \mathbf{v}^*$ imply that for any $s, a$, either (1) $\boldsymbol{\lambda}_{s,a}^* = 0$ and $\frac{\partial}{\partial \boldsymbol{\lambda}_{s,a}} J_L(\boldsymbol{\lambda}^*, v^*) \leq 0$ or (2) $\frac{\partial}{\partial \boldsymbol{\lambda}_{s,a}} J_L(\boldsymbol{\lambda}^*, \mathbf{v}^*) = 0$. The partial derivative of $J_L$ is given by,

$$\frac{\partial}{\partial \boldsymbol{\lambda}_{s,a}} J_L(\boldsymbol{\lambda}^*, \mathbf{v}^*) = \mathbf{r}_{s,a} + \gamma \sum_{s'} P_a(s'|s)\mathbf{v}_{s'}^* - \mathbf{v}_s^* - \frac{\alpha}{\eta(\alpha-1)}\left(\frac{\boldsymbol{\lambda}_{s,a}^*}{\mathbf{q}_{s,a}}\right)^{\alpha-1} + \frac{1}{\eta(\alpha-1)}. \tag{44}$$

Thus, for any $s, a$, either

$$\boldsymbol{\lambda}_{s,a}^* = 0 \text{ and } \mathbf{v}_s^* \geq \mathbf{r}_{s,a} - \frac{\alpha}{\eta(\alpha-1)}\left(\frac{\boldsymbol{\lambda}_{s,a}^*}{\mathbf{q}_{s,a}}\right)^{\alpha-1} + \frac{1}{\eta(\alpha-1)} + \gamma \sum_{s'} P_a(s'|s)\mathbf{v}_{s'}^*, \tag{45}$$

or,

$$\boldsymbol{\lambda}_{s,a}^* > 0 \text{ and } \mathbf{v}_s^* = \mathbf{r}_{s,a} - \frac{\alpha}{\eta(\alpha-1)}\left(\frac{\boldsymbol{\lambda}_{s,a}^*}{\mathbf{q}_{s,a}}\right)^{\alpha-1} + \frac{1}{\eta(\alpha-1)} + \gamma \sum_{s'} P_a(s'|s)\mathbf{v}_{s'}^*. \tag{46}$$

Recall that $\boldsymbol{\lambda}^*$ is the discounted state-action visitations of some policy $\pi_\star$; *i.e.*, $\boldsymbol{\lambda}_{s,a}^* = \boldsymbol{\lambda}_s^{\pi_\star} \cdot \pi_\star(a|s)$ for some $\pi_\star$. Note that by Assumption 3, any policy $\pi$ has $\boldsymbol{\lambda}_s^{\pi_\star} > 0$ for all $s$. Accordingly, the KKT conditions imply,

$$\pi_\star(a|s) = 0 \text{ and } \mathbf{v}_s^* \geq \mathbf{r}_{s,a} - \frac{\alpha}{\eta(\alpha-1)}\left(\frac{\boldsymbol{\lambda}_{s,a}^*}{\mathbf{q}_{s,a}}\right)^{\alpha-1} + \frac{1}{\eta(\alpha-1)} + \gamma \sum_{s'} P_a(s'|s)\mathbf{v}_{s'}^*, \tag{47}$$

or,

$$\pi_\star(a|s) > 0 \text{ and } \mathbf{v}_s^* = \mathbf{r}_{s,a} - \frac{\alpha}{\eta(\alpha-1)}\left(\frac{\boldsymbol{\lambda}_{s,a}^*}{\mathbf{q}_{s,a}}\right)^{\alpha-1} + \frac{1}{\eta(\alpha-1)} + \gamma \sum_{s'} P_a(s'|s)\mathbf{v}_{s'}^*. \tag{48}$$

Equivalently,

$$\mathbf{v}_s^* = \mathbb{E}_{a \sim \pi_\star(s)} \left[ \mathbf{r}_{s,a} - \frac{\alpha}{\eta(\alpha - 1)} \left( \frac{\boldsymbol{\lambda}_{s,a}^*}{\mathbf{q}_{s,a}} \right)^{\alpha - 1} + \frac{1}{\eta(\alpha - 1)} + \gamma \sum_{s'} P_a(s'|s) \mathbf{v}_{s'}^* \right] \qquad (49)$$

$$(50)$$

We may express these conditions as a Bellman recurrence for $\mathbf{v}_s^*$ and the solution to these Bellman equations is bounded when $\mathbf{r}_{s,a} - \frac{\alpha}{\eta(\alpha-1)} \left( \frac{\boldsymbol{\lambda}_{s,a}^*}{\mathbf{q}_{s,a}} \right)^{\alpha - 1} + \frac{1}{\eta(\alpha-1)}$ is bounded [24]. And indeed, by Assumptions 2 and 1, $\left| \mathbf{r}_{s,a} - \frac{\alpha}{\eta(\alpha-1)} \left( \frac{\boldsymbol{\lambda}_{s,a}^*}{\mathbf{q}_{s,a}} \right)^{\alpha - 1} + \frac{1}{\eta(\alpha-1)} \right| \leq 1 + \frac{1}{\eta(\alpha-1)} + \frac{1}{\eta(\alpha-1)\beta^{\alpha-1}}$ We may thus bound the solution as,

$$\|\mathbf{v}^*\|_\infty \leq \frac{1}{1 - \gamma} \left( 1 + \frac{2}{\eta(\alpha - 1)\beta^{\alpha-1}} \right). \qquad (51)$$

$\square$

## I.3   Gradient descent results for the Tsallis Entropy

**Remark 1.** *Throughout this section we make the assumption that $\alpha \in (1, 2]$.*

We start by characterizing the smoothness properties of $J_D^{\mathcal{T}, \alpha}(\mathbf{v})$, the dual function of the Tsallis regularized LP.

**Lemma 23.** *If $\alpha \in (1, 2]$ the dual function $J_D^{\mathcal{T}, \alpha}(\mathbf{v})$ is $\frac{\eta|\mathcal{S}||\mathcal{A}|}{\alpha}$-smooth in the $\| \cdot \|_2$ norm.*

*Proof.* Recall that PrimalReg-$\boldsymbol{\lambda}$ can be written as RegLP:

$$\max_{\boldsymbol{\lambda} \in \mathcal{D}} \langle \mathbf{r}, \boldsymbol{\lambda} \rangle - F(\boldsymbol{\lambda})$$
$$\text{s.t. } \mathbf{E}\boldsymbol{\lambda} = b.$$

Where the regularizer ($F(\boldsymbol{\lambda}) := \frac{1}{\eta} D_\alpha^{\mathcal{T}}(\boldsymbol{\lambda} \| \mathbf{q})$) is $\frac{\alpha}{\eta} - \| \cdot \|_2$ strongly convex. In this problem $\mathbf{r}$ corresponds to the reward vector, the vector $\mathbf{b} = (1 - \gamma)\boldsymbol{\mu} \in \mathbb{R}^{|\mathcal{S}|}$ and matrix $\mathbf{E} \in \mathbb{R}^{|\mathcal{S}| \times |\mathcal{S}| \times |\mathcal{A}|}$ takes the form:

$$\mathbf{E}[s, s', a] = \begin{cases} \gamma \mathbf{P}_a(s|s') & \text{if } s \neq s' \\ 1 - \gamma \mathbf{P}_a(s|s) & \text{o.w.} \end{cases}$$

Therefore (since $\|\mathbf{E}\|_{2,2}$ is simply the Frobenius norm of matrix $\mathbf{E}$),

$$\|\mathbf{E}\|_{2,2} \leq 2|\mathcal{S}||\mathcal{A}|$$

The result follows as a corollary of Lemma 1. $\square$

Throughout this section we use the notation $D_{\mathcal{T},\alpha}$ to refer to $\|\mathbf{v}^*\|_\infty \leq \frac{1}{1-\gamma} \left( 1 + \frac{2}{\eta(\alpha-1)\beta^{\alpha-1}} \right)$. We are ready to prove convergence guarantees for Algorithm 4 when applied to the objective $J_D^{\mathcal{T}, \alpha}$.

**Lemma 24.** *Let Assumptions 1, 2 and 3 hold. Let $\mathcal{D}_{\mathcal{T},\alpha} = \{\mathbf{v} \text{ s.t. } \|\mathbf{v}\|_\infty \leq D_{\mathcal{T},\alpha}\}$, and define the distance generating function to be $w(\mathbf{x}) = \|\mathbf{x}\|_2^2$. After $T$ steps of Algorithm 4, the objective function $J_D^{\mathcal{T}, \alpha}$ evaluated at the iterate $\mathbf{v}_T = y_T$ satisfies:*

$$J_D^{\mathcal{T}, \alpha}(\mathbf{v}_T) - J_D^{\mathcal{T}, \alpha}(\mathbf{v}^*) \leq 4\eta \frac{|\mathcal{S}|^2 |\mathcal{A}|}{\alpha} \frac{(1 + c')^2}{(1 - \gamma)^2 T^2}.$$

*Where $c' = \frac{2}{\eta(\alpha-1)\beta^{\alpha-1}}$.*

*Proof.* This results follows simply by invoking the guarantees of Theorem 1, making use of the fact that $J_D^{\mathcal{T}, \alpha}$ is $\frac{\eta|\mathcal{S}||\mathcal{A}|}{\alpha}$−smooth as proven by Lemma 3, observing that as a consequence of Lemma 22, $\mathbf{v}^\star \in \mathcal{D}_{\mathcal{T},\alpha}$ and using the inequality $\|\mathbf{x}\|_2^2 \leq |\mathcal{S}| \|\mathbf{x}\|_\infty^2$ for $\mathbf{x} \in \mathbb{R}^{|\mathcal{S}|}$. $\square$

Lemma 24 can be easily turned into the following guarantee regarding the dual function value of the final iterate:

**Corollary 6.** *Let $\epsilon > 0$. If Algorithm 4 is ran for at least $T$ rounds*

$$T \geq 2\eta^{1/2}(|\mathcal{S}||\mathcal{A}|^{1/2})\frac{(1+c')}{\alpha^{1/2}(1-\gamma)\sqrt{\epsilon}}$$

*then $\mathbf{v}_T$ is an $\epsilon-$optimal solution for the dual objective $J_D^{T,\alpha}$.*

If $T$ satisfies the conditions of Corollary 6 a simple use of Lemma 6 allows us to bound the $\|\cdot\|_2$ norm of the dual function's gradient at $\mathbf{v}_T$:

$$\|\nabla J_D(\mathbf{v}_T)\|_2 \leq \sqrt{\frac{2|\mathcal{S}||\mathcal{A}|\eta\epsilon}{\alpha}}$$

If we denote as $\pi_T$ to be the policy induced by $\boldsymbol{\lambda}^{\mathbf{v}_T}$, and $\boldsymbol{\lambda}_\eta^\star$ is the candidate dual solution corresponding to $\mathbf{v}^\star$. A simple application of Lemma 21 yields:

$$J_P(\boldsymbol{\lambda}^{\pi_T}) \geq J_P(\boldsymbol{\lambda}_\eta^\star) - \frac{1}{1-\gamma}(2+c+c')\sqrt{\frac{2|\mathcal{S}||\mathcal{A}|\eta\epsilon}{\alpha}}$$

Where $c = \frac{1}{\eta(\alpha-1)}\frac{1}{\beta^{\alpha-1}}\left(\max(\alpha-1, \frac{2}{\rho^{\alpha-1}}) + 2\right)$, $c' = \frac{2}{\eta(\alpha-1)\beta^{\alpha-1}}$ and $\boldsymbol{\lambda}_\eta^\star$ is the $J_P$ optimum.

This leads us to the main result of this section:

**Corollary 7.** *Let $\alpha \in (1, d]$. For any $\xi > 0$. If $T \geq 4\eta|\mathcal{S}|^{3/2}|\mathcal{A}|^{1/2}\frac{(2+c+c')^2}{\alpha(1-\gamma)^2\xi}$ then:*

$$J_P(\boldsymbol{\lambda}^{\pi_T}) \geq J_P(\boldsymbol{\lambda}_\eta^\star) - \xi.$$

Thus Algorithm 4 achieves an $\mathcal{O}(1/(1-\gamma)^2\epsilon)$ rate of convergence to an $\epsilon-$optimal regularized policy. We now proceed to show that an appropriate choice for $\eta$ can be leveraged to obtain an $\epsilon-$optimal policy.

**Theorem 5.** *For any $\epsilon > 0$, let $\eta = \frac{2}{(\alpha-1)\epsilon\beta^\alpha}$. If $T \geq 8|\mathcal{S}|^{3/2}|\mathcal{A}|^{1/2}\frac{(2+c+c')^2}{(\alpha-1)\alpha(1-\gamma)^2\beta^\alpha\epsilon^2}$, then $\pi_T$ is an $\epsilon-$optimal policy.*

*Proof.* As a consequence of Corollary 7, we can conclude that:

$$J_P(\boldsymbol{\lambda}^{\pi_T}) \geq J_P(\boldsymbol{\lambda}^{\star,\eta}) - \frac{\epsilon}{2}.$$

Where $\boldsymbol{\lambda}_\eta^\star$ is the regularized optimum. Recall that:

$$J_P(\boldsymbol{\lambda}) = \sum_{s,a}\boldsymbol{\lambda}_{s,a}\mathbf{r}_{s,a} - \frac{1}{(\alpha-1)\eta}\left(\mathbb{E}_{(s,a)\sim\mathbf{q}}\left[\left(\frac{\boldsymbol{\lambda}_{s,a}}{\mathbf{q}_{s,a}}\right)^\alpha - 1\right]\right).$$

Since $\boldsymbol{\lambda}^{\star,\eta}$ is the maximizer of the regularized objective, it satisfies $J_P(\boldsymbol{\lambda}^{\star,\eta}) \geq J_P(\boldsymbol{\lambda}^*)$ where $\boldsymbol{\lambda}^*$ is the visitation frequency of the optimal policy corresponding to the unregularized objective. We can conclude that:

$$\sum_{s,a}\boldsymbol{\lambda}_{s,a}^{\pi_T}\mathbf{r}_{s,a} \geq \sum_{s,a}\boldsymbol{\lambda}_{s,a}^\star\mathbf{r}_{s,a} + \frac{1}{(\alpha-1)\eta}\left(\sum_{s,a}\mathbf{q}_{s,a}\left(\left(\frac{\boldsymbol{\lambda}_{s,a}^{\pi_T}}{\mathbf{q}_{s,a}}\right)^\alpha - 1\right) - \sum_{s,a}\mathbf{q}_{s,a}\left(\left(\frac{\boldsymbol{\lambda}_{s,a}^\star}{\mathbf{q}_{s,a}}\right)^\alpha - 1\right)\right) - \frac{\epsilon}{2}$$

$$= \sum_{s,a}\boldsymbol{\lambda}_{s,a}^\star\mathbf{r}_{s,a} + \frac{1}{(\alpha-1)\eta}\left(\sum_{s,a}\mathbf{q}_{s,a}\left(\frac{\boldsymbol{\lambda}_{s,a}^{\pi_T}}{\mathbf{q}_{s,a}}\right)^\alpha - \sum_{s,a}\mathbf{q}_{s,a}\left(\frac{\boldsymbol{\lambda}_{s,a}^\star}{\mathbf{q}_{s,a}}\right)^\alpha\right) - \frac{\epsilon}{2}$$

$$\geq \sum_{s,a}\boldsymbol{\lambda}_{s,a}^\star\mathbf{r}_{s,a} - \frac{1}{(\alpha-1)\eta}\left(\frac{1}{\beta}\right)^\alpha - \frac{\epsilon}{2}$$

And therefore if $\eta = \frac{2}{(\alpha-1)\epsilon\beta^\alpha}$, we can conclude that:

$$\sum_{s,a}\boldsymbol{\lambda}_{s,a}^{\pi_T}\mathbf{r}_{s,a} \geq \sum_{s,a}\boldsymbol{\lambda}_{s,a}^\star\mathbf{r}_{s,a} - \epsilon.$$

$\square$