# OpenReview forum: "Near Optimal Policy Optimization via REPS"
_NeurIPS.cc/2021/Conference — NeurIPS 2021 Poster_

### Official Review · Reviewer_xopv · 2021-07-15

**Rating:** 6
**Confidence:** 5

**Summary:**

In this paper the authors the sample complexity of the relative entropy policy search (REPS). This algorithm aims at finding the optimal policy in an MDP using linear programming formulation. First the authors consider the planning setting where they assume access to the exact model of the environment. For this result they show that near stationary policies result in a near optimal policy.
Next they provide the convergence when the updates are being performed by biased stochastic gradients.

**Limitations And Societal Impact:**

- The result of this paper is only working for tabular setting.
- The sample complexity provided for relative entropy policy search in this paper is significantly worse than the state of the art sample complexity for actor critic and Q-learning type of algorithms which is O(1/epsilon^2).
- Need for iid sampling of the data.

**Main Review:**

- First of all, the setting of the paper is tabular, which is a limited setting for RL applications. Furthermore, the sample complexity is worse than the state of the art sample complexity in the literature for actor-critic and Q-learning methods. These two reasons puts the benefit of the results in the paper in jeopardy.
- Secondly, in the writing of the paper there are multiple definitions and lemmas (such as definition 6,7 or Theorem 1) which are well known in the literature. If the authors remove these definitions from the main body of the paper and just mention them briefly, it would save more space. As a suggestion, stating Theorem 3 somewhere earlier in the paper would make the presentation more clear.
- The sampling scheme in Algorithm 2 is completely iid. This is indeed another limitation of this paper compared to the related work. In addition, is there a way to obtain unbiased estimate of the gradient? Since the data sampling is iid, I suspect it should be possible.
typo: line 240: x_0-x_*

****After rebuttal comment: After reading through the responses of the authors, I decided to increase my score to 6. I believe this is a good paper, and deserve presentation in Neurips.

**Time Spent Reviewing:**

6

---

> ### Author Response · Authors · 2021-08-10
> **Response to Reviewer 4**
>
> We thank the reviewer for their comments. As the reviewer has noted, the objective of our paper is to prove convergence guarantees for Relative Entropy Search performed only in the dual objective of the regularized linear programming RL formulation.  Since the dual objective is convex, a variety of existing results can be leveraged to obtain provable guarantees when the model is known. When the model is not known, more effort needs to be put into coming up with ways for estimating the gradients. Since there is no readily available way to produce true unbiased stochastic gradients enjoying a small enough variance, we had to use a batch sampling approach. Our main contributions are:
>
> Lemma 4. We show that a candidate dual variable having small gradient gives rise to a policy whose true visitation distribution has large primal value $J_P$. This result is quite involved and by all means not obvious. It requires a deep understanding of bellman optimality. This is not an immediate corollary from existing results of convex optimization because the true value of the extracted policy corresponding to a candidate dual solution is not the same as the dual function evaluated at that candidate solution. Taking into account the extra “projection” step forms the basis of the proof of this Lemma. This result is not present in the existing literature and we consider it one of our main results.  We will make sure this is more prominently stated in our final manuscript.
>
> Lemma 5. We show a sharp upper bound on the norm of $\| \mathbf{v}^{\star} \|_\infty$. Our bound has the advantage of depending logarithmically on $|S|$, $|A|$, $\beta$ and $\rho$. This result is not present in the existing literature and we consider it one of our main results. We will make sure this is more prominently stated in our final manuscript.
>
> Theorems 2 and 3. We make use of a series of existing and new techniques in conjunction with the results of section 5.1 (particularly Lemma 5) to obtain these results. The analysis of stochastic gradient descent with biased and stochastic gradients for smooth convex functions is our own and it may be of independent interest (Lemma 9).
>
> We address the reviewer’s comments below.
>
> “First of all, the setting of the paper is tabular, which is a limited setting for RL applications” We think the tabular domain is a good first laboratory for proving new RL results. This has been true since the inception of RL as a field, it was true when UCRL and UCBVI were introduced and it is true in this first analysis of the REPS algorithm with stochastic gradients. Analyzing REPS under any of the existing models of function approximation in RL such as Linear MDPs, Mixture MDPs or others wouldn’t be particularly different but would only serve to obfuscate the results and the discussion. We prefer cleaner, leaner and simpler results without extra unnecessary ornamentation.
>
> “Furthermore, the sample complexity is worse than the state of the art sample complexity in the literature for actor-critic and Q-learning methods” We are afraid this comment makes no sense. This is not a fair comparison. The objective of our work is to obtain the first provable guarantees for the REPS algorithm. We believe our contribution should be judged with respect to the relative difficulties of achieving this and not with respect to other algorithmic approaches in the literature. We think it isn’t a prerequisite for a theoretical paper to have any value that it presents a rate that beats all other rates for the same problem, particularly when (as it is the case for REPS) the algorithm has not yet been proven to work in the finite sample regime. The objective of theoretical RL research should not be measured by the yardstick of rate chasing as it is sadly done in empirical Deep RL research where benchmark beating is the norm, but by the inherent value of the problem and the solution provided instead. We also think this manner of blanket rejection mentality lies at the heart of what is rotten in the conference reviewing process, where submissions are judged by the easiest and most convenient superficial metrics and not by their actual value.
>
> “Secondly, in the writing of the paper there are multiple definitions and lemmas (such as definition 6,7 or Theorem 1) … clear” We will make changes in the final version of our manuscript to improve its readability. Thanks a lot for the suggestions.
>
> “The sampling scheme in Algorithm 2 is completely iid.” The sampling scheme doesn’t need to be i.i.d. As a result of assumption 3, this can be substituted by a more complicated on-policy sampling scheme where the current policy is mixed with a small epsilon-greedy exploration policy. We chose not to present this sampling scheme as it would obfuscate the main presentation of the algorithm. We can add that discussion in the camera ready version.
>
> “In addition, is there a way to obtain unbiased estimate of the gradient? Since the data sampling is iid, I suspect it should be possible” We spent a lot of time trying to figure this out. We did come up with an unbiased version of the gradient making use of random expansions of the Taylor series of the exponential function. Unfortunately this approach suffers from a huge variance and is therefore unable to yield the high probability convergence rates that our biased gradients algorithm achieves. We can add a discussion about this in the Appendix of the camera ready version. We invite the reviewer to really think about how would they try obtaining an unbiased estimator of $exp(E[x])$ given only samples of $x$ as this would be the first step necessary to derive a truly unbiased estimator of the gradient of J_D(v).
>
> “The result of this paper is only working for tabular setting” We do not think this is a fair criticism. The paper introducing UCBVI (Minimax Regret Bounds for Reinforcement Learning, Mohammad Azar, Ian Osband, Remi Munos) also did so in the tabular setting first. This in no way detracted from the validity and importance of their results.
>
> “The sample complexity provided for relative entropy policy search in this paper is significantly worse .... ” We don’t think this is a valid criticism. These results are not comparable.
> “I.i.d. Sampling of the data” We have added a discussion about this above.

---

### Official Review · Reviewer_3BWW · 2021-07-15

**Rating:** 8
**Confidence:** 3

**Summary:**

This paper studies the convergence rate of (tabular) Relative Entropy Policy Search (REPS), both assuming the knowledge of the transition matrix $P$ and then relaxing this assumption.

**Limitations And Societal Impact:**

The authors present the limitations of their submission. They do not discuss the societal impact, as there is no need.

**Main Review:**

The paper starts by identifying two main issues of REPS.

The first issue is that convex duality asserts a relationship at the exact optimum of the primal or dual variables. However, REPS does not fully optimize the dual, and therefore it is not clear whether a near-optimal solution of the dua results in a near-optimal solution of the primal.
Furthermore, the dual problem of REPS identifies the minimization of (an entropic mapping of) the advantage; but this requires the knowledge of the transition matrix $P$. One way to overcome the problem is to rely on samples, at the cost of the introduction of a biased solution.

The paper starts with a gentle introduction of a linear program in Section~4, and then introduces a regularized version where a convex regularization $F$ is added to the linear objective. The convex constraint is usually a $f$-divergence (such as the Kullback-Leibler).

At this point, the authors notice that if $F$ is strongly convex, then its Fenchel conjugate must be $1/\beta$-smooth. This helps to characterize the smoothness of the dual function for REPS.

In Section 5.1, the authors provide a bound on the optimality of the solution. Building on these results and assuming Accelerated Gradient Descent as used optimization technique, the authors provide convergence rates in Section 5.2., assuming known $P$. In Section~6, the authors assume a sample estimate of the advantage (which causes a biased gradient) and provided a convergence rate for this scenario as well.

__Meriths of the paper__

I was not able (both due to time constraints and also on the complexity of this work) to check in detail the proposed theoretical results. However, the work seems to follow a logical structure, and I find that most of the assumptions made are not particularly restrictive. Yet, the results presented, to the best of my knowledge, are new and important.


__A question__

The original paper from Peters et al, defined the stationary _undiscounted_ state distribution, i.e.,

$\sum_a \lambda_{s,a} = \sum_{a, s'}P_a(s|s') \lambda_{s', a},$

hence, defining an average reward setting.
Your theory is instead defined on the discounted setting. _How do you plan to overcome this limitation?_ Lemma~4, and other results in your paper seem to do not allow $\gamma \to 1$.

__Two sides notes__

Would be nice if the authors could support some of their theoretical statements with empirical analysis. This should be doable since the theory is defined for a tabular setting and does not require strong assumptions.

As a future work (as also mentioned in the paper) would be really nice to extend this to continuous state space using a linear assumption of the state distribution w.r.t. a set of (non-linear) features. Even though, I am convinced that the current state of the paper is ready to be accepted at NeurIPS.

__UPDATE__

I want to thank the authors for their detailed response, and all the other reviewers for the useful insights.
The authors addressed my concerns in a satisfying manner, however, I think that other reviewers raised very important points. I do agree that the work needs to be polished. In particular, it needs to be a clearer separation of contributions and background, as highlighted by other reviewers; and a better justification of choice made.

I do not detect problems with the theory.
For example, I think that the strong convexity required on $F$ is reasonable, or the choice of utilizing an oracle in section 5 and then relaxing later the assumption by assuming stochastic gradients. I do not think that the analysis is immediate (e.g., although NAC and REPS are linked, I agree with the author that the results on NAC cannot be transported easily on REPS). As I also remarked in my previous review, I think that the assumption of finite state-action spaces is still interesting (even though continuous state-action might be desirable) and helps to analyze the algorithm.

The fact that I do not detect the problems might be due to the fact that I do not have a strong background in convergence analysis (that is why I score my confidence with 3). I did not check derivations in detail.

In light of the other reviewer's analysis, and considering the points of agreement with them, I will downgrade the score to 8 (therefore, still a good submission!).





**Time Spent Reviewing:**

7

---

> ### Author Response · Authors · 2021-08-10
> **Response to Reviewer 3**
>
> We thank the reviewer for their kind comments. As the reviewer has noted, the objective of our paper is to prove convergence guarantees for Relative Entropy Search performed only in the dual objective of the regularized linear programming RL formulation.  Since the dual objective is convex, a variety of existing results can be leveraged to obtain provable guarantees when the model is known. When the model is not known, more effort needs to be put into coming up with ways for estimating the gradients. Since there is no readily available way to produce true unbiased stochastic gradients enjoying a small enough variance, we had to use a batch sampling approach. Our main contributions are:
>
> Lemma 4. We show that a candidate dual variable having small gradient gives rise to a policy whose true visitation distribution has large primal value $J_P$. This result is quite involved and by all means not obvious. It requires a deep understanding of bellman optimality. This is not an immediate corollary from existing results of convex optimization because the true value of the extracted policy corresponding to a candidate dual solution is not the same as the dual function evaluated at that candidate solution. Taking into account the extra “projection” step forms the basis of the proof of this Lemma. This result is not present in the existing literature and we consider it one of our main results.  We will make sure this is more prominently stated in our final manuscript.
>
> Lemma 5. We show a sharp upper bound on the norm of $\| \mathbf{v}^{\star} \|_\infty$. Our bound has the advantage of depending logarithmically on $|S|$, $|A|$, $\beta$ and $\rho$. This result is not present in the existing literature and we consider it one of our main results. We will make sure this is more prominently stated in our final manuscript.
>
> Theorems 2 and 3. We make use of a series of existing and new techniques in conjunction with the results of section 5.1 (particularly Lemma 5) to obtain these results. The analysis of stochastic gradient descent with biased and stochastic gradients for smooth convex functions is our own and it may be of independent interest (Lemma 9).
>
> We address the reviewer’s comments below.
>
> "Your theory is instead defined on the discounted setting. How do you plan to overcome this limitation? Lemma~4, and other results in your paper seem to do not allow $\gamma \rightarrow 1$." We believe our results would generalize to the average reward setting, with appropriate ergodicity assumptions (for example something like Assumption 1 in "Faster Saddle-point optimization for solving large-scale Markov decision processes", Joan Bas-Serrano, Gergely Neu) and after receiving each of our key results (Lemma 4, 5 and each of the convergence theorems) using the appropriate LP.
>
> "empirical analysis" We agree it would be a nice addition to our submission but we also believe that in line with the vast majority of accepted theoretical works it is not required. We do not see the immediate benefit of having an empirical section if it is not to show a specific phenomenon that can really complement the rates here derived. It is not obvious what would that be in our case.
>
> "future work" This is indeed an interesting avenue of future work. Nonetheless we also think this would be quite easy to achieve given the results derived in our paper.

---

> > ### Comment · Reviewer_3BWW · 2021-08-18
> > **Thanks**
> >
> > Dear authors,
> >
> > I want to notify you that I have read your reply, and I also read all the other reviews and replies.
> > I will update my review accordingly.
> >
> > Thank you for your time,
> > Best regards.

---

> > > ### Author Response · Authors · 2021-08-19
> > > **Thanks!**
> > >
> > > Thanks so much for taking time to review our paper.

---

### Official Review · Reviewer_EGrv · 2021-07-16

**Rating:** 4
**Confidence:** 4

**Summary:**

This paper goes back to a well known algorithm, Relative Entropy Policy Search (REPS) and provides convergence guarantees for REPS by exploiting its relationship of RL via duality. Recently, there has been a surge of work exploiting the dual formulation of RL objectives, and this paper exploits that relation to performance convergence analysis of the REPS algorithm. Besides, very recently, a number of works has provided convergence of policy search and policy gradient algorithms - and the contributions of this work can be seen in light of recent works. The major contributoon of this work is to provide convergence guarantees for the dual objective of REPS, relying solely on on-policy samples and using accelerated gradient ascent on the dual formulation (whereas prior works exploiting duality often need to rely on off-policy samples, estimating distribution ratios).


**Ethical Concerns:**

..

**Ethics Review Area:**

["I don’t know"]

**Limitations And Societal Impact:**

..

**Main Review:**

The key contribution of this work is to exploit duality in RL, and while relying solely on on-policy samples, the paper shows convergence guarantees of the dual objective based on advantage estimates, since the primal objective of REPS relies on the commonly used dual formulation in literature, based on state-action occupancy measures.

By exploiting literature from convex duality and its applications in RL,  and the similarity of optimizing value functions versus state-action occupancy measures, this paper proposes convergence results of REPS via looking at the objectives from the dual perspective. In the primal form, REPS optimizes an objective based on state-action distributions, and using the convex duality formulation, this paper in turn looks at the dual formulation of REPS based on advantage functions and proposes convergence results for both exact and stochastic gradient estimates. A useful corollary of the approach is that the paper shows convergence for a purely dual approach based on on-policy samples, in contrast to existing works which rely on off-policy samples with distribution ratio corrections when exploiting duality in RL.


Overall comments on the contributions of this work :

Firstly, an overall comment on clarity of this work and presentation : There has been a surge of recent papers establishing convergence of policy search/optimization methods, including natural policy gradients, TRPO, actor-critic etc, which startsfrom tabular to function approximation, and also ranges to sample complexity results for stochastic gradient estimates of actor-critic algorithms. To my understanding, this paper is in light of such recent works, where the key contribution is to establish convergence of the REPS algorithm. While doing this directly is difficult, since REPS optimizes an objective based on state-action occupancy measures - this paper exploits duality in RL to in turn establish convergence of the dual formulation based on advantage estimates.

In context of this, it seems to me that the presentation and clarity of this work is overly complicated. It can be written in a much more precise and clear formulation. While I appreciate the extensive details on RL via duality included in text - I think the main contribution of this work is not presented in a clear way. To me, almost the entire section of section 4 is a repetition of existing works, except explaining it in the context of REPS? Do we really need this? Instead of it, maybe clear objectives and explaining why the dual formulation is need along with the difficulty of optimizing REPS in the primal form would have been sufficient? Or am I missing something here? Can the authors point exactly on what's different in section 4 compared to existing works - I understand this is a background section establishing context of the work - but perhaps we do not need such in depth discussion, as it seems to make the paper overly complicated to understand. For example, I believe even equation 5, derived based on the Fenchel conjugate is known to the community now…

The first key result is established in section 5 Theorem 1? Is this correct? This is a re-establishment of existing work based on accelerated gradient ascent. Why do we have to resort to accelerated gradient ascent in this context for REPS? To me it seems like it is just another well known tool from the optimization community that is directly plugged in here? I expected a more clearer explanation of why we need this, compared to say using standard gradient estimates (since the main contribution is to analyse REPS via the dual formulation anyway).

As stated by the authors, the main difficulty is to ensure that the value function optimality guarantees will ensure convergence to a sub-optimal policy. This is similar to perhaps establishing a performance bound or sub-optimality gap for value functions and then deriving the convergence result based on that, for example, as in Agarwal et al., or Cen et al (for convergence of NPG with entropy regularization). What is so different here and what is the main difficulty? If we start with the dual objective of REPS based on advantages, why can we not similarly establish a performance difference bound for the advantage function, with an entropy regularization in the objective, and derive the convergence result from there?

The paper includes a lot of details for the analysis, but is written in a very poor manner and it is difficult to understand the key technical contributions. As stated before, the main trick is to use accelerated gradient ascent on the dual formulation of REPS based on advantage estimates. The majority of the appendix is known to the optimization community, and is a direct plug in for the REPS formulation. Can the authors please comment on the key pillars of the convergence result, and what the main technical contribution is?

My opinion is it seems there is a lot of tools used here from the optimization community establishing convergence results for the dual forms of the objective. While it seems like a lot of technical contributions, I could not find the key novel technical contribution used, specifically for the convergence of REPS? For example, if I do NPG/PG or TRPO with an accelerated gradient ascent - the same analysis would hold anyway? Yes, the paper is different since it uses the dual formulation, but this is already known to the community too. Therefore, can the authors comment on what are the major contributions from a theoretical viewpoint for this work?

Overall, my opinion is that the novelty of technical contributions is limited here, even though a lot of details are included (they are mostly from existing literature anyway). I agree analysing convergence of REPS is an important problem to consider - but I am not convinced there is a significantly new tool used for the analysis that would benefit the community. This paper seems more of a combination of different theoretical result blocks merged together and applied to REPS for deriving a convergence result. The length of the paper, including appendix, and the way the paper is written highlighting major contributions, therefore seems a bit misleading.

**Time Spent Reviewing:**

4

---

> ### Author Response · Authors · 2021-08-10
> **Response to Reviewer 2**
>
> As the reviewer has noted, the objective of our paper is to prove convergence guarantees for Relative Entropy Search performed only in the dual objective of the regularized linear programming RL formulation.  Since the dual objective is convex, a variety of existing results can be leveraged to obtain provable guarantees when the model is known. When the model is not known, more effort needs to be put into coming up with ways for estimating the gradients. Since there is no readily available way to produce true unbiased stochastic gradients enjoying a small enough variance, we had to use a batch sampling approach. Our main contributions are:
>
> Lemma 4. We show that a candidate dual variable having small gradient gives rise to a policy whose true visitation distribution has large primal value $J_P$. This result is quite involved and by all means not obvious. It requires a deep understanding of bellman optimality. This is not an immediate corollary from existing results of convex optimization because the true value of the extracted policy corresponding to a candidate dual solution is not the same as the dual function evaluated at that candidate solution. Taking into account the extra “projection” step forms the basis of the proof of this Lemma. This result is not present in the existing literature and we consider it one of our main results.  We will make sure this is more prominently stated in our final manuscript.
>
> Lemma 5. We show a sharp upper bound on the norm of $\| \mathbf{v}^{\star} \|_\infty$. Our bound has the advantage of depending logarithmically on $|S|$, $|A|$, $\beta$ and $\rho$. This result is not present in the existing literature and we consider it one of our main results. We will make sure this is more prominently stated in our final manuscript.
>
> Theorems 2 and 3. We make use of a series of existing and new techniques in conjunction with the results of section 5.1 (particularly Lemma 5) to obtain these results. The analysis of stochastic gradient descent with biased and stochastic gradients for smooth convex functions is our own and it may be of independent interest (Lemma 9).
>
> We address the reviewer’s comments below.
>
> “There has been a surge of recent papers establishing convergence of policy search/optimization methods, including natural policy gradients, TRPO ....” The comments of this paragraph and the previous two are correct.
>
> "REPS optimizes an objective based on state-action occupancy measures - this paper exploits duality in RL to in turn establish convergence of the dual formulation based on advantage estimates..." This is incorrect. The REPS algorithm is to optimize the dual formulation in terms of the advantage estimates and then use those estimates to derive a policy. Our contribution does not encompass these algorithmic details. Rather, our contribution focuses on showing that such an algorithm is correct, in the sense that a near-optimal advantage estimate actually yields a near-optimal policy (Lemma 4) and deriving the appropriate convergence rates for this near-optimal policy in various settings.
>
> “extensive details on RL via duality included in text … except explaining it in the context of REPS?t” We think this is absolutely necessary. The LP formulation of RL is not often treated by the mainstream RL community. Similarly, the derivation of the REPS objective as a result of the regularization of the primal RL LP is hardly an extremely well known fact for a vast majority of researchers . Since our results rely heavily on the LP derivation of the REPS objective, we think it would be extremely hard to present our results without this background being made available to the reader.
>
> “The first key result is established in section 5 Theorem 1? Is this correct?” This is incorrect (read the start of this response).
>
> “Why do we have to resort to accelerated gradient ascent in this context for REPS? “ The REPs objective is convex. Accelerated gradient descent for convex functions is a well understood optimization tool. It makes no sense to develop a new convergence algorithm when there are existing results that give us (as far as we know) the fastest rates available for said objectives. “I expected a more clearer explanation of why we need this”. This statement sounds to us as puzzling as asking why we use gradient descent for optimizing a function whose gradients we have access to? Because it works.
>
> “As stated by the authors, the main difficulty is to ensure that the value function optimality guarantees will ensure convergence to a sub-optimal policy.” This is correct. “This is similar to perhaps establishing a … or example, as in Agarwal et al., or Cen et al (for convergence of NPG with entropy regularization).” This is incorrect. Policy gradients work directly with a policy parametrization and therefore they do not suffer from the extra technical requirements (the projection step)  we had to deal with in Lemma 4. In REPS the variable we are optimizing for is $\mathbf{v}$. Translating between $\mathbf{v}$ space and policy space is not obvious. The resulting advantage functions are not related (at least not in an immediate way) to advantages of a specific policy. We plead the reviewer to think about this point for a bit and read the statement and proof of Lemma 4. This is a very important point.
>
> “As stated before, the main trick is to use accelerated gradient ascent on the dual formulation of REPS based on advantage estimates” As we have explained multiple times in this response this is just plain wrong. “The majority of the appendix is known to the optimization community, and is a direct plug in for the REPS formulation.“ This is also not true. Lemmas 4, 5 are completely novel and are an important part of our contribution. We will make sure this is better explained in the final version of our manuscript. Additionally, as far as we know, Lemma 9 is a novel proof of convergence for stochastic gradient descent for smooth functions in the presence of biased gradients. We did not find it anywhere in what the reviewer calls ‘standard known tricks of the optimization community’.
>
> “I could not find the key novel technical contribution used, specifically for the convergence of REPS?” Please refer to the contributions map at the beginning of this response. “For example, if I do NPG/PG or TRPO with an accelerated gradient ascent - the same analysis would hold anyway? “ This is completely wrong. The objectives in PG and TRPO are not convex, therefore one cannot use accelerated gradient descent, a method whose rates only apply to convex objectives. The magic behind REPS is that the dual objective we are trying to optimize here is actually convex so all the classic optimization tools apply to it. We do not see this as a limitation but rather as a positive thing because it means there is a lot of hope to understand optimization in RL much better via this algorithm. Oftentimes in research “simpler” solutions are not worse. “Therefore, can the authors comment on what are the major contributions from a theoretical viewpoint for this work?” We have explained them multiple times in the discussion above.
>
> “I agree analysing convergence of REPS is an important problem to consider - but I am not convinced there is a significantly new tool used for the analysis that would benefit the community.” This is again wrong. First, studying an algorithm for RL optimization for which existing tools in the optimization literature readily apply and that has often been neglected (perhaps because it hasn’t been theoretically analyzed yet) is an important step in itself if we want to design better algorithms for RL problems. Second, we strongly disagree with the general perception of the reviewer who is intent on believing our work is a simple application of off the shelf results in optimization. As we have explained throughout this rebuttal this statement is just not factual.

---

### Official Review · Reviewer_u5HF · 2021-07-17

**Rating:** 4
**Confidence:** 4

**Summary:**

The paper studies the convergence of the relative entropy policy search (REPS) method for solving LP-formulated MDP problems. The authors first assume the exact computation of gradients and prove the sublinear convergence of REPS regarding the regularized primal objective function and the policy. Second, the authors propose to estimate gradients using samples and prove the convergence or sample-complexity of sample-based REPS.

The main contribution is a convergence theory of REPS in both exact and stochastic settings.

**Ethical Concerns:**

No.

**Limitations And Societal Impact:**

Yes.

**Main Review:**

I have a few questions regarding originality and quality.

(1). In Section 5, the authors prove the properties of the Fenchel dual in Lemmas 1-3, relate the primal and dual variables in Lemma 4, and apply the existing gradient descent result to the Fenchel dual function in Lemma 7 and prove the convergence in Theorem 2. Most of these results are leveraged from convex optimization, e.g., [1]. An important step is Lemma 4 that gives a sufficient condition for getting a near-optimal policy. The rest is to apply the existing fast rate result (Theorem 1) to the smooth dual function under some problem assumptions. It is interesting that it does not work when $|S|=1$ due to Lemma 4. What is the technical reason? I roughly went through the proof of Lemma 4 and it still needs to explain more, e.g., weak duality, construction of $\bar{v}_s$.

(2). The stochastic gradient method in Section 6 takes the visitation data to estimate gradients in Algorithm 2. It is less discussed the issues with high variance and biased in estimates, and thereby highly-suboptimal error rate.

Regarding significance, I believe that the established convergence is useful in theory. Although the authors have established some interesting results, it still needs to be improved in many aspects (some are mentioned above).

(1). The established convergence could be sub-optimal. By assuming computing gradients exactly, the minimization of dual function is strongly convex and smooth, and it might be true for linear convergence. So, Theorem 2 can be loose. Any technical reasons? The sample-based convergence in Section 6 is highly sub-optimal.

(2). As mentioned, it is not practical to compute the REPS objective exactly. However, the theory in Section 5 still assumes the exact computation. What is the convergence if errors enter into Algorithm 1, e.g., step 1 does not have a perfect oracle? By adding this consideration, it would be useful to have a more practical convergence theory.

(3). It seems that your convergence theory relies on the strongly convex regularizer $F$. What happens if $F$ is convex? I feel that this will bring some technical challenges in analysis since the dual function is not smooth anymore.

(4). There is no evidence from the experimental results showing the effectiveness of the theory. This gap has to be filled to convince readers.

Regarding clarity, the paper does not seem to be well structured, e.g., results and definitions are mixed in Section 5, lemmas and theorems are mixed so that it is not easy for readers to tell which are key results quickly. I recommend putting thoughts into explaining your results. The proofs in Supplemental Material are also not nicely explained. Here are some other clarity questions:

(1). Line 22-30: it is a long technical description. Is this helpful to the motivation of this paper?

(2). Line 33: what do you mean by 'is likely not optimized fully'? any evidence or examples?

(3). Line 36: any reasons why you say 'it is far from clear'?

(4). Line 128, Line 132: is $\mathbb{E}^\pi$ the same?

(5). Line 133: what is the domain for $\lambda^\pi$?

(6). Line 414: what are $(i)$, $(ii)$, $(iii)$?

(7). Line 525: what is the contraction property of projections? Is it non-expansiveness? A typo is the norm $\Vert x_{t+1}-x_t \Vert^2$.

(8). Line 534, Line 540, Line 541: missing citations.


References:

[1] On the Fenchel Duality between Strong Convexity and Lipschitz Continuous Gradient. arXiv:1803.06573

============================

POST-REBUTTAL: Thank you for your response. I have read the response and other reviews.

**Time Spent Reviewing:**

20

---

> ### Author Response · Authors · 2021-08-10
> **Response to Reviewer 1**
>
> The objective of our paper is to prove convergence guarantees for Relative Entropy Search performed only in the dual objective of the regularized linear programming RL formulation.  Since the dual objective is convex, a variety of existing results can be leveraged to obtain provable guarantees when the model is known. When the model is not known, more effort needs to be put into coming up with ways for estimating the gradients. Since there is no readily available way to produce true unbiased stochastic gradients enjoying a small enough variance, we had to use a batch sampling approach. Our main contributions are:
>
> Lemma 4. We show that a candidate dual variable having small gradient gives rise to a policy whose true visitation distribution has large primal value $J_P$. This result is quite involved and by all means not obvious. It requires a deep understanding of bellman optimality. This is not an immediate corollary from existing results of convex optimization because the true value of the extracted policy corresponding to a candidate dual solution is not the same as the dual function evaluated at that candidate solution. Taking into account the extra “projection” step forms the basis of the proof of this Lemma. This result is not present in the existing literature and we consider it one of our main results.  We will make sure this is more prominently stated in our final manuscript.
>
> Lemma 5. We show a sharp upper bound on the norm of $\| \mathbf{v}^{\star} \|_\infty$. Our bound has the advantage of depending logarithmically on $|S|$, $|A|$, $\beta$ and $\rho$. This result is not present in the existing literature and we consider it one of our main results. We will make sure this is more prominently stated in our final manuscript.
>
> Theorems 2 and 3. We make use of a series of existing and new techniques in conjunction with the results of section 5.1 (particularly Lemma 5) to obtain these results. The analysis of stochastic gradient descent with biased and stochastic gradients for smooth convex functions is our own and it may be of independent interest (Lemma 9).
>
>
> We address each of the reviewer’s comments below.
>
> As we mention above, lots of the supporting results are based on existing techniques. We do not see this as a flaw but rather as a necessity. The case $|S| = 1$ is a corner case that would have made our analysis more cumbersome. It can also be handled at the cost of making the notation more burdensome. As we have mentioned above, the main result of section 5.1 is Lemma 5. This lemma is extremely important as it establishes a dependence on $\beta$ and $\rho$ that is logarithmic and not polynomial as less refined analysis may have yielded. We would encourage the reviewer to read this section of our paper in detail. We believe the proof is sufficiently well written for a knowledgeable reader to be able to understand its contents. That being said, we will make an effort to clean up the language and add extra explanations in the camera ready version.
>
> “It is less discussed the issues with high variance and biased in estimates, and thereby highly-suboptimal error rate.” This is wrong. This is clearly discussed in the lead up and proof of the convergence rates for this setting. Please read section 6 in detail. We are puzzled as to what suboptimal may mean here? Does the reviewer know of any other work that proves a better rate for the specific setup that we study? (dual REPS). We are not aiming to compete against any other method because we are the first to produce convergence rates for this problem.
>
> Also called (1). How are they suboptimal? Can the reviewer elaborate? Is there any other work the reviewer has in mind that has studied the same problem and has a much better rate? The reason why obtaining a rate that looks optimal to the reviewer (lower exponent) is hard to obtain is because obtaining true stochastic gradients is perhaps impossible in this setting, which means that the only remaining alternative is to use batch samples to estimate the ‘Q’ values and use a plug in estimator while controlling the gradients bias and variance. We leave the challenging problem of developing a ‘true’ stochastic gradient approach for future work.
>
> Also called (2).  This is an interesting question but completely orthogonal to the objective of our work. Similar results could be obtained by proving biased versions of the traditional gradient descent guarantees as we did for the stochastic gradients setting. (See Lemma 9)
>
> Also called (3). This is an interesting question. It is unclear what would happen in this case. In general this is not desirable since in this case it would be impossible to establish smoothness of the dual. This is why the type of regularizers used are typically strongly convex under some norm.
>
> Also called (4). We think this is a moot point. Our work is theoretical, most of the theoretical papers accepted to conferences do not present any experimental evaluation particularly when they are not necessary to accurately convey the paper’s results. Our contribution is theoretical in nature and would benefit little from any experimental evaluation. Therefore we think the use of “has” in this phrase (“This gap has to be filled to convince readers.”)  is monumentally misplaced since as the reviewer is probably well aware this is in general not a valid criticism of theory papers.
>
> We will fix all the typos the reviewer has found.

---

> ### Comment · Area_Chair_6VC7 · 2021-08-31
> **Would you please elaborate on your disagreements publicly?**
>
> Hi,
>
> Would you please elaborate on your points of disagreement with the authors' response in more detail?
>
> The authors cannot see our private discussions. It would help if you can specify them as a part of your POST-REBUTTAL comment or as a separate comment with an access setting that authors can read too, so if they need, they can clarify any remaining issues.
>
> Thank you,
> Area Chair

---

### Official Review · Reviewer_59AU · 2021-09-06

**Rating:** 7
**Confidence:** 4

**Summary:**

The paper studies the performance of the classic Relative Entropy Policy Search (REPS) algorithm of Peters et al. (2010) in the context of tabular reinforcement learning, focusing on the effect of optimization errors on the quality of the policy extracted from the solution. REPS is based on formulating the policy optimization problem as a regularized linear optimization problem which is equivalent to an unconstrained convex optimization problem, and calculating policy updates via approximately minimizing the dual function. The current paper sets out to understand the optimization issues surrounding solving these problems, and provides the following contributions:

- Showing that the policy suboptimality can be controlled in terms of the gradient norm associated with the approximate solution and some problem-dependent constants.
- Showing that the dual REPS objective is smooth, which implies that bounds on the additive optimization errors can be translated to bounds on the gradient norm at the approximate solution.
- Providing concrete performance bounds for an accelerated gradient descent method when the gradients of the objective can be evaluated exactly. This setting is rather unrealistic, but the results are insightful in that they shed light on the best attainable performance to be expected from this approach.
- Providing concrete performance bounds for a stochastic gradient descent method that uses a generative model to obtain estimates of the gradients. The challenge here is that one cannot straightforwardly obtain unbiased gradient estimates, and one has to take several sample transitions to make sure that the bias in the updates is small enough. The authors address this challenge thoroughly and obtain guarantees that are polynomial in the desired accuracy level.

**Limitations And Societal Impact:**

I foresee no negative societal impact or any limitations that are not adequately discussed in the paper itself.

**Main Review:**

The paper is generally well-written and addresses a very important problem. REPS and its variants have achieved significant empirical successes and its exact version is known to achieve excellent performance guarantees, but understanding the performance of its empirical version using sample transitions has remained challenging. This paper takes an important step into this direction, although with some limitations: the performance bounds are proved under rather restrictive assumptions (such as uniformly lower-bounded state visitation probabilities) and are unlikely to be tight in general. Also, I am almost sure that there could be more effective ways of reducing the bias of the gradient estimator other than taking several independent samples and throwing them away after one use. That said, I believe that the merits of the paper outweigh its limitations. Indeed, studying the performance of primal-only methods like REPS is likely to be a very important direction in the future of RL theory, and the paper does a great job outlining the key challenges and providing some elementary solution methods to overcome said challenges.

While I did not have the capacity to check the proof details, the overall proof strategy is clearly explained in the main body of the paper so I could verify that it is entirely plausible. Being familiar with the related literature, I can confidently say that there are quite a few original ideas in the analysis which definitely do not follow directly from previous work. Maybe one possible complaint is that the writing sometimes becomes a bit dense with technical details, but overall I had no serious trouble understanding the main message and the technical content.

All in all, I find it very likely that this paper will inspire some further progress in the field, and that the community will benefit from it being published at NeurIPS 2021.

**Time Spent Reviewing:**

2

---

### Decision · Program_Chairs · 2021-09-27

**Decision:**

Accept (Poster)

**Comment:**

This is a challenging paper, as the reviewers have a wide range of opinions about it.
We had a lot of discussions among the reviewers: there are more than 40 messages in total in the forum (including authors' responses), which is more than any other paper in my batch (most messages are private). Even though one of the reviewers changed their stance from the negative side to the positive side, the majority kept their initial viewpoints. In order to help with the final decision, I invited an expert emergency reviewer who provided a positive evaluation of this work. At the end of the day, we have 3 positive reviewers and 2 negative ones.

Several issues have been brought up. The core issues can be summarized into the following two categories. I explain the viewpoint of reviewers and provide my commentary:

1) Whether the results are novel and significant or not.

Some reviewers believe that the results are not novel or significant, yet others believe that they are.

In my viewpoint, the results are novel because they theoretically analyze one of the important RL algorithms (REPS) for the first time.
The provided convergence rates may or may not be optimal, as we do not know what the lower bound is. This is the first work with such an analysis, and I believe is an important one.

The negative reviewers' complaints regarding the novelty and significance are along these lines:

Concern: The results look suboptimal because the rates are not fast (perhaps compared to other policy search algorithms).

My take: Given that this is the first result for REPS, and we do not have a lower bound for the rate, this is not a real concern.

Concern: The results are not novel because similar papers analyzed similar algorithms. Are we going to analyze all algorithms using similar tools and claim novelty?

My take: I would argue that REPS is not any algorithm. It is one of the important RL algorithms in the past two decades, and its analysis is worthwhile.

Concern: The technical tools used in the analysis are standard; each component of the result (lemmas, etc.) have been proved elsewhere; the proofs are not creative applications of prior tools.

My take: I did not check the proofs myself, so I do not have a strong opinion about their content and technical novelty and creativity, but I do not think this is the most significant evaluation criteria for this type of paper.
Even if the proof techniques might be well-established, some of the results (Lemmas 4 and 5 and 9) appear to be novel.
How difficult is it to prove them? I do not know, but I do not think it matters much. They are not trivial corollaries of already known results.
Since NeurIPS is not a math venue, the novelty and creativity of the proof techniques are not the main consideration anyway, in my opinion.

I believe some of the reviewers have a different standard of novelty and significance in evaluating this paper from mine, and that might be the main reason for the difference in opinion.

2) The exposition should be improved.

The fact that some of the reviewers got confused about what is novel here and what is not, and the authors had to correct them in the rebuttal multiple times (a frustration that was expressed in their rebuttal too) should be a signal to the authors that maybe their paper is not written very well.
All reviewers spent a lot of time on this paper. Many of them are expert in areas closely related to the topic of this paper. The closeness of their expertise to this paper is way more than a typical RL researcher at NeurIPS. The reviews show that they tried hard to understand the paper, yet they occasionally misunderstood some important aspects of the paper, for example, the relationship of the results of this paper and the currently existing results in the literature.

This issue was not raised only by the negative reviewers. The expert emergency reviewer also believed that the writing sometimes becomes a bit dense with technical details. I shared this sentiment after reading the paper myself.

Although I realize that writing a paper that can be understood by everyone is challenging, especially a theoretical paper, I would recommend the authors to seriously consider the feedback provided by the reviewers to improve the exposition of their work.

**Evaluation:** Overall, given the novelty and significance of the paper, according to what I believe should be the novelty and significance criteria for a venue such as NeurIPS, and that we have enough positive and enthusiastic support from three reviewers, I recommend the acceptance of this paper.
I encourage the authors to revise the exposition of their paper and make it more accessible to a broader range of NeurIPS audience.